# QUANTUM (INSPIRED) $D^2$-SAMPLING WITH APPLICATIONS

**Poojan Chetan Shah and Ragesh Jaiswal**
Department of Computer Science and Engineering
Indian Institute of Technology Delhi
{cs1221594, rjaiswal}@cse.iitd.ac.in

## ABSTRACT

$D^2$-sampling is a fundamental component of sampling-based clustering algorithms such as $k$-means++. Given a dataset $V \subset \mathbb{R}^d$ with $N$ points and a center set $C \subset \mathbb{R}^d$, $D^2$-sampling refers to picking a point from $V$ where the sampling probability of a point is proportional to its squared distance from the nearest center in $C$. The popular $k$-means++ algorithm is simply a $k$-round $D^2$-sampling process, which runs in $O(Nkd)$ time and gives $O(\log k)$-approximation in expectation for the $k$-means problem. In this work, we give a quantum algorithm for (approximate) $D^2$-sampling in the QRAM model that results in a quantum implementation of $k$-means++ with a running time $\tilde{O}(\zeta^2 k^2)$. Here $\zeta$ is the aspect ratio (*i.e., largest to smallest interpoint distance*) and $\tilde{O}$ hides polylogarithmic factors in $N, d, k$. It can be shown through a robust approximation analysis of $k$-means++ that the quantum version preserves its $O(\log k)$ approximation guarantee. Further, we show that our quantum algorithm for $D^2$-sampling can be *dequantized* using the *sample-query access* model of Tang (2023). This results in a fast quantum-inspired classical implementation of $k$-means++, which we call *QI-k-means++*, with a running time $O(Nd) + \tilde{O}(\zeta^2 k^2 d)$, where the $O(Nd)$ term is for setting up the sample-query access data structure. Experimental investigations show promising results for QI-$k$-means++ on large datasets with bounded aspect ratio. Finally, we use our quantum $D^2$-sampling with the known $D^2$-sampling-based classical approximation scheme to obtain the first quantum approximation scheme for the $k$-means problem with polylogarithmic running time dependence on $N$.

## 1 INTRODUCTION

Data clustering and the $k$-means problem, in particular, have many applications in data processing. The $k$-means problem is defined as: given a set of $N$ points $V = \{v_1, ..., v_N\} \subset \mathbb{R}^d$, and a positive integer $k$, find a set $C \subset \mathbb{R}^d$ of $k$ centers such that the cost function,

$$\Phi(V, C) \equiv \sum_{v \in V} \min_{c \in C} D^2(v, c),$$

is minimised. Here, $D(v, c) \equiv \|v - c\|$ is the Euclidean distance between points $v$ and $c$. Partitioning the points based on the closest center in the center set $C$ gives a natural clustering of the data points. Due to its applications in data processing, a lot of work has been done in designing algorithms from theoretical and practical standpoints. The $k$-means problem is known to be NP-hard, so it is unlikely to have a polynomial time algorithm. Much research has been done on designing polynomial time *approximation* algorithms for the $k$-means problem. However, the algorithm used in practice to solve $k$-means instances is a heuristic, popularly known as the $k$-means algorithm (*not to be confused with the $k$-means problem*). This heuristic, also known as Lloyd's iterations Lloyd (1982), iteratively improves the solution in several rounds. The heuristic starts with an arbitrarily chosen set of $k$ centers. In every iteration, it (i) partitions the points based on the nearest center and (ii) updates the center set to the centroids of the $k$ partitions. In the classical computational model, it is easy to see that every Lloyd iteration costs $O(Nkd)$ time. This hill-climbing approach may get stuck at a local minimum or take a huge amount of time to converge, and hence, it does not give provable guarantees

on the quality of the final solution or the running time. In practice, Lloyd's iterations are usually preceded by the $k$-means++ algorithm Arthur & Vassilvitskii (2007), a fast sampling-based approach for picking the initial $k$ centers that also gives an approximation guarantee. So, Lloyd's iterations, preceded by the $k$-means++ algorithm, give the best of both worlds, theory and practice. Hence, it is unsurprising that a lot of work has been done on these two algorithms. The work ranges from efficiency improvements in specific settings to implementations in distributed and parallel models. With the quantum computing revolution imminent, it is natural to talk about quantum versions of these algorithms and quantum algorithms for the $k$-means problem in general.

Early work on the $k$-means problem within the quantum setting involved efficiency gains from quantizing Lloyd's iterations. In particular, Aïmeur et al. (2013) gave an $O(\frac{N^{3/2}}{\sqrt{k}})$ time algorithm for executing a single Lloyd's iteration for the Metric $k$-median clustering problem that is similar to the $k$-means problem. This was using the quantum minimum finding algorithm of Durr & Hoyer (1999). Using quantum distance estimation techniques assuming quantum data access, Lloyd et al. (2013) gave an $O(kN \log d)$ time algorithm for the execution of a single Lloyd's iteration for the $k$-means problem. More recently, Kerenidis et al. (2019) gave an approximate quantization of the $k$-means++ method and Lloyd's iteration assuming *QRAM data structure* Kerenidis & Prakash (2017) access to the data. Interestingly, the running time has only a polylogarithmic dependence on the size $N$ of the dataset. The algorithm uses quantum linear algebra procedures, and hence, there is dependence on certain parameters that appear in such procedures, such as the condition number $\kappa(V)$. Since Lloyd's iterations do not give an approximation guarantee, its quantum version is also a heuristic without a provable approximation guarantee. A quantum implementation of $k$-means++ was also given in Kerenidis et al. (2019). Note that $k$-means++ gives an approximation guarantee of $O(\log k)$ in expectation for the $k$-means problem. However, this approximation guarantee does not immediately extend to the quantum implementation of $k$-means++ since the quantum procedure introduces $D^2$*sampling* errors, and one must show that the approximation analysis is robust against those errors. This was indeed missing from Kerenidis et al. (2019). $D^2$-sampling is a fundamental component of $k$-means++. Given the dataset $V$ and a center set $C$, $D^2$-sampling refers to picking a point from $V$ such that the sampling probability is proportional to its squared distance from the nearest center in $C$. Starting with $C = \{\}$ and $D^2$-sampling and updating $C$ in $k$ rounds is precisely the $k$-means++ algorithm. In a recent line of work Bhattacharya et al. (2020a); Grunau et al. (2023), a robust version of $k$-means++ has been analyzed where the $D^2$-sampling probability has a multiplicative $(1 \pm \varepsilon)$ error. This is called *noisy $k$-means++*, and it is now known Grunau et al. (2023) that this algorithm gives $O(\log k)$ approximation. This shows that $k$-means++ is indeed robust against small relative $D^2$-sampling errors. This further means that the quantum implementation of $k$-means++ in Kerenidis et al. (2019) gives an approximation guarantee of $O(\log k)$.

In this work, we design quantum and quantum-inspired classical algorithms for $D^2$-sampling-based clustering algorithms. We start with $k$-means++. *Quantizing*[1] $k$-means++ reduces to quantizing the $D^2$-sampling step. Most of the ideas for quantizing $D^2$-sampling such as *coherent amplitude estimation*, *median estimation*, *distance estimation*, *minimum finding*, etc., have already been developed in previous works on quantum unsupervised learning (see Wiebe et al. (2015) for examples of several such tools). In particular, Kerenidis et al. (2019) used these tools to give a quantum implementation of $D^2$-sampling. We provide a similar quantum algorithm and then combine this with the known $O(\log k)$ approximation guarantee for noisy $k$-means++ to state the following result.

**Theorem 1.** *There is a quantum implementation of $k$-means++ that runs in time $\tilde{O}(\zeta^2 k^2)$ and gives an $O(\log k)$ factor approximate solution for the $k$-means problem with a probability of at least $0.99$. Here, $\tilde{O}$ hides $\log^2(Nd)$ and $\log^2(kd)$ terms.*[2]

**Dependence on aspect ratio $\zeta$:** Note that the running time depends on the aspect ratio $\zeta$, which is defined to be the ratio of the maximum to the minimum interpoint distance. For the remaining discussion, it is important to understand that the $k$-means problem remains NP-hard to approximate

---

[1]Quantizing is a term that is used to refer to designing a quantum version of an algorithm with a significant efficiency advantage over the corresponding classical algorithm.

[2]The output of $k$-means++ is a subset of data points, which can be stated as a subset of indices. If the description of centers is the expected output, the running time will also include a factor of $d$.

(beyond a fixed constant) even when restricted to instances where $\zeta$ upper bounded by a small constant. See Awasthi et al. (2015) for such an hardness reduction.[3]

The above result is another addition to the growing area of *Quantum Machine Learning (QML)* where the goal is to design quantum algorithms for machine learning problems with large speed-ups over their classical counterparts. There was a flurry of activity in QML where significant speedups were shown for problems such as recommendation systems Kerenidis & Prakash (2017). However, it was not clear whether such speedups could entirely be attributed to the quantum data access model (QRAM) upon which these works were built. The works of Tang (2019; 2022; 2023) gave much clarity on this aspect. These works showed that similar speedups (up to some polynomial factors) are achievable in the classical model using a classical counterpart of QRAM, now popularly known as the *sample-query (SQ) access data structure*. In other words, most QML algorithms could be *dequantized*. Interestingly, by setting up the sample-query access data structure, the dequantized algorithms give rise to *quantum-inspired* classical algorithms. The interesting property of such a quantum-inspired algorithm is that apart from the linear time spent on setting up the sample-query access data structure, which can be thought of as preprocessing time, the algorithm is extremely fast (otherwise, it would not have given the quantum speedup). Our second contribution is to dequantize $D^2$-sampling using Tang's SQ model. This turns out to be simple since the sample-query access model primarily involves sampling vectors based on squared $\ell_2$ norms, which is how $D^2$-sampling also works. Based on this, we give a quantum-inspired classical implementation of $k$-means++. The following theorem formally states our main result on this.

**Theorem 2.** *There is a classical implementation of $k$-means++ (which we call QI-$k$-means++) that runs in time $O(Nd) + O(\zeta^2 k^2 d \log k \log Nd)$. With probability at least $0.99$, QI-$k$-means++ outputs $k$ centers that give $O(\log k)$-approximation in expectation for the $k$-means problem.*

Note that the $O(Nd)$ term in the running time is for setting up the SQ data structure, which can be thought of as a preprocessing operation. This setting of the data structure can easily be parallelized on a multi-core processor. Once the data structure is available, the running time is sublinear in $N$. The linear dependence on the dimension $d$ can also be changed to sublinear at the cost of worsening the dependence on $\zeta$. This is through implementing a small relative error $D^2$-sampling procedure, which gives an implementation of the noisy $k$-means++ , which we know also gives $O(\log k)$-approximation. Since the aspect ratio $\zeta$ can be efficiently computed in advance Shamos & Hoey (1975); Dietzfelbinger et al. (1997), we can appropriately choose between the algorithms in Theorem 2 and Theorem 3. We state this result formally below.

**Theorem 3.** *There is a classical implementation of noisy $k$-means++ (which we call QI-noisy-$k$-means++) that runs in time $O(Nd) + O(\zeta^6 k^2 \log k \log Nd)$. With probability at least $0.99$, QI-noisy-$k$-means++ outputs $k$ centers that give $O(\log k)$-approximation in expectation for the $k$-means problem.*

Finally, we design a quantum approximation scheme for the $k$-means problem with polylogarithmic dependence on the size $N$ of the data set. We do this by quantizing the highly parallel, $D^2$-sampling-based approximation scheme of Bhattacharya et al. (2020b). An approximation scheme is an algorithm that, in addition to the dataset and $k$, takes an error parameter $\varepsilon > 0$ as input and outputs a solution with a cost within $(1 + \varepsilon)$ factor of the optimal. The $k$-means problem is NP-hard to efficiently approximate beyond an approximation factor of $1.07$ Cohen-Addad & C.S. (2019). The tradeoff in obtaining this fine-grained approximation of $(1 + \varepsilon)$ is that the running time of our algorithm has an exponential dependence on $k$ and error parameter $\varepsilon$. In the classical setting, such algorithms are categorized as Fixed Parameter Approximation Schemes (fpt-AS). Such $(1 + \varepsilon)$-approximation algorithms can have exponential running time dependence on the *parameter* (e.g., the number of clusters $k$ in our setting). The practical motivation for studying Fixed-Parameter Tractability (FPT) for computationally hard problems is that when the parameter is small (e.g., number of clusters $k \sim 5$), the running time is not prohibitively large. We state our main result as the following theorem, which we will prove in the remainder of the paper.

---

[3]The hardness of approximation argument in Awasthi et al. (2015) goes through the vertex cover problem. For a given graph $G = (V, E)$, the $k$-means instance includes $|E|$ points in a $|V|$-dimensional space. The point corresponding to an edge $(i, j)$ has 1 in coordinates $i, j$ and 0 everywhere else. It is clear that $\zeta = \sqrt{2}$ in this case.

**Theorem 4.** *Let $0 < \varepsilon < 1/2$ be the error parameter. There is a quantum algorithm that, when given QRAM data structure access to a dataset $V \in \mathbb{R}^{N \times d}$, runs in time $\tilde{O}\left(2^{\tilde{O}(\frac{k}{\varepsilon})} d \zeta^{O(1)}\right)$ and outputs a $k$ center set $C \in \mathbb{R}^{k \times d}$ such that with high probability $\Phi(V, C) \leq (1 + \varepsilon) \cdot OPT$. Here, $\zeta$ is the aspect ratio, i.e., the ratio of the maximum to the minimum distance between two given points in $V$.*[4]

We end on the note that the above quantum approximation scheme can also be dequantized in the sample-query-access model of Tang (2023).

## 1.1 COMPARISION WITH PREVIOUS WORK

There have been several fast classical implementations of $k$-means++. These implementations are fast under certain assumptions on the data set or achieved through trading efficiency with an approximation guarantee or both. Bachem et al. (2016b) gave a Monte Carlo Markov Chain (MCMC) based implementation of $k$-means++. However, this algorithm, called K-MC$^2$, preserves the $O(\log k)$ approximation guarantee of $k$-means++ only when the dataset satisfies some strong properties that are NP-hard to check. The MCMC-based algorithm was improved by the same authors in Bachem et al. (2016a), but the approximation guarantee of this algorithm had an additive term (in addition to the multiplicative $O(\log k)$ factor), which could possibly be large. More recently, a fast implementation based on the multi-tree embedding of the data was given by Cohen-Addad et al. (2020)

| Algorithm | Time Complexity | Approx. Guarantee | Key Parameters |
|---|---|---|---|
| QI-$k$-means++ (This Work) | $O(Nd) +$ $O(\zeta^2 k^2 d \log k \log N d)$ | $O(\log k)\Phi^k_{OPT}$ | $\zeta$ is the aspect ratio |
| K-MC$^2$ Bachem et al. (2016b) | $O(\gamma k^2 d \log k \beta)$, $^\dagger O(k^3 d \log^2 N \log k \beta)$ | $O(\log k)\Phi^k_{OPT}$ | $\gamma = 4\frac{\max_{x \in V} D^2(x, \mu)}{\Phi^k_{OPT}}$ $\mu$ is the dataset mean. |
| AFK-MC$^2$ Bachem et al. (2016a) | $O(Nd) + O\left(\frac{1}{\varepsilon} k^2 d \log \frac{k}{\varepsilon}\right)$, $^\dagger O(Nd) + O\left(k^3 d \log k\right)$ | $O(\log k)\Phi^k_{OPT} + \varepsilon\Phi^1_{OPT}$ | $\varepsilon \in (0, 1)$ controls runtime - solution quality tradeoff |
| RejectionSampling Cohen-Addad et al. (2020) | $O(N(d + \log N) \log \zeta d) +$ $O\left(kc^2 d^3 \log \zeta (N \log \zeta)^{O(1/c^2)}\right)$ | $O(c^6 \log k)\Phi^k_{OPT}$ | $\zeta$ is the aspect ratio. $c > 1$ controls runtime - solution quality tradeoff |

Table 1: Comparison of fast implementations of $k$-means++. $\Phi^k_{OPT}$ represents the optimal $k$-means cost of the dataset. $\dagger$ represents the runtime under the assumptions described in Bachem et al. (2016b).

Let us now compare our algorithm with K-MC$^2$ , AFK-MC$^2$ and RejectionSampling .

**Pre-processing**. QI-$k$-means++ takes $O(Nd)$ to setup the SQ data structures. AFK-MC$^2$ and RejectionSampling also require similar pre-processing cost for computing the initial distribution of the markov chain and for performing the multi-tree embedding respectively. An advantage of the SQ data structure is that it is very efficient to update, i.e., add or delete a point which costs $O(\log N)$ time. This is useful in scenarios where new data points get added periodically, and one needs to recluster the data.

**Runtime.** K-MC$^2$ , AFK-MC$^2$ have runtime which is sublinear in $N$ under certain assumptions. Their runtimes depend on the parameters $\gamma$ and $\varepsilon$ respectively, which can be thought of as being analogous to $\zeta$. It is important to note that computing an estimate of $\gamma$ involves solving the $k$-means problem itself, while $\zeta$ is efficiently computable. Even under assumptions, QI-$k$-means++ has a better dependence on $k$ ($k^2$ vs $k^3$). We see that QI-$k$-means++ and RejectionSampling have a tradeoff between $N$ ($\log N$ vs $N^{O(1/c^2)}$) and $\zeta$ ($\zeta^2$ vs $(\log \zeta)^{1+O(1/c^2)}$). Moreover, RejectionSampling has better dependence on $k$ ($k^2$ vs $k$) but worse dependence on $d$ ($d$ vs $d^3$).

So, in cases where the aspect ratio is not large (e.g., binarized data such as MNIST Deng (2012)) and $k$ is reasonably small, our implementation should be expected to run fast and give good solutions.

---

[4]The $\tilde{O}$ notation hides logarithmic factors in $N$. In the exponent, it hides logarithmic factors in $k$ and $1/\varepsilon$.

Note that unlike RejectionSampling and AFK-MC$^2$ , our results have no tradeoff between efficiency and approximation ratio. Even though our work is mainly theoretical in nature, we conduct some experimental investigations to see the running time advantage of QI-$k$-means++ over $k$-means++ (the classical implementation).

## 1.2 RELATED WORK

We have already discussed past research works on quantum versions of the $k$-means algorithm (i.e., Lloyd's iterations). This includes Aïmeur et al. (2013), Lloyd et al. (2013), and Kerenidis et al. (2019). All these have been built using various quantum tools and techniques developed for various problems in quantum unsupervised learning, such as coherent amplitude and median estimation, distance estimation, minimum finding, etc. See Wiebe et al. (2015) for examples of several such tools. We have also discussed previous works on fast implementations of the $k$-means++ algorithm. Other directions on quantum $k$-means includes *adiabatic* algorithms (e.g., Lloyd et al. (2013)) and algorithms using the *QAOA* framework (e.g., Otterbach et al. (2017); Farhi et al. (2014)). However, these are without provable guarantees. A recent work of Doriguello et al. Doriguello et al. (2023) improves the results of Kerenidis et al. (2019) on Lloyd's iterations and obtains a quantum algorithm with better running time dependence and removes the dependence on the data matrix dependent parameters such as the condition number. They also dequantize their algorithm in the sample-query access model of Tang Tang (2023). However, since Lloyd's iterations do not give any approximation guarantee, their algorithms also do not have a provable approximation guarantee. In another recent work Kumar et al. (2023), the quantum algorithm of Kerenidis et al. (2019) is used in a supervised setting to construct decision trees.

## 2 QUANTUM (INSPIRED) $D^2$-SAMPLING

In this section, we discuss the key ideas in the design of our quantum algorithm for $D^2$-sampling and its dequantization in the sample-query (SQ) access model of Tang Tang (2023). Let us first look at the main ideas of our quantum algorithm in the following subsection.

## 2.1 QUANTUM $D^2$-SAMPLING

We will work under the assumption that the minimum distance between two data points is 1, which can be achieved using scaling. This makes the aspect ratio $\zeta$ simply the maximum distance between two data points. We will use $i$ for an index into the rows of the data matrix $V \in \mathbb{R}^{N \times d}$, and $j$ for an index into the rows of the center matrix $C \in \mathbb{R}^{m \times d}$. We would ideally like to design a quantum algorithm that performs the transformation:

$$|i\rangle |j\rangle |0\rangle \to |i\rangle |j\rangle |D(v_i, c_j)\rangle \tag{1}$$

The known quantum tools such as *swap test* followed by *coherent amplitude estimation*, and *median estimation* allow one to obtain a state that is an approximation of the ideal state that is shown on the right in (1). Moreover, this can be done in time that has only polylogarithmic dependence on the input size. The details are provided in the Appendix. For the current high-level discussion, we will assume that the ideal state can be prepared. We would like to perform sampling from the $D^2$ distribution over $V$ with respect to the center set $C$. This means that we would like to sample an index $i \in [N]$ with probability:

$$\mathbf{Pr}[i] = \frac{\min_{j \in [m]} D(v_i, c_j)^2}{\sum_{i' \in [N]} \min_{j \in [m]} D(v_{i'}, c_j)^2} \tag{2}$$

So, if we can prepare the state:

$$\frac{1}{\sqrt{N}} \sum_{i \in [N]} |i\rangle \left| \min_{j \in [m]} D(v_i, c_j) \right\rangle,$$

then we can obtain a good estimate of the denominator of the right expression in (2), which is basically the $k$-means cost with respect to the current center set $C$. This follows from the fact that measuring the second register of the above state gives the distance of a random data point to its closest center in $C$. Repeatedly preparing the state and measuring helps obtain a close estimate of

the $k$-means cost. Now that the $k$-means cost (i.e., the denominator of (2)) is known, using a few more standard quantum transformations (controlled rotations), we can pull out the state of the second register as part of the amplitude of an ancilla qubit. In other words, an approximate version of the following quantum state can be prepared:

$$\frac{1}{\sqrt{N}} \sum_{i \in [N]} |i\rangle \left( \beta_i |0\rangle + \sqrt{1 - |\beta_i|^2} |1\rangle \right),$$

where $\beta_i = \frac{1}{Z} \min_{j \in [m]} D(v_i, c_j)$ and $Z$ is some appropriate normalization term. So, by measuring the above state and doing rejection sampling (ignoring the measurement when the ancilla qubit is 1), we obtain a $D^2$-sample. Some of the quantum states shown above are 'ideal' quantum states that we have used to convey the main ideas of the quantum algorithm. The real states will be close approximations of these states, and we need to carefully account for all the errors introduced. All the details are given in the Appendix. Finally, note that this also gives a quantum implementation of the $k$-means++ algorithm since the algorithm simply performs $D^2$-sampling in $k$ iterations while updating the center set.

In the following subsection, we discuss how to dequantize this quantum algorithm in the sample-query access model of Tang Tang (2023). This gives a quantum-inspired classical algorithm for $D^2$-sampling.

## 2.2 Quantum inspired $D^2$-sampling

Ewin Tang's thesis Tang (2023) nicely categorizes quantum machine learning (QML) algorithms into ones where the quantum advantage (i.e., polylogarithmic running time) is solely because of the quantum access to the data and the ones where it is not (e.g., the HHL algorithm Harrow et al. (2009)). Quantum access means that a (normalized) data vector $v := (v(1), ..., v(d))^T$ can be loaded onto the quantum workspace in $O(1)$ time as $|v\rangle = v(1) |1\rangle + ... + v(d) |d\rangle$. One implication of having the quantum state $|v\rangle$ is that $i$ can be sampled with probability $\frac{v(i)^2}{\sum_j v(j)^2}$ by simply making a measurement in the standard basis. One of the key insights in Tang's Thesis Tang (2023) is that QML algorithms that benefit solely from the quantum data access can be 'dequantized' if we work with an appropriate classical counterpart of the quantum data access that allows *sampling access* of the kind described above (in addition to the classical access to the data). This is called *sample-query access*, or SQ access in short. Dequantization means that a classical algorithm can simulate the steps of the quantum algorithm using SQ access with similar running time dependence. The nice property of the SQ access data structures is that it can be constructed classically in linear time. Note that linear time means that we lose the quantum advantage. However, if we keep this aside and count the SQ access data structure construction as preprocessing, the remaining computation has a similar advantage as that of the quantum algorithm. This is a level playing field with the QML algorithm that allows fair comparison, specifically given that setting up quantum access (called QRAM) also takes linear time. Such classical algorithms obtained by dequantization are called *quantum inspired* algorithms. Our quantum algorithms based on $D^2$-sampling fall into the category of QML algorithms that can be dequantized in Tang's SQ access model. This section discusses the quantum-inspired classical algorithm we obtain by dequantizing our quantum algorithm using the SQ access data structure. Naturally, we start the discussion by looking at the definition of the SQ access data structure.

**Definition 1** (Query access, Definition 1.1 in Tang (2023)). *For a vector $\vec{v} \in \mathbb{C}^n$, we have query access to $\vec{v}$, denoted by $Q(\vec{v})$, if for all $i \in [n]$, we can query for $\vec{v}(i)$. We use $\mathbf{q}(\vec{v})$ to denote the time (cost) of such a query.*

**Definition 2** (SQ-access to a vector, Definition 1.2 in Tang (2023)). *For a vector $\vec{v} \in \mathbb{C}^n$, we have sampling and query access to $\vec{v}$, denoted by $SQ(\vec{v})$, if we can:*

1. *query for entries in $\vec{v}$ as in $Q(\vec{v})$. The time cost is denoted by $\mathbf{q}(\vec{v})$.*

2. *obtain independent samples $i \in [n]$ where the probability of sampling $i$ is $\frac{\vec{v}(i)^2}{\|\vec{v}\|^2}$. This distribution is denoted by $\mathcal{D}_{\vec{v}}$. The time cost is denoted by $\mathbf{s}(\vec{v})$.*

3. *query for $\|\vec{v}\|$. The time cost is denoted by $\mathbf{n}(\vec{v})$.*

*Let $\mathbf{sq}(\vec{v}) \equiv \max \{\mathbf{q}(\vec{v}), \mathbf{s}(\vec{v}), \mathbf{n}(\vec{v})\}$ denote the time cost of SQ-access.*

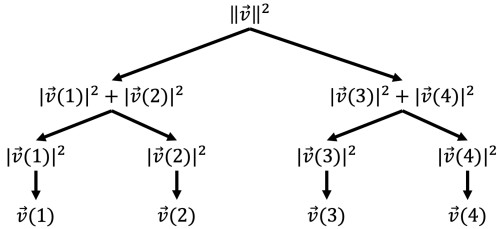

Figure 1: A tree data structure to enable sample-query access to an example vector of dimension $n = 4$. Index $i$ can be sampled with probability $\frac{|\vec{v}_i|^2}{\sum_j |v_j|^2}$ in $O(\log n)$ time by traversing down the tree.

A simple tree-based data structure (see Figure 1) supports sample-query access for vectors and matrices. The details can be found in Tang (2023). Here, we give a summary of the running time of various operations.

**Lemma 1** (Remark 4.12 in Tang (2023)). *There is a data structure for storing a vector $\vec{v} \in \mathbb{R}^n$ supporting the following operations:*

1. *Reading and updating an entry of $\vec{v}$ in $O(\log n)$ time. So, $\mathbf{q}(\vec{v}) = O(\log n)$.*

2. *Finding $\|\vec{v}\|^2$ in $O(1)$ time. So, $\mathbf{n}(\vec{v}) = O(1)$.*

3. *Sampling from $\mathcal{D}_{\vec{v}}$ in $O(\log n)$ time. So, $\mathbf{s}(\vec{v}) = O(\log n)$*

*It follows from the above that $\mathbf{sq}(\vec{v}) = O(\log n)$. Moreover, the time required to construct the data structure is $O(n)$.*

Let us outline the key ideas in the quantum-inspired classical algorithm in the SQ access model defined above. Let $v_1, ..., v_N$ denote the data vectors, i.e., $v_i = V(i, .)$. Suppose we have set up SQ access for the vectors $v_1, ..., v_N$, $(\|v_1\|^2, ..., \|v_N\|^2)^T$ (together denoted by $SQ(V)$), and $m$ centers $c_1, ..., c_m$ (denoted by $SQ(c_1), ..., SQ(c_m)$). Our goal is to set up SQ access for the vector $w$ defined as:

$$w := \left( \min_{j \in [m]} D(v_1, c_j), ..., \min_{j \in [m]} D(v_N, c_j) \right)^T \tag{3}$$

The reason is that once we have SQ access, $SQ(w)$, a sampling query returns an index with distribution $\mathcal{D}_w$, which is precisely the $D^2$-sampling distribution. We enable SQ access for $w$ in a sequence of steps:

$$(1)\ SQ(V), SQ(c_1), ..., SQ(c_m) \quad (2)\ SQ(u_1), ..., SQ(u_m) \quad (3)\ SQ(w).$$

Here $u_j$ is the vector defined as:

$$\forall i \in [N], u_j(i) = D(v_i, c_j) = \|v_i - c_j\|, \tag{4}$$

the vector of distances of $N$ data points from the $j^{th}$ center. The above steps are a gross simplification of the real steps to understand the high-level ideas and draw a parallel to the corresponding quantum steps. In actual implementation, we may not be able to enable sample-query access for $w$ but something known as *oversampling* and query access, which is a generalization SQ access. Much of the technical effort is spent designing these oversampling query accesses. These details are given in the Appendix.

Let us draw a parallel to the quantum algorithm to see that the above SQ-based algorithm is indeed quantum-inspired. The relevant quantum states involved in the $D^2$-sampling algorithm are given below:

$$(1)\ \forall i, |v_i\rangle, \forall j, |c_j\rangle, \frac{1}{\sqrt{N}} \sum_i |i\rangle |v_i\rangle \quad (2)\ \frac{1}{\sqrt{N}} \sum_i |i\rangle |D(v_i, c_1)\rangle ... |D(v_i, c_m)\rangle$$

$$(3)\ \frac{1}{\sqrt{N}} \sum_i |i\rangle \left| \min_{j \in [m]} D(v_i, c_j) \right\rangle \quad (4)\ \frac{1}{\sqrt{N}} \sum_{i \in [N]} |i\rangle \left( \beta_i |0\rangle + \sqrt{1 - |\beta_i|^2} |1\rangle \right)$$

The correspondence between quantum states and SQ accesses is apparent. So, every step in the quantum algorithm can be simulated using oversampling query access, which leads to a quantum-inspired classical algorithm since the SQ access framework is completely classical. Let us now discuss some key elements of the quantum-inspired classical implementation of $k$-means++ that results from the above quantum-inspired classical $D^2$-sampling.

## 2.3 QI-$k$-MEANS++

If we use the quantum-inspired classical $D^2$-sampling iteratively in $k$ rounds to pick $k$ centers, the resulting algorithm is a quantum-inspired classical implementation of $k$-means++, which we call the QI-$k$-means++ algorithm. This can be seen as a fast implementation of $k$-means++. We give a high-level outline of the algorithm and discuss some of its important properties.

---
**Algorithm 1** QI-$k$-means++$(V, k)$
---
1: Setup sample-query access for dataset $V$
2: $C \leftarrow$ random data point in $V$
3: **for** $i = 2$ to $k$ **do**
4:     Use sample-query access for $w$ (*which uses sample-query access for $u_1, ..., u_{i-1}$, which in turn uses sample-query access for $V$ and $c_1, ..., c_{i-1}$*) to $D^2$-sample a center $c$. The vectors $w$ and $u_1, ..., u_k$ are as defined in (3) and (4).
5:     $C \leftarrow C \cup \{c\}$;
6: **end for**
7: **return** $C$
---

Setting up the sample-query access for the dataset $V \in \mathbb{R}^{N \times d}$ costs $O(Nd)$ time. This can be thought of as the preprocessing time. Setting up the sample-query access is highly parallelizable, as can be seen from the tree-based data structure in Figure 1. On a shared memory, multi-processor system with $M$ processing units, the task can be done in $O(Nd/M)$ time. After the preprocessing in step (1), the cost of the remaining $D^2$-sampling steps is $\tilde{O}(\zeta^2 k^2 d)$. Here, again, it is possible to parallelize on a multi-processor system since much of the time in $D^2$-sampling is spent on waiting for rejection sampling to succeed. The rejection sampling can be performed independently on multiple processors until one of the processing units succeeds. This cuts down the time to $\tilde{O}(\zeta^2 k^2 d/M)$. We implement the QI-$k$-means++ algorithm to compare the $k$-means cost and time with the classical implementation of $k$-means++ that has a running time $O(Nkd)$.

## 3 A QUANTUM APPROXIMATION SCHEME

We convert the $D^2$-sampling-based approximation scheme of Bhattacharya et al. (2020b) to a Quantum version. The approximation scheme is simple and highly parallel, which can be described in a few lines.

---
**Algorithm 2** Approximation Scheme
---
**Input**: Dataset $V$, integer $k > 0$, and error $\varepsilon > 0$
**Output**: A center set $C'$ with $\Phi(V, C') \leq (1 + \varepsilon)OPT$
- (*Constant approximation*) Find a center set $C$ that is a constant factor approximate solution. An $(\alpha, \beta)$ *pseudo-approximate solution*, for constants $\alpha, \beta$, also works.
- (*$D^2$-sampling*) Pick a set $T$ of $poly(\frac{k}{\varepsilon})$ points independently from the dataset using $D^2$-sampling with respect to the center set $C$.
- (*All subsets*) Out of all $k$-tuples $(S_1, ..., S_k)$ of (multi)subsets of $T \cup \{$copies of points in $C\}$, each $S_i$ of size $O(\frac{1}{\varepsilon})$, return $(\mu(S_1), ..., \mu(S_k))$ that gives the least $k$-means cost. Here, $\mu(S_i)$ denotes the centroid of points in $S_i$.
---

We will discuss the quantization of the above three steps of the approximation scheme of Bhattacharya et al. (2020b), thus obtaining a quantum approximation scheme. [5]

---
[5]Steps (2) and (3) in the algorithm are within a loop for probability amplification. For simplicity, this loop is skipped in this high-level description.

**(Constant approximation)** The first step requires finding a constant factor approximate solution for the $k$-means problem. Even though several constant factor approximation algorithms are known, we need one with a quantum counterpart that runs in time that is polylogarithmic in the input size $N$. One such algorithm is the $k$-means++ seeding algorithm Arthur & Vassilvitskii (2007). Kerenidis et al. (2019) give an approximate quantum version of $D^2$-sampling. The approximation guarantee of the $k$-means++ algorithm is $O(\log k)$ instead of the constant approximation required in the approximation scheme of Bhattacharya et al. (2020b). It is known from the work of Aggarwal et al. (2009) that if the $D^2$-sampling in $k$-means++ is continued for $2k$ steps instead of stopping after sampling $k$ centers, then we obtain a center set of size $2k$ that is a $(2, O(1))$-pseudo approximate solution. This means that this $2k$-size center set has a $k$-means cost that is some constant times the optimal. Such a pseudo-approximate solution is sufficient for the approximation scheme of Bhattacharya et al. (2020b) to work. We show that the pseudo-approximation guarantee of Aggarwal et al. (2009) also holds when using the approximate quantum version of the $D^2$-sampling procedure.

**($D^2$-sampling)** The second step of Bhattacharya et al. (2020b) involves $D^2$-sampling, which we already discussed how to quantize. This is no different than the $D^2$-sampling involved in the $k$-means++ algorithm of the previous step. The sampling in this step is simpler since the center set $C$, with respect to which the $D^2$-sampling is performed, does not change (as is the case with the $k$-means++ algorithm.)

**(All subsets)** Since the number of points sampled in the previous step is $poly(\frac{k}{\varepsilon})$, we need to consider a list of $\left(\frac{k}{\varepsilon}\right)^{\tilde{O}(\frac{k}{\varepsilon})}$ tuples of subsets, each giving a $k$-center set (*a tuple $(S_1, ..., S_k)$ defines* $(\mu(S_1), ..., \mu(S_k))$). We need to compute the $k$-means cost for every $k$ center set in the list and then pick the one with the least cost. We give quantization of the above steps. [6]

Note that the quantization of the classical steps of Bhattacharya et al. (2020b) will incur precision errors. So, we first need to ensure that the approximation guarantee of Bhattacharya et al. (2020b) is robust against small errors in distance estimates, $D^2$-sampling probabilities, and $k$-means cost estimates. We must carefully account for errors and ensure that the quantum algorithm retains the $(1 + \varepsilon)$ approximation guarantee of the robust version of Bhattacharya et al. (2020b). We give the detailed analysis in the Appendix.

## 4 EXPERIMENTS

We provide an experimental investigation of the QI-$k$-means++ algorithm. We implemented our code in C++ for both $k$-means++ and QI-$k$-means++ and the results are averaged over 5 runs. No pre-processing/dimensionality reductions were done on any of the datasets used. The experiments were performed on an AMD Ryzen 9 5900HX 4.68 GHz 8 Core processor (parallelization on multicore was not used). We must set up the sample and query data structure for the dataset only once to be able to perform seeding for any value of $k$, with each run taking time only polylogarithmic in $N$. Specifically, if we wanted to perform seeding for $q$ different values of $k$, say $\{k_1, ..., k_q\}$, then the total runtime for QI-$k$-means++ would be $O(Nd) + \tilde{O}(\zeta^2 d \sum_{j \in [q]} k_j^2)$ as compared to $O(Nd \sum_{j \in [q]} k_j)$ for $k$-means++. This is one reason to consider the $O(Nd)$ term as pre-processing.

We study the runtime (including setup time for the sample and query data structure) of QI-k-means++ on two extreme types of datasets to demonstrate its possible use cases. The first is binarized MNIST Salakhutdinov & Murray (2008) (70,000 data points, each being a $28 \times 28$ pixel image . Each pixel has a value of either 0 or 1, as opposed to the original MNIST, which had values from 0 to 255) and the second is IRIS Fisher (1988) (150 data points with four feature values). The plots in Figures 2 and 3 show the *cumulative runtime* (i.e., the total time for setting up and calculating cluster centers for all values of $k$ up to a certain value).

**Advantageous regime**: For a dataset with a large number of points and small aspect ratio (for example, binary MNIST), we find that for one iteration of seeding, our algorithm performs slower but quickly catches up to be significantly faster when we require seeding to be done for multiple

---

[6]Note that when picking the center set with the least cost, we can get quadratic improvement in the search for the best $k$-center set using quantum search. Given that the search space is of size $\left(\frac{k}{\varepsilon}\right)^{\tilde{O}(\frac{k}{\varepsilon})}$, this results only in a constant factor improvement in the exponent. So, for simplicity, we leave out the quantum search from the discussion.

values of $k$ (which is generally the case in practice since in the unsupervised setting, the optimal value of $k$ is not known beforehand). This is because most of the time is spent in setting up the data structure, after which the sampling becomes significantly faster. For example, see the left plot for the MNIST dataset. Notice that the cumulative runtime for QI-$k$-means++ is almost constant, while for $k$-means++, it does not scale well due to the multiplicative dependence on $N$.

**Disadvantageous regime**: For a dataset with a small number of points and a large aspect ratio, we find that the runtime is dominated by the sampling term and due to the quadratic dependence on $k$ and $\zeta$, QI-$k$-means++ may not scale well. For example, see the right plot for the IRIS dataset.

We give additional experiments on larger datasets and a comparative analysis with AFKMC$^2$ Bachem et al. (2016a) in Section F of the Appendix.

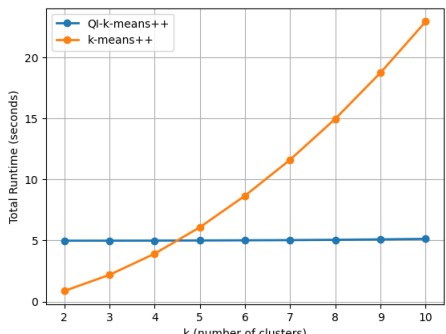

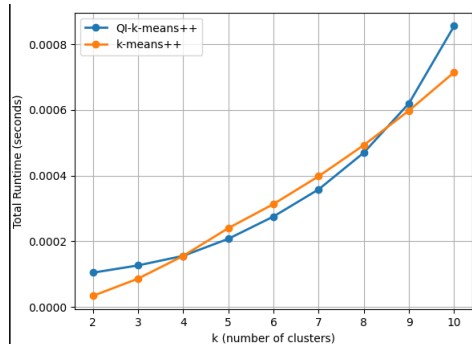

Figure 2: Cumulative runtime plot for MNIST          Figure 3: Cumulative runtime plot for IRIS

| k | 2 | 3 | 4 | 5 | 6 | 7 | 8 | 9 | 10 |
|---|---|---|---|---|---|---|---|---|---|
| QI-k-means++ | 8.43 | 8.13 | 7.81 | 7.43 | 7.29 | 7.09 | 6.79 | 6.78 | 6.66 |
| k-means++ | 8.35 | 8.10 | 7.74 | 7.38 | 7.44 | 7.17 | 7.12 | 6.91 | 6.83 |

Table 2: Clustering cost for binarized MNIST (costs are scaled down by a factor of $10^6$)

| k | 2 | 3 | 4 | 5 | 6 | 7 | 8 | 9 | 10 |
|---|---|---|---|---|---|---|---|---|---|
| QI-k-means++ | 538.82 | 253.41 | 112.53 | 96.04 | 83.72 | 64.92 | 58.62 | 56.25 | 49.92 |
| k-means++ | 773.26 | 227.50 | 115.33 | 93.45 | 83.21 | 60.14 | 57.98 | 53.99 | 50.35 |

Table 3: Clustering cost for IRIS (costs are rounded to 2 decimal places)

## 5  CONCLUSION AND FUTURE WORK

In this work, we used $D^2$-sampling framework to design quantum algorithms for the $k$-means problem. We also gave their classical counterparts obtained through the dequantization framework of Tang (2023). These algorithms have logarithmic dependence on the number of points in the dataset $N$, after appropriate linear time pre-processing for the classical ones. Our algorithms depend on the parameter $\zeta$, which is the aspect ratio of the dataset. Our algorithm QI-$k$-means++ is an addition to the collection of fast implementations of the $k$-means++ seeding. Interestingly, the performance of all of these algorithms depend on certain *aspect ratio like quantities*. Even for a simple dataset where the clusters are located very far apart from each other, the parameter $\zeta$ (for QI-$k$-means++ and RejectionSampling ) and the ratio of optimal 1-means cost to the optimal $k$-means cost (for K-MC$^2$ and AFK-MC$^2$ ) can become quite large. We think that improving the dependence on these quantities and understanding the tradeoff between them and the data size are interesting open problems.

## 6  ACKNOWLEDGEMENTS

This work was partially supported by the CSE Research Acceleration Fund of IIT Delhi.

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

## A  QUANTUM PRELIMINARIES

In this section , we briefly describe the core notions of quantum computing and refer to Nielsen & Chuang (2010) for a more formal and complete overview of quantum computing.

A qubit (*quantum bit*) is represented by a unit vector $|\psi\rangle = \alpha_0 |0\rangle + \alpha_1 |1\rangle$ where $\alpha_0, \alpha_1 \in \mathbb{C}$ are complex numbers such that $|\alpha_0|^2 + |\alpha_1|^2 = 1$ and $|0\rangle = \begin{pmatrix} 1 \\ 0 \end{pmatrix}$ and $|1\rangle = \begin{pmatrix} 0 \\ 1 \end{pmatrix}$ represent the standard basis of the vector space $\mathbb{C}^2(\mathbb{C})$ i.e, the vector space whose elements are pairs of complex numbers with the underlying field being $\mathbb{C}$.

Similar to a single qubit, an $n$-qubit system is represented by a unit vector in $(\mathbb{C}^2(\mathbb{C}))^{\otimes n}$ i.e, as $|\psi\rangle = \sum_{j=0}^{2^n-1} \alpha_j |j\rangle$ where the *amplitudes* are complex numbers satisfying $\sum_{j=0}^{2^n-1} |\alpha_j|^2 = 1$.

A quantum algorithm performs computation on qubits by applying *quantum gates* which are represented as unitary matrices. For example, the hadamard gate $H = \frac{1}{\sqrt{2}} \begin{pmatrix} 1 & 1 \\ 1 & -1 \end{pmatrix}$ can perform the mapping $|0\rangle \mapsto \frac{1}{\sqrt{2}}(|0\rangle + |1\rangle)$ and $|1\rangle \mapsto \frac{1}{\sqrt{2}}(|0\rangle - |1\rangle)$.

A quantum state can also be *measured* in the standard basis. For example, when $|\psi\rangle = \alpha_0 |0\rangle + \alpha_1 |1\rangle$ is measured, we obtain 0 with probability $|\alpha_0|^2$ and 1 with probability $|\alpha_1|^2$. Similarly, when $|\psi\rangle = \sum_{j=0}^{2^n-1} \alpha_j |j\rangle$ is measured, we observe the outcome $j = j_0 j_1 \ldots j_{n-1}$, the binary representation of $j$ with probability $|\alpha_j|^2$.

In the context of QML, the input data is assumed to be provided in the QRAM data structure. We refer to Kerenidis & Prakash (2017) for the specific details of QRAM implementation. Any vector $v \in \mathbb{R}^d$ can be encoded as a quantum state using $\lceil \log d \rceil$ qubits as $|v\rangle = \frac{1}{\|v\|} \sum_{j=0}^{d-1} v_j |j\rangle$.

## B   QUANTUM $D^2$-SAMPLING (PROOF OF THEOREM 1)

We will work under the assumption that the minimum distance between two data points is 1, which can be achieved using scaling. This makes the aspect ratio $\zeta$ simply the maximum distance between two data points. We will use $i$ for an index into the rows of the data matrix $V \in \mathbb{R}^{N \times d}$, and $j$ for an index into the rows of the center matrix $C \in \mathbb{R}^{k \times d}$. We would ideally like to design a quantum algorithm that performs the transformation:

$$|i\rangle |j\rangle |0\rangle \rightarrow |i\rangle |j\rangle |D(v_i, c_j)\rangle$$

Let us call the state on the right $|\Psi_{ideal}\rangle$. This is an ideal quantum state for us since $|\Psi_{ideal}\rangle$ helps to perform $D^2$-sampling and to find the $k$-means cost of clustering, which are the main components of the approximation scheme of Bhattacharya et al. (2020b) that we intend to use. One caveat is that we will only be able to perform the following transformation (instead of the abovementioned transformation)

$$|i\rangle |j\rangle |0\rangle \rightarrow |i\rangle |j\rangle |\psi_{i,j}\rangle ,$$

where $|\psi_{i,j}\rangle$ is an approximation for $|\mathcal{D}(v_i, c_j)\rangle$ in a sense that we will make precise below. We will use $|\Psi_{real}\rangle$ to denote the state $|i\rangle |j\rangle |\psi_{i,j}\rangle$. This state is prepared using tools such as *swap test* followed by *coherent amplitude estimation*, and *median estimation*. Since these tools and techniques are known from previous works Wiebe et al. (2015); Lloyd et al. (2013); Kerenidis et al. (2019), we summarise the discussion (see Section 4.1 and 4.2 in Kerenidis et al. (2019)) in the following lemma.

**Lemma 2** (Kerenidis et al. (2019) and Wiebe et al. (2015)). *Assume for a data matrix $V \in \mathbb{R}^{N \times d}$ and a center set matrix $C \in \mathbb{R}^{t \times d}$ that the following unitaries: (i) $|i\rangle |0\rangle \rightarrow |i\rangle |v_i\rangle$, (ii) $|j\rangle |0\rangle \rightarrow |j\rangle |c_j\rangle$ can be performed in time $T$ and the norms of the vectors are known. For any $\Delta > 0$, there is a quantum algorithm that in time $O\left(\frac{T \log \frac{1}{\Delta}}{\varepsilon}\right)$ computes:*

$$|i\rangle |j\rangle |0\rangle \rightarrow |i\rangle |j\rangle |\psi_{i,j}\rangle ,$$

*where $|\psi_{i,j}\rangle$ satisfies the following two conditions for every $i \in [N]$ and $j \in [t]$:*

*(i)* $\left\| |\psi_{i,j}\rangle - \left|0^{\otimes \ell}\right\rangle \left|\tilde{D}(v_i, c_j)\right\rangle \right\| \leq \sqrt{2\Delta}$, *and*

*(ii) For every $i, j$, $\tilde{D}(v_i, c_j) \in (1 \pm \varepsilon) \cdot D(v_i, c_j)$.*

In the subsequent discussions, we will use $T$ as the time to access the *QRAM data structure* Kerenidis & Prakash (2017), i.e., for the transitions $|i\rangle |0\rangle \rightarrow |i\rangle |v_i\rangle$ and $|j\rangle |0\rangle \rightarrow |j\rangle |c_j\rangle$ as given in the above lemma. This is known to be $T = O(\log^2 (Nd))$. Moreover, the time to update each entry in this data structure is also $T = O(\log^2 (Nd))$. This is the logarithmic factor that is hidden in the $\tilde{O}$ notation. In the following subsections, we discuss the utilities of $|\Psi_{real}\rangle$ for the various components of the approximation scheme of Bhattacharya et al. (2020b). During these discussions, it will be easier to see the utility first with the ideal state $|\Psi_{ideal}\rangle$ before the real state $|\Psi_{real}\rangle$ that can actually be prepared. We will see how $|\Psi_{real}\rangle$ is sufficient within a reasonable error bound.

### B.1   FINDING DISTANCE TO CLOSEST CENTER

Let us see how we can estimate the distance of any point to its closest center in a center set $C$ with $t \leq k$ centers. We can use the transformation $|i\rangle |j\rangle |0\rangle \rightarrow |i\rangle |j\rangle |D(v_i, c_j)\rangle$ to prepare the following state for any $i$:

$$|i\rangle |D(v_i, c_1)\rangle |D(v_i, c_2)\rangle \ldots |D(v_i, c_t)\rangle$$

We can then iteratively compare and swap pairs of registers to prepare the state $|i\rangle \left|\min_{j\in[t]} D(v_i, c_j)\right\rangle$. If we apply the same procedure to $|i\rangle |\psi_{i,1}\rangle ... |\psi_{i,t}\rangle$, then with probability at least $(1 - 2\Delta)^t$, the resulting state will be $|i\rangle \left|\min_{j\in[t]} \tilde{D}(v_i, c_j)\right\rangle$. So, the contents of the second register will be an estimate of the distance of the $i^{th}$ point to its closest center in the center set $C$. This further means that the following state can be prepared with probability at least $(1 - 2\Delta)^{Nt}$:[7]

$$|\Psi_C\rangle \equiv \frac{1}{\sqrt{N}} \sum_{i=1}^{N} |i\rangle \left|\min_{j\in[t]} \tilde{D}(v_i, c_j)\right\rangle.$$

This quantum state can be used to find the approximate clustering cost of the center set $C$, which we discuss in the following subsection. However, before we do that, let us summarise the main ideas of this subsection in the following lemma.

**Lemma 3.** *There is a quantum algorithm that, with probability at least $(1 - 2\Delta)^{Nt}$, prepares the quantum state $|\Psi_C\rangle$ in time $O\left(\frac{Tt\log\frac{1}{\Delta}}{\varepsilon}\right)$.*

### B.2 $D^2$-SAMPLING

$D^2$-sampling from the point set $V$ with respect to a center set $C \in \mathbb{R}^{t\times d}$ with $t$ centers, samples $v_i$ with probability proportional to $\min_{j\in[t]} D^2(v_i, c_j)$. Let us see if we can use our state $|\Psi_C\rangle = \frac{1}{\sqrt{N}} \sum_{i=1}^{N} |i\rangle \left|\min_{j\in[t]} \tilde{D}(v_i, c_j)\right\rangle$ is useful to perform this sampling. If we can pull out the value of the second register as the amplitude, then the measurement will give us close to $D^2$-sampling. This is possible since we have an estimate of the clustering cost from the previous subsection. We can use controlled rotations on an ancilla qubit to prepare the state:

$$|\Psi_{sample}\rangle \equiv \frac{1}{\sqrt{N}} \sum_{i=1}^{N} |i\rangle \left(\beta_i |0\rangle + \sqrt{1 - |\beta_i|^2} |1\rangle\right),$$

where $\beta_i = \frac{\min_{j\in[t]} \tilde{D}(v_i, c_j)}{2\zeta}$. Note that $\beta_i \leq 1$ since $\zeta$ is the aspect ratio, and by scaling, we assume that $\min_{i,j} D(v_i, v_j) = 1$.[8] Note that this suggests we must know the aspect ratio (or an upper bound on the ratio), which should be considered an input parameter. This is a standard assumption in parameterised quantum machine learning algorithms such as ours (*for example, in the HHL algorithm, a bound on the condition number of the matrix is used as a parameter*). The probability of measurement of $(i, 0)$ is $\frac{\min_{j\in[t]} \tilde{D}(v_i, c_j)^2}{4N\cdot\zeta^2} \geq \frac{1}{8N\zeta^2}$. Since we do rejection sampling, ignoring $(.,1)$'s that are sampled with probability $\leq \left(1 - \frac{1}{8\zeta^2}\right)$, we end up sampling with a distribution where the conditional probability of sampling $i$ is $\frac{\min_{j\in[t]} \tilde{D}(v_i, c_j)^2}{\tilde{\Phi}(V,C)} \in (1 \pm \varepsilon)^4 \cdot \frac{\min_{j\in[t]} D(v_i, c_j)^2}{\Phi(V,C)}$. Moreover, the rejection sampling gives us a usable sample with high probability in $m = O(\zeta^2 \ln 10N)$ rounds of sampling. This means that points get sampled with a probability close to the actual $D^2$-sampling probability. As mentioned earlier, this is sufficient for the approximation guarantees of Bhattacharya et al. (2020b) to hold. We summarise the observations of this section in the next lemma.

**Lemma 4.** *Let $0 < \delta \leq 1$ and let $m = \zeta^2 \ln 10N$. Given a dataset $V \in \mathbb{R}^{N\times d}$ and a center set $C \in \mathbb{R}^{t\times d}$ in the QRAM model, there is a quantum algorithm that runs in time $\tilde{O}\left(\frac{Ttm\log\frac{1}{\Delta}}{\delta}\right)$ and with probability at least $(1 - 1/5N) \cdot (1 - 2\Delta)^{Ntm}$ outputs a sample with $\mathcal{D}^2$ distribution such that the distance function $\mathcal{D}$ is $\delta$-close to $D$.*

*Proof.* The proof follows from Lemma 3 and the preceding discussion. □

---

[7]The state prepared is actually $\frac{1}{\sqrt{N}} \sum_{i=1}^{N} |i\rangle \left(\alpha \left|\min_{j\in[t]} \tilde{D}(v_i, c_j)\right\rangle + \beta |G\rangle\right)$ with $|\alpha|^2 \geq (1 - 2\Delta)^{Nk}$. However, instead of working with this state, subsequent discussions become much simpler if we assume that $|\Psi_C\rangle$ is prepared with probability $|\alpha|^2$.

[8]In case scaling only ensures $\min_{i,j} D(v_i, v_j) \geq 1$ (e.g., vectors with integer coordinates), we can use a bound on $\max_{i,j} D(v_i, v_j)$ as the parameter instead of the aspect ratio.

The above lemma says that for $\Delta = \tilde{O}(\frac{\varepsilon^2}{Nt\zeta^2})$, we obtain the required $D^2$-sample with probability $\geq (1 - 1/N)$. We can now give proof of Theorem 1, assembling the quantum tools for $D^2$-sampling in this section. Note that we obtain samples from $\mathcal{D}^2$ distribution such that $\mathcal{D}$ is close to $D$. Using our quantum sampling algorithm to pick $k$ centers in $k$ rounds is not the classical $k$-means++ algorithm that requires that the samples be picked with $D^2$ distribution and not something close. The resulting algorithm, however, has been analysed in the literature. If the sampling distribution is within $(1 \pm \varepsilon)$ of the $D^2$ distribution, then the algorithm is called the *noisy-k-means++* algorithm Bhattacharya et al. (2020a); Grunau et al. (2023). We will use the following result in Grunau et al. (2023).

**Theorem 5** (Theorem 1.1 in Grunau et al. (2023)). *The noisy-k-means++ algorithm is $O(\log k)$ approximate, in expectation.*

Theorem 1, restated below, follows from the above theorem and Lemma 4. Note that the running time of the quantum implementation is $\sum_t \tilde{O}(\zeta^2 t) = \tilde{O}(\zeta^2 k^2)$.

**Theorem** (Restatement of Theorem 1). *There is a quantum implementation of $k$-means++ that runs in time $\tilde{O}(\zeta^2 k^2)$ and gives an $O(\log k)$ factor approximate solution for the $k$-means problem with a probability of at least $0.99$. Here, $\tilde{O}$ hides $\log^2(Nd)$ and $\log^2(kd)$ terms.*[9]

## C   QUANTUM INSPIRED $D^2$-SAMPLING

Ewin Tang's thesis Tang (2023) nicely categorizes quantum machine learning (QML) algorithms into ones where the quantum advantage (i.e., polylogarithmic running time) is solely because of the quantum access to the data and the ones where it is not (e.g., the HHL algorithm Harrow et al. (2009)). Quantum access means that a (normalized) data vector $\vec{v} := (\vec{v}(1), ..., \vec{v}(d))^T$ can be loaded onto the quantum workspace in $O(1)$ time as $|v\rangle = \vec{v}(1)|1\rangle + ... + \vec{v}(d)|d\rangle$. In this section, we will use the overhead arrow notation for vectors since we would want to distinguish between vectors and their approximations using different overhead symbols. One implication of having the quantum state $|v\rangle$ is that $i$ can be sampled with probability $\frac{\vec{v}(i)^2}{\sum_j \vec{v}(j)^2}$ by simply making a measurement in the standard basis. One of the key insights in Tang's Thesis Tang (2023) is that QML algorithms that benefit solely from the quantum data access can be 'dequantized' if we work with an appropriate classical counterpart of the quantum data access that allows *sampling access* of the kind described above (in addition to the classical access to the data). This is called *sample-query access*, or SQ access in short. Dequantization means that a classical algorithm can simulate the steps of the quantum algorithm using SQ access with similar running time dependence. The nice property of the SQ access data structures is that it can be constructed classically in linear time. Note that linear time means that we lose the quantum advantage. However, if we keep this aside and count the SQ access data structure construction as preprocessing, the remaining computation has a similar advantage as that of the quantum algorithm. This is a level playing field with the QML algorithm that allows fair comparison, specifically given that setting up quantum access (called QRAM) also takes linear time. Such classical algorithms obtained by dequantization are called *quantum inspired* algorithms. Our quantum algorithms based on $D^2$-sampling fall into the category of QML algorithms that can be dequantized in Tang's SQ access model. This section discusses the quantum-inspired classical algorithm we obtain by dequantizing our quantum algorithm using the SQ access data structure. Naturally, we start the discussion by looking at the definition of the SQ access data structure.

**Definition 3** (Query access, Definition 1.1 in Tang (2023)). *For a vector $\vec{v} \in \mathbb{C}^n$, we have $Q(\vec{v})$, query access to $\vec{v}$, if for all $i \in [n]$, we can query for $\vec{v}(i)$. The time (cost) of such a query is denoted by $\mathbf{q}(\vec{v})$.*

**Definition 4** (SQ-access to a vector, Definition 1.2 in Tang (2023)). *For a vector $\vec{v} \in \mathbb{C}^n$, we have $SQ(\vec{v})$, sampling and query access to $\vec{v}$, if we can:*

1. *query for entries in $\vec{v}$ as in $Q(\vec{v})$. The time cost is denoted by $\mathbf{q}(\vec{v})$.*

2. *obtain independent samples $i \in [n]$ where the probability of sampling $i$ is $\frac{\vec{v}(i)^2}{\|\vec{v}\|^2}$. This distribution is denoted by $\mathcal{D}_{\vec{v}}$. The time cost is denoted by $\mathbf{s}(\vec{v})$.*

---

[9]The output of $k$-means++ is a subset of data points, which can be stated as a subset of indices. If the description of centers is the expected output, the running time will also include a factor of $d$.

3. *query for $\|\vec{v}\|$. The time cost is denoted by $\mathbf{n}(\vec{v})$.*

Let $\mathbf{sq}(\vec{v}) \equiv \max\{\mathbf{q}(\vec{v}), \mathbf{s}(\vec{v}), \mathbf{n}(\vec{v})\}$ *denote the time cost of SQ-access.*

In quantum computation an $n$-dimensional state vector $|v_0\rangle$ evolves using unitary transformations $|v_1\rangle = U |v_0\rangle$, where $U$ is a $2^n \times 2^n$ unitary matrix. To be able to simulate such operations within the SQ-access model, we must make sure that SQ-access is available for intermediate states of the quantum operation. This means that we must (i) define an appropriate SQ-access notion for matrices and (ii) show closure properties for SQ-access. Fortunately, this has been shown in Tang's Thesis Tang (2023). A more flexible version of SQ-access is needed to show closure properties. This is called *oversampling SQ-access*. SQ-access for matrices and oversampling SQ-access is defined below.

**Definition 5** (SQ-access to a matrix). *For a matrix $A \in \mathbb{C}^{m \times n}$, we have $SQ(A)$ if we have $SQ(A(i,.))$ for all $i \in [m]$ (i.e., SQ-access to the row vectors), and $SQ(\vec{a})$, where $\vec{a}$ is the vector of row norms of $A$ (i.e., $\vec{a} = (\|A(1,.)\|, ..., \|A(m,.)\|)^T$). The complexity of operations on a matrix is defined as: $\mathbf{q}(A) := \max\{\mathbf{q}(A(i,.)), \mathbf{q}(\vec{a})\}$, $\mathbf{n}(A) := \mathbf{n}(\vec{a})$, $\mathbf{s}(A) := \max\{\mathbf{s}(A(i,.)), \mathbf{s}(\vec{a})\}$, and $\mathbf{sq}(A) := \max\{\mathbf{q}(A), \mathbf{n}(A), \mathbf{s}(A)\}$*

**Definition 6** (Oversampling and query access: Definition 4.4 in Tang (2023)). *For $\vec{v} \in \mathbb{C}^n$ and $\phi \geq 1$, we have $SQ_\phi(\vec{v})$, $\phi$-oversampling and query access to $\vec{v}$ ($\phi$-OSQ access, in short), if we have $Q(\vec{v})$ and $SQ(\tilde{v})$ for $\tilde{v} \in \mathbb{C}^n$ a vector satisfying (i) $\|\tilde{v}\|^2 = \phi\|\vec{v}\|^2$ and (ii) $|\tilde{v}(i)|^2 \geq |\vec{v}(i)|^2$ for all $i \in [n]$. Denote $\mathbf{s}_\phi(\vec{v}) := \mathbf{s}(\tilde{v})$, $\mathbf{q}_\phi(\vec{v}) := \mathbf{q}(\tilde{v})$, $\mathbf{n}_\phi(\vec{v}) := \mathbf{n}(\tilde{v})$, and $\mathbf{sq}_\phi(\vec{v}) := \max\{\mathbf{s}_\phi(\vec{v}), \mathbf{q}_\phi(\vec{v}), \mathbf{q}(\vec{v}), \mathbf{n}_\phi(\vec{v})\}$.*

Note that $\mathcal{D}_{\tilde{v}}$ for $\tilde{v}$ can be seen as an oversampling distribution of $\mathcal{D}_{\vec{v}}$ by a factor of $\phi$. This is because the sampling probability of index $i$ is given by

$$\mathcal{D}_{\tilde{v}}(i) = \frac{|\tilde{v}(i)|^2}{\|\tilde{v}\|^2} = \frac{|\tilde{v}(i)|^2}{\phi\|\vec{v}\|^2} \geq \frac{|\vec{v}(i)|^2}{\phi\|\vec{v}\|^2} = \frac{\mathcal{D}_{\vec{v}}(i)}{\phi}.$$

$\phi$-OSQ access also allows one to sample from $\mathcal{D}_{\vec{v}}$ and find $\|\vec{v}\|$ by paying a small time penalty (as a function of $\phi$ and allowed error). This is captured in the following lemma from Tang (2023).

**Lemma 5** (Lemma 4.5 in Tang (2023)). *Let $0 < \delta \leq 1$ and $0 < \varepsilon \leq 1$. Suppose we are given $SQ_\phi(\vec{v})$. Then we can sample from $\mathcal{D}_{\vec{v}}$ with proability at least $(1 - \delta)$ in time $O\left(\phi \cdot \mathbf{sq}_\phi(\vec{v}) \cdot \log\frac{1}{\delta}\right)$. We can also estimate $\|\vec{v}\|$ to within $(1 \pm \varepsilon)$ factor with probability at least $(1 - \delta)$ in $O\left(\frac{\phi}{\varepsilon^2} \cdot \mathbf{sq}_\phi(\vec{v}) \cdot \log\frac{1}{\delta}\right)$ time.*

We can also define oversampling query access to a matrix.

**Definition 7** (Oversampling and query access for a matrix). *For a matrix $A \in \mathbb{C}^{N \times d}$ and $\phi \geq 1$, we have $SQ_\phi(A)$, $\phi$-oversampling and query access to $A$ ($\phi$-OSQ access, in short), if we have $Q(A)$ and $SQ(\tilde{A})$ for $\tilde{A} \in \mathbb{C}^{N \times d}$ satisfying $\left\|\tilde{A}\right\|_F^2 = \phi\|A\|_F^2$ and $|\tilde{A}(i,j)|^2 \geq |A(i,j)|^2$ for all $(i,j) \in [N] \times [d]$. The $\phi$-oversampling complexity is given by $\mathbf{sq}_\phi(A) = \max\{\mathbf{q}(A), \mathbf{sq}(\tilde{A})\}$.*

There is a simple tree-based data structure that supports sample-query access for vectors and matrices. The details can be found in Tang (2023). Here, we give a summary of the running time of various operations.

**Lemma 6** (Remark 4.12 in Tang (2023)). *There is a data structure for storing a vector $\vec{v} \in \mathbb{R}^n$ with supporting the following operations:*

1. *Reading and updating an entry of $\vec{v}$ in $O(\log n)$ time. So, $\mathbf{q}(\vec{v}) = O(\log n)$.*

2. *Finding $\|\vec{v}\|^2$ in $O(1)$ time. So, $\mathbf{n}(\vec{v}) = O(1)$.*

3. *Sampling from $\mathcal{D}_{\vec{v}}$ in $O(\log n)$ time. So, $\mathbf{s}(\vec{v}) = O(\log n)$*

*It follows from the above that $\mathbf{sq}(\vec{v}) = O(\log n)$. Moreover, the time to construct the data structure is $O(n)$.*

A similar lemma holds for storing matrices.

**Lemma 7** (Remark 4.12 in Tang (2023))**.** *For any matrix $A \in \mathbb{R}^{m \times n}$, let $\vec{A}$ denote the vector of row norms of A, i.e., the $i^{th}$ entry of $\vec{A}$, denoted by $\vec{A}(i)$, is $\|A(i,.)\|$. There is a data structure for storing a matrix $A \in \mathbb{R}^{m \times n}$ with supporting the following operations:*

1. *Reading and updating an entry of A in $O(\log mn)$ time.*

2. *Finding $\vec{A}(i)$ in $O(\log m)$ time.*

3. *Finding $\|A\|_F^2$ in $O(1)$ time.*

4. *Sampling from $\mathcal{D}_{\vec{A}}$ and $\mathcal{D}_{A(i,.)}$ in $O(\log mn)$ time.*

*The time to construct the data structure is $O(mn)$.*

Let us now see how $D^2$-sampling can be done using the above SQ-access data structures. Let $V \in \mathbb{R}^{N \times d}$ be the data matrix with the row vectors $\vec{v}_1, ..., \vec{v}_N$ as data points. Let $C \in \mathbb{R}^{m \times d}$ be the center set matrix with the centers being the $i$ row vectors $\vec{c}_1, ..., \vec{c}_m$. Consider $D^2$-sampling a point from $V$ with respect to the center set $C$. The probability of sampling the $i^{th}$ point is given by:

$$\mathbf{Pr}[i] = \frac{\min_{t \in [m]} \|\vec{v}_i - \vec{c}_t\|^2}{\sum_{s \in [N]} \min_{t \in [m]} \|\vec{v}_s - \vec{c}_t\|^2}$$

Consider vectors $\vec{w}_1, ..., \vec{w}_m \in \mathbb{R}^N$ defined as:

$$\vec{w}_j := (\|\vec{v}_1 - \vec{c}_j\|, \|\vec{v}_2 - \vec{c}_j\|, ..., \|\vec{v}_N - \vec{c}_j\|)^T.$$

The first step towards $D^2$-sampling using the SQ-access data structure for matrices $V$ and $C$ is to check whether SQ-access for vectors $\vec{w}_1, ..., \vec{w}_m$ can be made available. In the next lemma, we show that $\phi$-oversampling SQ-access for an appropriately chosen value of $\phi$ is possible.

**Lemma 8.** *Let $\alpha$-oversampling SQ access to the data matrix $V \in \mathbb{R}^{N \times d}$ be available as $SQ_\alpha(V)$ and let $\beta$ oversampling SQ access be available for a vector $\vec{c} \in \mathbb{R}^d$ as $SQ_\beta(\vec{c})$. The we have $\gamma$ oversampling SQ access, $SQ_\gamma(\vec{w})$, for vector $\vec{w} := (\|V(1,.) - \vec{c}\|, ..., \|V(N,.) - \vec{c}\|)^T$ with $\gamma = \frac{2}{\|\vec{w}\|^2} \cdot \left(\alpha \|V\|^2 + N\beta \|\vec{c}\|^2\right)$. Moreover, the complexity is related as $\mathbf{sq}_\gamma(\vec{w}) \leq d \cdot (\mathbf{sq}_\alpha(V) + \mathbf{sq}_\beta(\vec{c}))$.*

*Proof.* $SQ_\alpha(V)$ means that $Q(V)$ and $SQ(\tilde{V})$ are available for a matrix $\tilde{V}$ such that (i) $\left\|\tilde{V}\right\|^2 = \alpha \|V\|^2$ and (ii) $|\tilde{V}(i,j)| \geq |V(i,j)|$ for all $(i,j) \in [N] \times [d]$. Similarly, $SQ_\beta(\vec{c})$ means that $Q(\vec{c})$ and $SQ(\tilde{c})$ are available for a vector $\tilde{c}$ such that (i) $\|\tilde{c}\|^2 = \beta \|\vec{c}\|^2$ and (ii) $|\tilde{c}(t)| \geq |\vec{c}(t)|$ for all $t \in [d]$. To show $SQ_\gamma(\vec{w})$, we first need to show $Q(\vec{w})$. This is simple since the $j^{th}$ coordinate of $\vec{w}$ can be found using all the coordinates of $V(j,.)$ and $\vec{c}$ which are available since we have $Q(V)$ and $Q(\vec{c})$. So, $\mathbf{q}(\vec{w}) = d \cdot (\mathbf{q}(V) + \mathbf{q}(\vec{c}))$. We now need to show $SQ(\tilde{w})$ for an appropriately chosen $\tilde{w}$. We consider:

$$\tilde{w}(j) = \sqrt{2\left(\left\|\tilde{V}(j,.)\right\|^2 + \|\tilde{c}\|^2\right)}.$$

Before we see why SQ access to $\tilde{w}$ is possible, let us first observe that $SQ(\tilde{w})$ would give a $\gamma$-oversampling SQ access to $\vec{w}$. Towards this, we note that

$$\|\tilde{w}\|^2 = \sum_j |\tilde{w}(j)|^2 = 2\sum_j \left(\left\|\tilde{V}(j,.)\right\|^2 + \|\tilde{c}\|^2\right) = \frac{2}{\|\vec{w}\|^2}\left(\alpha\|V\|^2 + N\beta\|\vec{c}\|^2\right) \cdot \|\vec{w}\|^2 = \gamma\|\vec{w}\|^2.$$

Moreover, we have:

$$|\vec{w}(j)| = \|V(j,.) - \vec{c}\| \leq \|V(j,.)\| + \|\vec{c}\| \leq \left\|\tilde{V}(j,.)\right\| + \|\tilde{c}\| \leq \sqrt{2\left(\left\|\tilde{V}(j,.)\right\|^2 + \|\tilde{c}\|^2\right)} = |\tilde{w}(j)|.$$

Finally, we must show $SQ(\tilde{w})$. Coordinates of $\tilde{w}$ can be found using $\left\|\tilde{V}(j,.)\right\|$'s and $\|\tilde{c}\|$ which are available through oversampling access to $V$ and $\vec{c}$. $\|\tilde{w}\|^2$ is available through $\left\|\tilde{V}\right\|^2$ and $\|\tilde{c}\|^2$.

Finally, $\mathcal{D}_{\tilde{w}}$ can be sampled by first selecting $\tilde{V}$ with probability $\frac{\|\tilde{V}\|^2}{\|\tilde{V}\|^2 + N\|\tilde{c}\|^2}$ or $\tilde{c}$ with probability $\frac{N\|\tilde{c}\|^2}{\|\tilde{V}\|^2 + N\|\tilde{c}\|^2}$ and then sampling from $\mathcal{D}_{\tilde{v}}$ or $\mathcal{U}_{[N]}$ (uniform distribution over $[N]$), respectively. Given this, the probability of sampling $i$ works out to be:

$$\mathbf{Pr}[i] = \frac{\left\|\tilde{V}\right\|^2}{\left\|\tilde{V}\right\|^2 + N\|\tilde{c}\|^2} \cdot \frac{\left\|\tilde{V}(i,.)\right\|^2}{\left\|\tilde{V}\right\|^2} + \frac{N\|\tilde{c}\|^2}{\left\|\tilde{V}\right\|^2 + N\|\tilde{c}\|^2} \cdot \frac{1}{N} = \frac{\tilde{w}(i)^2}{\|\tilde{w}\|^2} = \mathcal{D}_{\tilde{w}}(i).$$

From the above discussion, it follows that $\mathbf{sq}_\gamma(\vec{w}) \leq d \cdot (\mathbf{sq}_\alpha(V) + \mathbf{sq}_\beta(\vec{c}))$. $\qquad \square$

The previous lemma gives a way to obtain oversampling query access to distances from a single center. In $D^2$-sampling, we have a set $C$ of $m$ centers, and we sample a point based on the squared distance from the nearest center. To enable this operation in the SQ framework, we need sample and query access to a vector where the $i^{th}$ coordinate is the distance of $i^{th}$ data point to the nearest center in $C$. In the following lemma, we show how to do this. The proof closely follows Lemma 4.6 of Tang (2023).

**Lemma 9.** *Let* $\vec{u}_1, ..., \vec{u}_m \in (\mathbb{R}^{\geq 0})^N$ *and* $\phi_1, ..., \phi_m \in \mathbb{R}^{\geq 1}$. *Let*

$$\vec{w} := \left(\min_{i \in [m]} \vec{u}_i(1), \min_{i \in [m]} \vec{u}_i(2), ..., \min_{i \in [m]} \vec{u}_i(N)\right)^T.$$

*Suppose we have* $SQ_{\phi_i}(\vec{u}_i)$ *for every* $i \in [m]$. *Then there is a* $\phi$-*oversampling query access,* $SQ_\phi(\vec{u})$ *for* $\phi = \frac{\sum_{i \in [m]} \phi_i \|\vec{u}_i\|^2}{m\|\vec{w}\|^2}$. *The complexity is give by:* $\mathbf{q}(\vec{w}) = \sum_{i \in [m]} \mathbf{q}(\vec{u}_i)$, $\mathbf{q}_\phi(\vec{w}) = \sum_{i \in [m]} \mathbf{q}_{\phi_i}(\vec{u}_i)$, $\mathbf{n}_\phi(\vec{w}) = \sum_{i \in [m]} \mathbf{n}_{\phi_i}(\vec{u}_i)$. *The sampling complexity is* $\mathbf{s}_\phi(\vec{w}) = \max_{i \in [m]} \mathbf{s}_{\phi_i}(\vec{u}_i)$, *after a single-time pre-processing cost of* $\sum_{i \in [m]} \mathbf{n}_{\phi_i}(\vec{u}_i)$.

*Proof.* We need to show $Q(\vec{w})$ and $SQ(\tilde{w})$ for an appropriate $\tilde{w}$. We can respond to each coordinate of $\vec{w}$ by querying $\vec{u}_i$'s and taking the minimum. This gives $\mathbf{q}(\vec{w}) = \sum_{i \in [m]} \mathbf{q}(\vec{u}_i)$. Let $\tilde{u}_1, ..., \tilde{u}_m$ be the vectors corresponding to the oversampling access $SQ_{\phi_1}(\vec{u}_1), ..., SQ_{\phi_m}(\vec{u}_m)$. For oversampling query access, consider:

$$\tilde{w}(i) = \sqrt{\frac{1}{m} \cdot \sum_{j \in [m]} |\tilde{u}_j(i)|^2}.$$

Each coordinate of $\tilde{w}$ can be found by querying $\tilde{u}_i$'s and computing the above expression. So, we have $Q(\tilde{w})$ and $\mathbf{q}_\phi(\vec{w}) = \sum_{i \in [m]} \mathbf{q}_{\phi_i}(\vec{u}_i)$. The norm of $\tilde{w}$ is given by:

$$\|\tilde{w}\|^2 = \frac{1}{m} \sum_{i \in [N]} \sum_{j \in [m]} |\tilde{u}_j(i)|^2 = \frac{1}{m} \sum_{j \in [m]} \|\tilde{u}_j\|^2 = \frac{1}{m} \sum_{j \in [m]} \phi_j \|\vec{u}_j\|^2 = \left(\frac{\sum_{j \in [m]} \phi_j \|\vec{u}_j\|^2}{m\|\vec{w}\|^2}\right) \|\vec{w}\|^2 = \phi\|\vec{w}\|^2.$$

So, we can query the norm of $\tilde{w}$ by querying the norms of $\tilde{u}_j$'s. Hence, $\mathbf{n}_\phi(\vec{w}) = \sum_{j \in [m]} \mathbf{n}_{\phi_j}(\vec{u}_j)$. Now, we need to show that $\tilde{w}(i)$ upper bounds $\vec{w}(i)$. We have:

$$\tilde{w}(i) = \sqrt{\frac{1}{m} \sum_{j \in [m]} |\tilde{u}_j(i)|^2} \geq \sqrt{\frac{1}{m} \sum_{j \in [m]} \min_{j' \in [m]} |\tilde{u}_{j'}(i)|^2} = \min_{j' \in [m]} \tilde{u}_{j'}(i) \geq \min_{j' \in [m]} \vec{u}_j(i) = \vec{w}(i).$$

Now we describe how to sample from $\mathcal{D}_{\tilde{w}}$. We first query the norms of $\tilde{u}_j$. We pick a $j \in [m]$ with probability $\frac{\|\tilde{u}_j\|^2}{\sum_{j' \in [m]} \|\tilde{u}_{j'}\|^2}$, and then sample an index $i \in [N]$ from the distribution $\mathcal{D}_{\tilde{u}_j}$. The probability of sampling an index $i$ is given by:

$$\mathbf{Pr}[i] = \sum_{j \in [m]} \frac{\|\tilde{u}_j\|^2}{\sum_{j' \in [m]} \|\tilde{u}_{j'}\|^2} \cdot \frac{|\tilde{u}_j(i)|^2}{\|\tilde{u}_j\|^2} = \frac{\frac{1}{m} \sum_{j \in [m]} |\tilde{u}_j(i)|^2}{\frac{1}{m} \sum_{j' \in [m]} \|\tilde{u}_{j'}\|^2} = \frac{|\tilde{w}(i)|^2}{\|\tilde{w}\|^2}.$$

This is the correct probability for sampling from $\mathcal{D}_{\tilde{w}}$. Note that from the description, the sampling complexity is $\mathbf{s}_\phi(\vec{w}) = \max_{i \in [m]} \mathbf{s}_{\phi_i}(\vec{u}_i)$, after a single-time pre-processing cost of $\sum_{i \in [m]} \mathbf{n}_{\phi_i}(\vec{u}_i)$.
$\qquad \square$

We now have all the basic tools to perform $D^2$-sampling in the SQ access model and evaluate its time complexity. In the remaining discussion, we will assume that the origin is centered at one of the data points so that the maximum norm of a data vector is upper bounded by the maximum interpoint distance.

**Theorem 6.** *Let $0 < \delta \leq 1$. Let $V \in \mathbb{R}^{N \times d}$ be the dataset and let $\vec{c}_1, ..., \vec{c}_m \in \mathbb{R}^d$ be $m$ arbitrary centers from the dataset and let $SQ(V)$ and $SQ(\vec{c}_1), ..., SQ(\vec{c}_m)$ be available. Let $\zeta$ denote the aspect ratio of the dataset (i.e., the ratio of the maximum to minimum interpoint distance). Then there is a classical algorithm that with probability at least $(1 - \delta)$ outputs a sample from the $D^2$-distribution (i.e., $D^2$-sample) of $V$ with respect to centers $\{\vec{c}_1, ..., \vec{c}_m\}$ which runs in time $O(\zeta^2 md(\log \frac{1}{\delta}) \log Nd)$.*

*Proof.* Let $d_{norm}$ be the maximum norm of a data point. Let $d_{min}$ and $d_{max}$ be the minimum and maximum distance between two data points, respectively. Then, we have $\zeta = \frac{d_{max}}{d_{min}} \geq \frac{d_{norm}}{d_{min}}$. Let us define vectors $\vec{u}_1, ..., \vec{u}_m$ where $\vec{u}_j(i) := \|V(i,.) - \vec{c}_j\|$. Using Lemma 8, we can enable $SQ_{\phi_j}(\vec{u}_j)$ for every $j \in [m]$, for $\phi_j = \frac{2}{\|\vec{u}_j\|^2}(\|V\|^2 + N\|\vec{c}_j\|^2)$. Lemma 9 gives a way to define $SQ_\phi(\vec{w})$, where $\vec{w}$ is defined as $\vec{w}(i) := \min_{j \in [m]} |\vec{u}_j(i)|$ and $\phi = \frac{\sum_{j \in [m]} \phi_j \|\vec{u}_j\|^2}{m \|\vec{w}\|^2}$. Note that for the $\vec{w}$ defined, $\mathcal{D}_{\vec{w}}$ is precisely the $D^2$ distribution over $V$ with respect to centers $\vec{c}_1, ..., \vec{c}_m$ and from Lemma 5, we know how to use $SQ_\phi(\vec{w})$ to sample from $\mathcal{D}_{\vec{w}}$ with probability at least $(1 - \delta)$ in time $O(\phi \cdot \mathbf{sq}_\phi(\vec{w}) \cdot \log \frac{1}{\delta})$. For calculating the complexity, we first get a bound on $\phi$:

$$\phi = \frac{\sum_{j \in [m]} \phi_j \|\vec{u}_j\|^2}{m \|\vec{w}\|^2} = \frac{\sum_{j \in [m]} \frac{2}{\|\vec{u}_j\|^2}(\|V\|^2 + N\|\vec{c}_j\|^2)\|\vec{u}_j\|^2}{m \|\vec{w}\|^2} = \frac{2\|V\|^2}{\|\vec{w}\|^2} + \frac{2N}{\|\vec{w}\|^2} \frac{1}{m} \sum_{j \in [m]} \|\vec{c}_j\|^2.$$

The $i^{th}$ coordinate of $\vec{w}$ is the least distance of $V(i,.)$ from a center. This gives us $\|\vec{w}\|^2 \geq (N - m) \cdot d_{min}^2$. Also, $\|V\|^2 \leq Nd_{norm}^2$ and $\|\vec{c}_j\|^2 \leq d_{norm}^2$. Putting these bounds in the above equation (and that $m \leq N/2$), we get $\phi \leq 8\frac{d_{norm}^2}{d_{min}^2} \leq 8\zeta^2$. From Lemmas 6 and 7, we have $\mathbf{sq}(V) = O(\log Nd)$, $\mathbf{sq}(\vec{c}_j) = O(\log d)$. From Lemma 8, we have $\mathbf{sq}_{\phi_j}(\vec{u}_j) = d \cdot (\mathbf{sq}(V) + \mathbf{sq}(\vec{c}_j)) = O(d \log Nd)$ for every $j \in [m]$. From Lemma 9, we have $\mathbf{sq}_\phi(\vec{w}) = \sum_{j \in [m]} \mathbf{sq}_{\phi_j}(\vec{u}_j) = O(md \log Nd)$. So, from Lemma 5, the complexity of $D^2$-sampling a point is $O(\phi \cdot \mathbf{sq}_\phi(\vec{w}) \cdot \log \frac{1}{\delta}) = O(\zeta^2 md(\log \frac{1}{\delta}) \log Nd)$. $\qquad \square$

The $k$-means++ algorithm is simply $D^2$-sampling $k$ centers iteratively while updating the list of centers. We summarise the discussion of this section by giving the complexity of the $k$-means++ algorithm and the proof of Theorem 2. We restate the theorem for ease of reading.

**Theorem** (Restatement of Theorem 2). *There is a classical implementation of $k$-means++ (which we call QI-$k$-means++) that runs in time $O(Nd) + \tilde{O}(\zeta^2 k^2 d)$. Moreover, with probability at least $0.99$, QI-$k$-means++ outputs $k$ centers that give $O(\log k)$-approximation in expectation for the $k$-means problem.*

*Proof.* The $Nd$ term in the running time expression is for setting up $SQ(V)$, the sample-query access to the data matrix $V$. This may also be regarded as the preprocessing step. This is followed by a $D^2$-sampling $k$ centers iteratively while updating the set of centers by including the new centers sampled. From the previous theorem, we know that when we have $m$ centers, the cost of obtaining a $D^2$-sample is $O(\zeta^2 md(\log \frac{1}{\delta}) \log Nd)$ with probability at least $(1 - \delta)$. To obtain an overall probability of success of $0.99$, we need to have a success probability of at least $1 - O(1/k)$ in each iteration. This makes the overall $D^2$-sampling cost $O(\zeta^2 k^2 d \log k \log Nd)$. So, the overall time complexity of QI-$k$-means++ is $O(Nd) + \tilde{O}(\zeta^2 k^2 d)$. Since with probability at least $0.99$, we obtain $k$ perfectly $D^2$-sampled center, the approximation guarantee remains the same as that of $k$-means++, which is $O(\log k)$.[10] $\qquad \square$

---

[10]What happens with the remaining $0.01$ probability? $D^2$-sampling in the SQ model succeeding with probability at least $(1 - \delta)$ means that with probability at least $(1 - \delta)$ a center is sampled with $D^2$ distribution, and with the remaining probability, no point is sampled.

## C.1    QUANTUM INSPIRED APPROXIMATE $D^2$-SAMPLING

Sampling from a distribution that is $\varepsilon$-close to the $D^2$ distribution is sufficient to obtain $O(\log k)$ approximation. This follows from the analysis of the noisy-$k$-means++ algorithm Grunau et al. (2023). This relaxation (without any approximation penalty beyond constant factors) allows us to shave off the factor of $d$ from the time complexity of the quantum-inspired algorithm. However, as we will see in the analysis of this section, it is at the cost of worsening the dependence on the aspect ratio. So, the algorithm in this section should be used for high-dimensional cases where the aspect ratio is small. The factor of $d$ in the sampling component of QI-$k$-means++ appears because for enabling the oversampling and query access for the vector $\vec{w} := (\|V(1,.) - \vec{c}\|, ..., \|V(N,.) - \vec{c}\|)^T$ ($\vec{c}$ is a center), we need to access the $d$ coordinates of $V(i,.)$ and $\vec{c}$ to be able to answer the query access to the $i^{th}$ index of $\vec{w}$, which is $\|V(i,.) - \vec{c}\|$. Note that $\|V(i,.) - \vec{c}\| = \sqrt{\|V(i,.)\|^2 + \|\vec{c}\|^2 - 2\langle V(i,.), \vec{c}\rangle}$, where $\langle V(i,.), \vec{c}\rangle$ denotes the dot product. The terms $\|V(i,.)\|^2$ and $\|\vec{c}\|^2$ are available in $O(1)$ time. The dot product term is costly and adds a factor of $d$. So, the key idea in eliminating the factor of $d$ is to replace the dot product term with an appropriate approximation that requires a much smaller access time. This uses the standard sampling trick - randomly sample a subset of indices, compute the dot product restricted to these indices of the subset, and scale (see Lemma 5.17 in Tang (2023)). The number of indices we need to sample to ensure that the resulting approximation is within $(1 \pm \varepsilon)$ of $\|V(i,.) - \vec{c}\|$ for every $i$ with probability at least $(1 - \delta)$ is $\tilde{O}\left( \frac{\log(1/\delta)}{\varepsilon^2} \cdot \frac{\|\vec{c}\|^2 \cdot \|V(i,.)\|^2}{\|V(i,.) - \vec{c}\|^2} \right) = \tilde{O}\left( \frac{\zeta^4 \log 1/\delta}{\varepsilon^2} \right)$. The remaining steps are the same as those in the exact version, except that $d$ is replaced with $\tilde{O}\left( \frac{\zeta^4 \log 1/\delta}{\varepsilon^2} \right)$. This results in the overall running time $O(Nd) + \tilde{O}\left( \zeta^6 k^2 \right)$.

## D    A QUANTUM APPROXIMATION SCHEME (PROOF OF THEOREM 4)

We start the discussion with the $D^2$-sampling method. In particular, we would like to check the robustness of the approximation guarantee provided by the $D^2$-sampling method against errors in estimating the distances between points. We will show that the $D^2$-sampling method gives a constant pseudo-approximation even under sampling errors.

### D.1    PSEUDOAPPROXIMATION USING $D^2$-SAMPLING

Let the matrix $V \in \mathbb{R}^{N \times d}$ denote the dataset, where row $i$ contains the $i^{th}$ data point $v_i \in \mathbb{R}^d$. Let the matrix $C \in \mathbb{R}^{t \times d}$ denote any $t$-center set, where row $i$ contains the $i^{th}$ center $c_i \in \mathbb{R}^d$ out of the $t$ centers. Sampling a data point using the $D^2$ distribution w.r.t. (*short for with respect to*) a center set $C$ means that the datapoint $v_i$ gets sampled with probability proportional to the squared distance to its nearest center in the center set $C$. This is also known as $D^2$ sampling w.r.t. center set $C$. More formally, data points are sampled using the distribution $\left( \frac{D^2(v_1,C)}{\sum_j D^2(v_j,C)}, ..., \frac{D^2(v_N,C)}{\sum_j D^2(v_j,C)} \right)$, where $D^2(v_j, C) \equiv \min_{c \in C} D^2(v_j, c)$. For the special case $C = \emptyset$, $D^2$ sampling is the same as uniform sampling. The $k$-means++ seeding algorithm starts with an empty center set $C$ and, over $k$ iterations, adds a center to $C$ in every iteration by $D^2$ sampling w.r.t. the current center set $C$. It is known from the result of Arthur & Vassilvitskii (2007) that this $k$-means++ algorithm above gives an $O(\log k)$ approximation in expectation. It is also known from the result of Aggarwal et al. (2009) that if $2k$ centers are sampled, instead of $k$ (*i.e., the for-loop runs from* 1 *to* 2k), the cost with respect to these $2k$ centers is at most some constant times the optimal $k$-means cost. Such an algorithm is called a *pseudo approximation* algorithm. Such a pseudo approximation algorithm is sufficient for the approximation scheme of Bhattacharya et al. (2020b). So, we will quantize the following constant factor pseudo-approximation algorithm.

In the quantum simulation of the above sampling procedure, there will be small errors in the sampling probabilities in each iteration. We need to ensure that the constant approximation guarantee of the above procedure is robust against small errors in the sampling probabilities owing to errors in distance estimation. We will work with a relative error of $(1 \pm \delta)$ for small $\delta$. Following is a crucial lemma from Arthur & Vassilvitskii (2007) that we will need to show the pseudo approximation property of Algorithm 1.

---

**Algorithm 3** A pseudo-approximation algorithm based on $D^2$-sampling.

---

   **Input:** $(V, k)$
   $C \leftarrow \{\}$
   **for** $i = 1$ to $2k$ **do**
      Pick $c$ using $D^2$-sampling w.r.t. center set $C$
      $C := C \leftarrow \{c\}$
   **end for**
   **return** $C$

---

**Lemma 10** (Lemma 3.2 in Arthur & Vassilvitskii (2007)). *Let $A$ be an arbitrary optimal cluster, and let $C$ be an arbitrary set of centers. Let $c$ be a center chosen from $A$ with $D^2$-sampling with respect to $C$. Then $\mathbb{E}[cost(A, C \cup \{c\})] \leq 8 \cdot OPT(A)$.*

The above lemma is used as a black box in the analysis of Algorithm 1 in Aggarwal et al. (2009). The following version of the lemma holds for distance estimates with a relative error of $(1 \pm \delta)$ and gives a constant factor approximation guarantee. Since Lemma 10 is used as a black box in the analysis of Algorithm 1, replacing this lemma with Lemma 11 also gives a constant factor approximation to the $k$-means objective. We will use the following notion of the closeness of two distance functions.

**Definition 8.** *A distance function $D_1$ is said to be $\delta$-close to distance function $D_2$, denoted by $D_1 \sim_\delta D_2$, if for every pair of points $x, y \in \mathbb{R}^d$, $D_1(x, y) \in (1 \pm \delta) \cdot D_2(x, y)$.*[11]

**Lemma 11.** *Let $0 < \delta \leq 1/2$. Let $A$ be an arbitrary optimal cluster and $C$ be an arbitrary set of centers. Let $c$ be a center chosen from $A$ with $\mathcal{D}^2$-sampling with respect to $C$, where $\mathcal{D} \sim_\delta D$. Then $\mathbb{E}[cost(A, C \cup \{c\})] \leq 72 \cdot OPT(A)$.*

*Proof.* Let $D(a)$ denote the distance of the point $a$ from the nearest center in $C$ and let $\mathcal{D}(a)$ denote the estimated distance. We have $\mathcal{D}(a) \in D(a) \cdot (1 \pm \delta)$. The following expression gives the expectation:

$$\sum_{a_0 \in A} \frac{\mathcal{D}^2(a_0)}{\sum_{a \in A} \mathcal{D}^2(a)} \cdot \sum_{a' \in A} \min\left(D^2(a'), D^2(a', a_0)\right)$$

Note that for all $a_0, a \in A$, $D(a_0) \leq D(a) + D(a, a_0)$. This gives $\mathcal{D}(a_0) \leq \frac{1+\delta}{1-\delta} \cdot \mathcal{D}(a) + (1+\delta) \cdot D(a_0, a)$, which further gives $\mathcal{D}^2(a_0) \leq 2\left(\frac{1+\delta}{1-\delta}\right)^2 \cdot \mathcal{D}^2(a) + 2(1+\delta)^2 \cdot D^2(a_0, a)$ and $\mathcal{D}^2(a_0) \leq \frac{2}{|A|}\left(\frac{1+\delta}{1-\delta}\right)^2 \cdot \sum_{a \in A} \mathcal{D}^2(a) + \frac{2}{|A|}(1+\delta)^2 \cdot \sum_{a \in A} D^2(a_0, a)$. We use this to obtain the following upper bound on the expectation $\mathbb{E}[cost(A, C \cup \{c\})]$:

$$\sum_{a_0 \in A} \frac{\mathcal{D}^2(a_0)}{\sum_{a \in A} \mathcal{D}^2(a)} \cdot \sum_{a' \in A} \min\left(D^2(a'), D^2(a', a_0)\right)$$

$$\leq \sum_{a_0 \in A} \frac{\left(\frac{2}{|A|}\left(\frac{1+\delta}{1-\delta}\right)^2 \sum_{a \in A} \mathcal{D}^2(a)\right)}{\sum_{a \in A} \mathcal{D}^2(a)} \cdot \sum_{a' \in A} \min\left(D^2(a'), D^2(a', a_0)\right) +$$

$$\sum_{a_0 \in A} \frac{\left(\frac{2}{|A|}(1+\delta)^2 \sum_{a \in A} D^2(a_0, a)\right)}{\sum_{a \in A} \mathcal{D}^2(a)} \cdot \sum_{a' \in A} \min\left(D^2(a'), D^2(a', a_0)\right)$$

$$\leq \sum_{a_0 \in A} \sum_{a' \in A} \frac{2}{|A|}\left(\frac{1+\delta}{1-\delta}\right)^2 D^2(a', a_0) + \sum_{a_0 \in A} \sum_{a \in A} \frac{2}{|A|}\left(\frac{1+\delta}{1-\delta}\right)^2 D(a_0, a)^2$$

$$= \frac{4}{|A|}\left(\frac{1+\delta}{1-\delta}\right)^2 \sum_{a_0 \in A} \sum_{a \in A} D^2(a_0, a)$$

---

[11]We use the notation that for positive reals $P, Q$, $P \in (1 \pm \delta) \cdot Q$ if $(1 - \delta) \cdot Q \leq P \leq (1 + \delta) \cdot Q$.

$$
\begin{aligned}
&= \quad 8\left(\frac{1+\delta}{1-\delta}\right)^2 OPT(A) \\
&\leq \quad 72 \cdot OPT(A).
\end{aligned}
$$

This completes the proof of the lemma. $\qquad\square$

We will use this lemma in the approximation scheme of Bhattacharya et al. (2020b). However, this lemma may be of independent interest as this gives a quantum pseudo approximation algorithm with a constant factor approximation that runs in time that is polylogarithmic in the data size and linear in $k$ and $d$. We will discuss this quantum algorithm in the next section.

### D.2 APPROXIMATION SCHEME OF BHATTACHARYA ET AL. (2020B)

A high-level description of the approximation scheme of Bhattacharya et al. (2020b) was given in the introduction. A more detailed pseudocode is given in Algorithm 4. In addition to the input

---

**Algorithm 4** Algorithm of Bhattacharya et al. (2020b). For our Quantum application, we will replaces $D$ with $\delta$-close $\mathcal{D}$.

1: **Input**: $(V, k, \varepsilon, C_{init})$, where $V$ is the dataset, $k > 0$ is the number of clusters, $\varepsilon > 0$ is the error parameter, and $C_{init}$ is a $k$ center set that gives constant (pseudo)approximation.
2: **Output**: A list $\mathcal{L}$ of $k$ center sets such that for at least one $C \in \mathcal{L}$, $\Phi(V, C) \leq (1 + \varepsilon) \cdot OPT$.
3: **Constants**: $\rho = O(\frac{k}{\varepsilon^4})$; $\tau = O(\frac{1}{\varepsilon})$
4: $\mathcal{L} \leftarrow \emptyset$; $count \leftarrow 1$
5: **repeat**
6: $\quad$ Sample a multi-set $M$ of $\rho k$ points from $V$ using $D^2$-sampling wrt center set $C_{init}$
7: $\quad M \leftarrow M \cup \{\tau k \text{ copies of each element in } C_{init}\}$
8: $\quad$ **for all** disjoint subsets $S_1, ..., S_k$ of $M$ such that $\forall i, |S_i| = \tau$ **do**
9: $\qquad \mathcal{L} \leftarrow \mathcal{L} \cup (\mu(S_1), ..., \mu(S_k))$
10: $\quad$ **end for**
11: $\quad count$++
12: **until** $count < 2^k$
13: **return** $\mathcal{L}$

---

instance $(V, k)$ and error parameter $\varepsilon$, the algorithm is also given a constant approximate solution $C_{init}$, which is used for $D^2$-sampling. A pseudoapproximate solution $C_{init}$ is sufficient for the analysis in Bhattacharya et al. (2020b). The discussion from the previous subsection gives a robust algorithm that outputs a pseudoapproximate solution even under errors in distance estimates. So, the input requirement of Algorithm 4 can be met. Now, the main ingredient being $D^2$-sampling, we need to ensure that errors in distance estimate do not seriously impact the approximation analysis of Algorithm 4. We state the main theorem of Bhattacharya et al. (2020b) before giving the analogous statement for the modified algorithm where $D$ is replaced with $\mathcal{D}$ that is $\delta$-close to $D$.

**Theorem 7** (Theorem 1 in Bhattacharya et al. (2020b)). *Let $0 < \varepsilon \leq 1/2$ be the error parameter, $V \in \mathbb{R}^{N \times d}$ be the dataset, $k$ be a positive integer, and let $C_{init}$ be a constant (pseudo)approximate solution for dataset $V$. Let $\mathcal{L}$ be the list returned by Algorithm 4 on input $(V, k, \varepsilon, C_{init})$ using the Euclidean distance function $D$. Then with probability at least $3/4$, $\mathcal{L}$ contains a center set $C$ such that $\Phi(V, C) \leq (1 + \varepsilon) \cdot OPT$. Moreover, $|\mathcal{L}| = \tilde{O}\left(2^{\tilde{O}(\frac{k}{\varepsilon})}\right)$ and the running time of the algorithm is $O(Nd|\mathcal{L}|)$.*

We give the analogous theorem with access to the Euclidean distance function $D$ replaced with a function $\mathcal{D}$ that is $\delta$-close to $D$.

**Theorem 8.** *Let $0 < \varepsilon \leq \frac{1}{2}$ be the error parameter, $0 < \delta < 1/2$ be the closeness parameter, $V \in \mathbb{R}^{N \times d}$ be the dataset, $k$ be a positive integer, and let $C_{init}$ be a constant (pseudo)approximate solution for dataset $V$. Let $\mathcal{L}$ be the list returned by Algorithm 4 on input $(V, k, \varepsilon, C_{init})$ using the distance function $\mathcal{D}$ that is $\delta$-close to the Euclidean distance function $D$. Then with probability at least $3/4$, $\mathcal{L}$ contains a center set $C$ such that $\Phi(V, C) \leq (1 + \varepsilon) \cdot OPT$. Moreover, $|\mathcal{L}| = 2^{\tilde{O}(\frac{k}{\varepsilon})}$ and the running time of the algorithm is $O(Nd|\mathcal{L}|)$.*

The proof of the above theorem closely follows the proof of Theorem 7 of Bhattacharya et al. (2020b). This is similar to the proof of Theorem 11 that we saw earlier, closely following the proof of Lemma 10. The minor changes are related to approximate distance estimates using $\mathcal{D}$ instead of real estimates using $D$. The statement of Theorem 8 is not surprising in this light. Instead of repeating the entire proof of Bhattacharya et al. (2020b), we point out the one change in their argument caused by using $\mathcal{D}$ instead of $D$ as the distance function. The analysis of Bhattacharya et al. (2020b) works by partitioning the points in any optimal cluster $X_j$ into those that are close to $C_{init}$ and those that are far. For the far points, it is shown that when doing $D^2$-sampling, a far point will be sampled with probability at least $\gamma$ times the uniform sampling probability (see Lemma 21 in Goyal et al. (2020), which is a full version of Bhattacharya et al. (2020b)). It then argues that a reasonable size set of $D^2$-sampled points will contain a uniform sub-sample. A combination of the uniform sub-sample along with copies of points in $C_{init}$ gives a good center for this optimal cluster $X_j$. Replacing $D$ with $\mathcal{D}$ decrease the value of $\gamma$ by a multiplicative factor of $\frac{(1-\delta)^2}{(1+\delta)^2} \geq (1-\delta)^4$. This means that the number of points sampled should increase by a factor of $O(\frac{1}{(1-\delta)^4})$. This means that the list size increases to $\tilde{O}\left(2^{\tilde{O}(\frac{k}{\varepsilon(1-\delta)})}\right)$. Note that when $\delta \leq \frac{1}{2}$, the list size and running time retain the same form as that in Bhattacharya et al. (2020b) (i.e., $|\mathcal{L}| = 2^{\tilde{O}(\frac{k}{\varepsilon})}$ and time $O(Nd|\mathcal{L}|)$). A detailed proof of Theorem 8 is provided in the next section of the Appendix.

### D.3   THE QUANTUM ALGORITHM

The main idea of the quantum version of the approximation scheme of Bhattacharya et al. (2020b) is to $D^2$-sample a set $S$ of $poly(k/\varepsilon)$ points with respect to a center set $C_{init}$ that gives constant approximation to the $k$-means objective. We then try out all various subsets of possibilities for the $k$ center set out of $S$ and $C_{init}$ and then find the one with the least cost. $C_{init}$ is itself found quantumly by iteratively $D^2$-sampling $2k$ points. So, most of the quantum ideas developed for the quantum version of $k$-means++ can be reused in the quantum approximation scheme. An additional effort is in to calculate the $k$-means cost of a list of $L = 2^{\tilde{O}(\frac{k}{\varepsilon})}$ $k$-center sets and pick the one with the least cost. Here, instead of adding the relevant steps only, we add all the steps of the quantum algorithm for better readability.

We will work under the assumption that the minimum distance between two data points is 1, which can be achieved using scaling. This makes the aspect ratio $\zeta$ simply the maximum distance between two data points. We will use $i$ for an index into the rows of the data matrix $V \in \mathbb{R}^{N \times d}$, and $j$ for an index into the rows of the center matrix $C \in \mathbb{R}^{2k \times d}$. We would ideally like to design a quantum algorithm that performs the transformation:

$$|i\rangle |j\rangle |0\rangle \rightarrow |i\rangle |j\rangle |D(v_i, c_j)\rangle$$

Let us call the state on the right $|\Psi_{ideal}\rangle$. This is an ideal quantum state for us since $|\Psi_{ideal}\rangle$ helps to perform $D^2$-sampling and to find the $k$-means cost of clustering, which are the main components of the approximation scheme of Bhattacharya et al. (2020b) that we intend to use. One caveat is that we will only be able to perform the following transformation (instead of the abovementioned transformation)

$$|i\rangle |j\rangle |0\rangle \rightarrow |i\rangle |j\rangle |\psi_{i,j}\rangle,$$

where $|\psi_{i,j}\rangle$ is an approximation for $|\mathcal{D}(v_i, c_j)\rangle$ in a sense that we will make precise below. We will use $|\Psi_{real}\rangle$ to denote the state $|i\rangle |j\rangle |\psi_{i,j}\rangle$. This state is prepared using tools such as *swap test* followed by *coherent amplitude estimation*, and *median estimation*. Since these tools and techniques are known from previous works Wiebe et al. (2015); Lloyd et al. (2013); Kerenidis et al. (2019), we summarise the discussion (see Section 4.1 and 4.2 in Kerenidis et al. (2019)) in the following lemma.

**Lemma 12** (Kerenidis et al. (2019) and Wiebe et al. (2015))**.** *Assume for a data matrix $V \in \mathbb{R}^{N \times d}$ and a center set matrix $C \in \mathbb{R}^{t \times d}$ that the following unitaries: (i) $|i\rangle |0\rangle \rightarrow |i\rangle |v_i\rangle$, (ii) $|j\rangle |0\rangle \rightarrow |j\rangle |c_j\rangle$ can be performed in time $T$ and the norms of the vectors are known. For any $\Delta > 0$, there is a quantum algorithm that in time $O\left(\frac{T \log \frac{1}{\Delta}}{\varepsilon}\right)$ computes:*

$$|i\rangle |j\rangle |0\rangle \rightarrow |i\rangle |j\rangle |\psi_{i,j}\rangle,$$

*where $|\psi_{i,j}\rangle$ satisfies the following two conditions for every $i \in [N]$ and $j \in [t]$:*

(i) $\left\| |\psi_{i,j}\rangle - |0^{\otimes \ell}\rangle \left|\tilde{D}(v_i, c_j)\right\rangle \right\| \leq \sqrt{2\Delta}$, and

(ii) For every $i, j$, $\tilde{D}(v_i, c_j) \in (1 \pm \varepsilon) \cdot D(v_i, c_j)$.

In the subsequent discussions, we will use $T$ as the time to access the *QRAM data structure* Kerenidis & Prakash (2017), i.e., for the transitions $|i\rangle |0\rangle \to |i\rangle |v_i\rangle$ and $|j\rangle |0\rangle \to |j\rangle |c_j\rangle$ as given in the above lemma. This is known to be $T = O(\log^2(Nd))$. Moreover, the time to update each entry in this data structure is also $T = O(\log^2(Nd))$. This is the logarithmic factor that is hidden in the $\tilde{O}$ notation. In the following subsections, we discuss the utilities of $|\Psi_{real}\rangle$ for the various components of the approximation scheme of Bhattacharya et al. (2020b). During these discussions, it will be easier to see the utility first with the ideal state $|\Psi_{ideal}\rangle$ before the real state $|\Psi_{real}\rangle$ that can actually be prepared. We will see how $|\Psi_{real}\rangle$ is sufficient within a reasonable error bound.

### D.4 FINDING DISTANCE TO CLOSEST CENTER

Let us see how we can estimate the distance of any point to its closest center in a center set $C := C_{init}$ with $t := 2k$ centers. We can use the transformation $|i\rangle |j\rangle |0\rangle \to |i\rangle |j\rangle |D(v_i, c_j)\rangle$ to prepare the following state for any $i$:

$$|i\rangle |D(v_i, c_1)\rangle |D(v_i, c_2)\rangle ... |D(v_i, c_t)\rangle$$

We can then iteratively compare and swap pairs of registers to prepare the state $|i\rangle \left|\min_{j \in [t]} D(v_i, c_j)\right\rangle$. If we apply the same procedure to $|i\rangle |\psi_{i,1}\rangle ... |\psi_{i,t}\rangle$, then with probability at least $(1 - 2\Delta)^t$, the resulting state will be $|i\rangle \left|\min_{j \in [t]} \tilde{D}(v_i, c_j)\right\rangle$. So, the contents of the second register will be an estimate of the distance of the $i^{th}$ point to its closest center in the center set $C$. This further means that the following state can be prepared with probability at least $(1 - 2\Delta)^{Nt}$:[12]

$$|\Psi_C\rangle \equiv \frac{1}{\sqrt{N}} \sum_{i=1}^{N} |i\rangle \left|\min_{j \in [t]} \tilde{D}(v_i, c_j)\right\rangle.$$

This quantum state can be used to find the approximate clustering cost of the center set $C$, which we discuss in the following subsection. However, before we do that, let us summarise the main ideas of this subsection in the following lemma.

**Lemma 13.** *There is a quantum algorithm that, with probability at least $(1 - 2\Delta)^{Nt}$, prepares the quantum state $|\Psi_C\rangle$ in time $O\left(\frac{Tt \log \frac{1}{\Delta}}{\varepsilon}\right)$.*

### D.5 COMPUTING COST OF CLUSTERING

Suppose we want to compute the $k$-means cost, $\Phi(V, C) \equiv \sum_{i=1}^{N} \min_{j \in [t]} D^2(v_i, c_j)$, of the clustering given by a $t$ center set $C$. We can prepare $m$ copies of the state $|\Psi_C\rangle$ and then estimate the cost of clustering by measuring $m$ copies of this quantum state and summing the squares of the second registers. If $m$ is sufficiently large, we obtain a close estimate of $\Phi(V, C)$. To show this formally, we will use the following Hoeffding tail inequality.

**Theorem 9** (Hoeffding bound). *Let $X_1, ..., X_m$ be independent, bounded random variables such that $X_i \in [a, b]$. Let $S_m = X_1 + ... + X_m$. Then for any $\theta > 0$, we have:*

$$\mathbf{Pr}[|S_m - \mathbb{E}[S_m]| \geq \theta] \leq 2 \cdot e^{\frac{-2\theta^2}{m(b-a)^2}}.$$

Let $X_1, ..., X_m$ denote the square of the measured value of the second register in $|\Psi_C\rangle$. These are random values in the range $[1, \chi^2]$, where $\chi = \max_{i,j} \tilde{D}(v_i, c_j) \in (1 \pm \varepsilon) \cdot \max_{i,j} D(v_i, v_j)$. Note that by scaling, we assume that $\min_{i,j} D(v_i, v_j) = 1$. So, $\chi$ can be upper bounded in terms of the aspect ratio $\zeta$ as $\chi \leq 2\zeta$.[13] First, we note the expectation of these random variables equals

---

[12] The state prepared is actually $\frac{1}{\sqrt{N}} \sum_{i=1}^{N} |i\rangle \left(\alpha \left|\min_{j \in [t]} \tilde{D}(v_i, c_j)\right\rangle + \beta |G\rangle\right)$ with $|\alpha|^2 \geq (1 - 2\Delta)^{Nt}$. However, instead of working with this state, subsequent discussions become much simpler if we assume that $|\Psi_C\rangle$ is prepared with probability $|\alpha|^2$.

[13] In cases scaling only ensures $\min_{i,j} D(v_i, v_j) \geq 1$ (e.g., vectors with integer coordinates), we can use $\max_{i,j} D(v_i, v_j)$ as the parameter, instead of aspect ratio.

$\frac{\tilde{\Phi}(V,C)}{N}$, where $\tilde{\Phi}(V,C) \equiv \sum_{i=1}^{N} \min_{j \in [t]} \tilde{D}(v_i, c_j)^2 \in (1 \pm \varepsilon)^2 \cdot \Phi(V,C)$. We define the variable $S_m = X_1 + X_2 + ... + X_m$ and apply the Hoeffding bound on these bounded random variables to get a concentration result that can then be used.

**Lemma 14.** *Let $\alpha_m = S_m \cdot \frac{N}{m}$ and $L > 0$. If $m = O\left(\frac{\zeta^4 \ln (10L)}{\varepsilon^2}\right)$, then we have:*

$$\mathbf{Pr}[\alpha_m \in (1 \pm \varepsilon) \cdot \tilde{\Phi}(V,C)] \geq 1 - \frac{1}{5L}.$$

*Proof.* We know that $\mathbb{E}[S_m] = \frac{m}{N} \cdot \tilde{\Phi}(V,C)$ From the Hoeffding tail inequality, we get the following:

$$
\begin{aligned}
\mathbf{Pr}[|S_m - \mathbb{E}[S_m]| \geq \varepsilon \cdot \mathbb{E}[S_m]] &\leq 2 \cdot e^{\frac{-2\varepsilon^2 \mathbb{E}[S_m]^2}{m\zeta^4}} \\
&= 2 \cdot e^{\frac{-2\varepsilon^2 m}{\zeta^4} \cdot \left(\frac{\tilde{\Phi}(V,C)}{N}\right)^2} \\
&\leq 2 \cdot e^{-\ln(10L)} \leq \frac{1}{5L}.
\end{aligned}
$$

This implies that:

$$\mathbf{Pr}[|\alpha_m - \tilde{\Phi}(V,C)| \geq \varepsilon \cdot \tilde{\Phi}(V,C)] \leq \frac{1}{5L}.$$

This completes the proof of the lemma. $\qquad \square$

So, conditioned on having $m$ copies of the state $|\Psi_C\rangle$, we get an estimate of the clustering cost within a relative error of $(1 \pm \varepsilon)^3$ with probability at least $(1 - \frac{1}{5L})$. Removing the conditioning, we get the same with probability at least $(1 - 2\Delta)^{Nkm} \cdot (1 - \frac{1}{5L})$. We want to use the above cost estimation technique to calculate the cost for a *list* of center sets $C_1, ..., C_L$, and then pick the center set from the list with the least cost. We must apply the union bound appropriately to do this with high probability. We summarise these results in the following lemma. Let us first set some of the parameters with values that we will use to implement the approximation scheme of Bhattacharya et al. (2020b).

- $L$ denotes the size of the list of $k$-center sets we will iterate over to find the one with the least cost. This quantity is bounded as $L = \left(\frac{k}{\varepsilon}\right)^{O(\frac{k}{\varepsilon})}$.

- $m$ is the number of copies of the state $|\Psi_C\rangle$ made to estimate the cost of the center set $C$. This, as given is Lemma 14 is $m = \frac{\zeta^4 \ln(10L)}{\varepsilon^2}$, where $\zeta = (1 + \varepsilon) \cdot \max_{i,j} D(v_i, c_j)$.

**Lemma 15.** *Let $L = \left(\frac{k}{\varepsilon}\right)^{O(\frac{k}{\varepsilon})}$, $m = \frac{\zeta^4 \ln(10L)}{\varepsilon^2}$, and $\Delta = O\left(\frac{1}{NkmL}\right)$. Given a point set $V$ and a list of center sets $C_1, ..., C_L$ in the QRAM model, there is a quantum algorithm that runs in time $\tilde{O}\left(2^{\tilde{O}(\frac{k}{\varepsilon})} T \zeta^4\right)$ and outputs an index $l$ such that $\Phi(V, C_l) \leq (1 + \varepsilon)^3 \min_{j \in L} \Phi(V, C_j)$ with probbaility at least $\frac{3}{5}$.*

*Proof.* The algorithm estimates the cost of $C_1, ..., C_L$ using $m$ copies each of $|\Psi_{C_1}\rangle, ..., |\Psi_{C_L}\rangle$ and picks the index with the minimum value in time $O\left(\frac{TkmL \log \frac{1}{\Delta}}{\varepsilon}\right)$. Plugging the values of $L, m$, and $\Delta$ we get the running time stated in the lemma.

Let us bound the error probability of this procedure. By Lemma 3, the probability that we do not have the correct $m$ copies each of $|\Psi_{C_1}\rangle, ..., |\Psi_{C_L}\rangle$ is bounded by $1 - (1 - 2\Delta)^{NkmL}$. Conditioned on having $|\Psi_{C_1}\rangle, ..., |\Psi_{C_L}\rangle$, the probability that there exists an index $j \in [L]$, where the estimate is off by more than a $(1 \pm \varepsilon)^3$ factor is upper bounded by $\frac{1}{5}$ by the union bound. So, the probability that the algorithm will find an index $l$ such that $\Phi(V, C_l) > (1 + \varepsilon)^3 \min_{j \in [L]} \Phi(V, C_j)$ is upper bounded by $1 - (1 - 2\Delta)^{NkmL} + \frac{1}{5}$. This probability is at most $\frac{2}{5}$ since $\Delta = O(\frac{1}{NkmL})$. This completes the proof of the lemma. $\qquad \square$

### D.6 $D^2$-SAMPLING

$D^2$-sampling from the point set $V$ with respect to a center set $C \in \mathbb{R}^{t \times d}$ with $t$ centers, samples $v_i$ with probability proportional to $\min_{j \in [t]} D^2(v_i, c_j)$. Let us see if we can use our state $|\Psi_C\rangle =$

$\frac{1}{\sqrt{N}} \sum_{i=1}^{N} |i\rangle \left| \min_{j \in [t]} \tilde{D}(v_i, c_j) \right\rangle$ is useful to perform this sampling. If we can pull out the value of the second register as the amplitude, then the measurement will give us close to $D^2$-sampling. This is possible since we have an estimate of the clustering cost from the previous subsection. We can use controlled rotations on an ancilla qubit to prepare the state:

$$|\Psi_{sample}\rangle \equiv \frac{1}{\sqrt{N}} \sum_{i=1}^{N} |i\rangle \left( \beta_i |0\rangle + \sqrt{1 - |\beta_i|^2} |1\rangle \right),$$

where $\beta_i = \frac{\min_{j \in [t]} \tilde{D}(v_i, c_j)}{\zeta \sqrt{4 \cdot \tilde{\Phi}(V,C)/N}}$. So, the probability of measurement of $(i, 0)$ is $\frac{\min_{j \in [t]} \tilde{D}(v_i, c_j)^2}{4\zeta^2 \cdot \tilde{\Phi}(V,C)}$. Since we do rejection sampling, ignoring $(., 1)$'s that are sampled with probability $\leq (1 - \frac{1}{4\zeta^2})$, we end up sampling with a distribution where the probability of sampling $i$ is $\frac{\min_{j \in [t]} \tilde{D}(v_i, c_j)^2}{\tilde{\Phi}(V,C)} \in$ $(1 \pm \varepsilon)^4 \cdot \frac{\min_{j \in [t]} D(v_i, c_j)^2}{\Phi(V,C)}$. Moreover, we get a usable sample with high probability in at most $O(\zeta^2 \log 10N)$ rounds of sampling. This means that points get sampled with a probability close to the actual $D^2$-sampling probability. As we have mentioned earlier, this is sufficient for the approximation guarantees of Bhattacharya et al. (2020b) to hold. We summarise the observations of this section in the next lemma.

**Lemma 16.** *Given a dataset $V \in \mathbb{R}^{N \times d}$ and a center set $C \in \mathbb{R}^{t \times d}$ in the QRAM model, there is a quantum algorithm that runs in time $O\left( \frac{TtmS \log \frac{1}{\Delta}}{\delta} \right)$ and with probability at least $(1 - 2\Delta)^{NtmS}$ outputs $S$ independent samples with $\mathcal{D}^2$ distribution such that the distance function $\mathcal{D}$ is $\delta$-close to $D$.*

*Proof.* The proof follows from Lemma 13 and the preceding discussion. $\square$

The above lemma says that for $\Delta = \tilde{O}(\frac{1}{NkmS})$, we obtain the required samples with high probability. We can now give proof of Theorem 4, assembling the quantum tools of this section. We restate the theorem below for ease of reading.

**Theorem** (Restatement of Theorem 4). *Let $0 < \varepsilon < 1/2$ be the error parameter. There is a quantum algorithm that, when given QRAM data structure access to a dataset $V \in \mathbb{R}^{N \times d}$, runs in time $\tilde{O}\left( 2^{\tilde{O}(\frac{k}{\varepsilon})} d \zeta^{O(1)} \right)$ and outputs a $k$ center set $C \in \mathbb{R}^{k \times d}$ such that with high probability $\Phi(V, C) \leq (1 + \varepsilon) \cdot OPT$. Here, $\zeta$ is the aspect ratio, i.e., the ratio of the maximum to the minimum distance between two given points in $V$.*[14]

*Proof of Theorem 4.* The first requirement for executing the algorithm of Bhattacharya et al. (2020b) is a constant pseudo approximation algorithm, using which we obtain the initial center set $C_{init}$. By Lemma 11, we know that $2k$ points sampled using $\mathcal{D}^2$-sampling give such a center set. From Lemma 16, this can be done quantumly in time $\tilde{O}(\frac{k^2 \zeta^{O(1)}}{\varepsilon^2})$. The algorithm of Bhattacharya et al. (2020b) has an outer repeat loop for probability amplification. Within the outer loop, $poly(\frac{k}{\varepsilon})$ points are $\mathcal{D}^2$-sampled with respect to the center set $C_{init}$ (line 6). This can again be done quantumly using Lemma 16 in time $\tilde{O}(\zeta^{O(1)}(k/\varepsilon)^{O(1)})$. We can then classically process the point set $M$ (see line 7 in Algorithm 4) and create the QRAM data structure for the list $C_1, ..., C_L$ of $k$-center sets that correspond to all possible disjoint subsets of $M$ (see line 8 in Algorithm 4). This takes time $\tilde{O}(Lkd)$, where $L = \left(\frac{k}{\varepsilon}\right)^{O(\frac{k}{\varepsilon})}$. Theorem 8 shows that at least one center set in the list gives $(1 + \varepsilon)$-approximation. We use this fact in conjunction with the result of Lemma 15 to get that the underlying quantum algorithm runs in time $\tilde{O}(2^{\tilde{O}(\frac{k}{\varepsilon})} d \zeta^{O(1)})$ and with high probability outputs a center set $C'$ such that $\Phi(V, C') \leq (1 + \varepsilon)^4 \cdot OPT$.[15] $\square$

---

[14]The $\tilde{O}$ notation hides logarithmic factors in $N$. The $\tilde{O}$ in the exponent hides logarithmic factors in $k$ and $1/\varepsilon$.

[15]We needed $(1 + \varepsilon)$, but got $(1 + \varepsilon)^4$ instead. However, this can be handled with $\varepsilon' = \varepsilon/5$.

# E  A Robust Approximation Scheme for $k$-means (proof of Theorem 8)

This section gives the proof of Theorem 8. We restate the theorem and the algorithm for readability. We state the algorithm of Bhattacharya *et al.*  Bhattacharya et al. (2020b) replacing $D^2$-sampling with $\mathcal{D}^2$-sampling.

---

**Algorithm 5** Algorithm of Bhattacharya et al. (2020b) with $D$ replaced with $\delta$-close $\mathcal{D}$

---

1: **Input**: $(V, k, \varepsilon, C_{init})$, where $V$ is the dataset, $k > 0$ is the number of clusters, $\varepsilon > 0$ is the error parameter, and $C_{init}$ is a $k$ center set that gives constant (pseudo)approximation.
2: **Output**: A list $\mathcal{L}$ of $k$ center sets such that for at least one $C' \in \mathcal{L}$, $\Phi(V, C') \leq (1 + \varepsilon) \cdot OPT$.
3: **Constants**: $\rho = O(\frac{k}{(1-\delta)^4 \varepsilon^4}); \tau = O(\frac{1}{\varepsilon})$
4: $\mathcal{L} \leftarrow \emptyset; count \leftarrow 1$
5: **repeat**
6:     Sample a multi-set $M$ of $\rho k$ points from $V$ using $\mathcal{D}^2$-sampling wrt center set $C_{init}$
7:     $M \leftarrow M \cup \{\tau k$ copies of each element in $C_{init}\}$
8:     **for all** disjoint subsets $S_1, ..., S_k$ of $M$ such that $\forall i, |S_i| = \tau$ **do**
9:         $\mathcal{L} \leftarrow \mathcal{L} \cup (\mu(S_1), ..., \mu(S_k))$
10:     **end for**
11:     $count$++
12: **until** $count < 2^k$
13: **return** $\mathcal{L}$

---

**Theorem** (Restatement of Theorem 8). *Let $0 < \varepsilon \leq \frac{1}{2}$ be the error parameter, $0 < \delta < 1/2$ be the closeness parameter, $V \in \mathbb{R}^{N \times d}$ be the dataset, $k$ be a positive integer, and let $C_{init}$ be a constant (pseudo)approximate solution for dataset $V$. Let $\mathcal{L}$ be the list returned by Algorithm 5 on input $(V, k, \varepsilon, C_{init})$ using the distance function $\mathcal{D}$ that is $\delta$-close to the Euclidean distance function $D$. Then with probability at least $3/4$, $\mathcal{L}$ contains a center set $C$ such that $\Phi(V, C) \leq (1 + \varepsilon) \cdot OPT$. Moreover, $|\mathcal{L}| = 2^{\tilde{O}(\frac{k}{\varepsilon})}$ and the running time of the algorithm is $O(Nd|\mathcal{L}|)$.*

Note that what this theorem essentially says is that if the error in the distance estimates is bounded within a multiplicative factor of $(1 \pm \delta)$ with $\delta \leq 1/2$, then the guarantees of the algorithm of Bhattacharya *et al.*  Bhattacharya et al. (2020b) do not change except for some constant factors (that get absorbed in the $\tilde{O}$ of the exponent).

We need to define a few quantities that will be used to prove the above theorem. Let $\{X_1, ..., X_k\}$ be the optimal clusters. It is well-known that the center that optimizes the 1-means cost for any pointset is the centroid of the pointset. We denote the centroid of any pointset $Y \subset \mathbb{R}^d$ with $\mu(Y) \equiv \frac{\sum_{y \in Y} y}{|Y|}$. This follows from the following well-known fact:

**Fact 1.** *For any $Y \subset \mathbb{R}^d$ and any $c \in \mathbb{R}^d$, we have $\sum_{y \in Y} \|y - c\|^2 = \sum_{y \in Y} \|y - \mu(Y)\|^2 + |Y| \cdot \|c - \mu(Y)\|^2$.*

We have used the norm notation instead of $D(.,.)$ for the Euclidean distance function. This is to easily distinguish between the usage of $D$ and $\mathcal{D}$. We define the optimal 1-means cost for pointset $X_i$ as $\Delta(X_i) \equiv \sum_{x \in X_i} \|x - \mu(X_i)\|^2$. We denote the optimal $k$-means cost using $OPT \equiv \sum_{i=1}^{k} \Delta(X_i)$. We will use the following sampling lemma from Inaba *et al.*  Inaba et al. (1994), which says that the centroid of a small set of uniformly sampled points from the dataset $Y$ is a good center with respect to the 1-means cost for dataset Y.

**Lemma 17** (Inaba et al. (1994)). *Let $S$ be a set of points obtained by independently sampling $M$ points with replacement uniformly at random from a point set $Y \subset \mathbb{R}^d$. Then for any $\delta > 0$,*

$$\mathbf{Pr}\left[\Phi(\mu(S), Y) \leq \left(1 + \frac{1}{\delta M}\right) \cdot \Delta(Y)\right] \geq (1 - \delta).$$

We will also use the following approximate triangle inequality that holds for squared distances.

**Fact 2** (Approximate triangle inequality). *For any $x, y, z \in \mathbb{R}^d$, we have $\|x - z\|^2 \leq 2 \cdot (\|x - y\|^2 + \|y - z\|^2)$.*

The proof of Theorem 8 follows from the following theorem, which we will prove in the remaining section.

**Theorem 10.** *Let $0 < \varepsilon, \delta \leq 1/2$. Let $\mathcal{L}$ denote the list returned by Algorithm 5 on input $(V, k, \varepsilon, C_{init})$. Then with probability at least $3/4$, $\mathcal{L}$ contains a center set $C$ such that*

$$\Phi\left(C, \{X_1, ..., X_k\}\right) \leq \left(1 + \frac{\varepsilon}{2}\right) \cdot \sum_{i=1}^{k} \Delta(X_i) + \frac{\varepsilon}{2} \cdot OPT.$$

*Moreover, $|\mathcal{L}| = 2^{\tilde{O}(\frac{k}{\varepsilon})}$ and the running time of the algorithm is $O(nd|\mathcal{L}|)$.*

Let $C_{init}$ be an $(\alpha, \beta)$-approximate solution to the $k$-means problem on the dataset $V$. That is:

$$\Phi(V, C_{init}) \leq \alpha \cdot OPT \quad \text{and} \quad |C_{init}| \leq \beta \cdot k.$$

Let us now analyze the algorithm. Note that the outer repetition (lines 5-12) operation is to amplify the probability of the list containing a good $k$ center set. We will show that the probability of finding a good $k$ center set in one iteration is at least $(3/4)^k$, and the theorem will follow from standard probability analysis. So, we shall focus on only one iteration of the outer loop. Consider the multi-set $M$ obtained on line (7) of Algorithm 5. We will show that with probability at least $(3/4)^k$, there are disjoint multi-subsets $T_1, ..., T_k$ each of size $\tau$ such that for every $i = 1, ..., k$:

$$\Phi(\mu(T_i), X_i) \leq \left(1 + \frac{\varepsilon}{2}\right) \cdot \Delta(X_i) + \frac{\varepsilon}{2k} \cdot OPT. \tag{5}$$

Since we try all possible subsets of $M$ in line (8), we will get the desired result. More precisely, we will argue in the following manner: consider the multi-set $C' = \{\tau k$ copies of each element in $C_{init}\}$. We can interpret $C'$ as a union of multi-sets $C'_1, C'_2, ..., C'_k$, where $C'_i = \{\tau$ copies of each element in $C_{init}\}$. Also, since $M$ consists of $\rho k$ independently sampled points, we can interpret $M$ as a union of multi-sets $M'_1, M'_2, ..., M'_k$ where $M'_1$ is the first $\rho$ points sampled, $M'_2$ is the second $\rho$ points and so on. For all $i = 1, ..., k$, let $M_i = C'_i \cup (M'_i \cap X_i)$.[16] We will show that for every $i \in \{1, ..., k\}$, with probability at least $(3/4)$, $M_i$ contains a subset $T_i$ of size $\tau$ that satisfies Eqn. (5). Note that $T_i$'s being disjoint follows from the definition of $M_i$. It will be sufficient to prove the following lemma.

**Lemma 18.** *Consider the sets $M_1, ..., M_k$ as defined above. For any $i \in \{1, ..., k\}$,*

$$\mathbf{Pr}\left[\exists T_i \subseteq M_i \text{ s.t. } |T_i| = \tau \text{ and } \left(\Phi(\mu(T_i), X_i) \leq \left(1 + \frac{\varepsilon}{2}\right) \cdot \Delta(X_i) + \frac{\varepsilon}{2k}OPT\right)\right] \geq \frac{3}{4}.$$

We prove the above lemma in the remaining discussion. We do a case analysis that is based on whether $\frac{\Phi(C_{init}, X_i)}{\Phi(C_{init}, V)}$ is large or small for a particular $i \in \{1, ..., k\}$.

- *Case-I* $\left(\Phi(C_{init}, X_i) \leq \frac{\varepsilon}{6\alpha k} \cdot \Phi(C_{init}, V)\right)$: Here, we will show that there is a subset $T_i \subseteq C'_i \subseteq M_i$ that satisfies Eqn. (5).

- *Case-II* $\left(\Phi(C_{init}, X_i) > \frac{\varepsilon}{6\alpha k} \cdot \Phi(C_{init}, V)\right)$: Here we will show that $M_i$ contains a subset $T_i$ such that $\Phi(\mu(T_i), X_i) \leq \left(1 + \frac{\varepsilon}{2}\right) \cdot \Delta(X_i)$ and hence $T_i$ also satisfies Eqn. (5).

We will discuss these two cases next.

**Case-I:** $\left(\Phi(C_{init}, X_i) \leq \frac{\varepsilon}{6\alpha k} \cdot \Phi(C_{init}, V)\right)$ First, observe that:

$$\Phi(C_{init}, X_i) \leq \frac{\varepsilon}{6k} \cdot OPT \qquad \text{(since $C_{init}$ is an $\alpha$-approximate solution)} \tag{6}$$

For any point $x \in V$, let $c(x)$ denote the center in the set $C$ that is closest to $x$. That is, $c(x) = \arg\min_{c \in C_{init}} \|c - x\|$. Given this definition, note that:

$$\sum_{x \in X_i} \|x - c(x)\|^2 = \Phi(C_{init}, X_i) \tag{7}$$

We define the multi-set $X'_i = \{c(x) : x \in X_i\}$. Let $m$ and $m'$ denote the centroids of the point sets $X_i$ and $X'_i$, respectively. So, we have $\Delta(X_i) = \Phi(m, X_i)$ and $\Delta(X'_i) = \Phi(m', X'_i)$. We will show that $\Delta(X_i) \approx \Delta(X'_i)$. First, we bound the distance between $m$ and $m'$.

---

[16] $M'_i \cap X_i$ in this case, denotes those points in the multi-set $M'_i$ that belongs to $X_i$.

**Lemma 19.** $\|m - m'\|^2 \leq \frac{\Phi(C_{init}, X_i)}{|X_i|}$.

*Proof.* We have:

$$\|m - m'\|^2 = \frac{\left\|\sum_{x \in X_i}(x - c(x))\right\|^2}{|X_i|^2} \leq \frac{\sum_{x \in X_j}\|(x - c(x))\|^2}{|X_j|} = \frac{\Phi(C_{init}, X_j)}{|X_j|}.$$

where the second to last inequality follows from Cauchy-Schwartz. $\square$

**Lemma 20.** $\Delta(X_i') \leq 2 \cdot \Phi(C_{init}, X_i) + 2 \cdot \Delta(X_i)$.

*Proof.* We have:

$$\Delta(X_i') \quad = \quad \sum_{x \in X_i}\|c(x) - m'\|^2 \leq \sum_{x \in X_i}\|c(x) - m\|^2$$

$$\overset{(Fact\ 2)}{\leq} \quad 2 \cdot \sum_{x \in X_i}(\|c(x) - x\|^2 + \|x - m\|^2) = 2 \cdot \Phi(C_{init}, X_i) + 2 \cdot \Delta(X_i)$$

This completes the proof of the lemma. $\square$

We now show that a good center for $X_i'$ will also be a good center for $X_i$.

**Lemma 21.** *Let $m''$ be a point such that $\Phi(m'', X_i') \leq (1 + \frac{\varepsilon}{8}) \cdot \Delta(X_i')$. Then $\Phi(m'', X_i) \leq (1 + \frac{\varepsilon}{2}) \cdot \Delta(X_i) + \frac{\varepsilon}{2k} \cdot OPT$.*

*Proof.* We have:

$$\Phi(m'', X_i) \quad \overset{(Fact\ 1)}{=} \quad \sum_{x \in X_i}\|x - m\|^2 + |X_i| \cdot \|m - m''\|^2$$

$$\overset{(Fact\ 2)}{\leq} \quad \Delta(X_i) + 2|X_i| \cdot (\|m - m'\|^2 + \|m' - m''\|^2)$$

$$\overset{(Lemma\ 19)}{\leq} \quad \Delta(X_i) + 2 \cdot \Phi(C_{init}, X_i) + 2|X_i| \cdot \|m' - m''\|^2$$

$$\overset{(Fact\ 1)}{\leq} \quad \Delta(X_i) + 2 \cdot \Phi(C_{init}, X_i) + 2(\Phi(m'', X_i') - \Delta(X_i'))$$

$$\leq \quad \Delta(X_i) + 2 \cdot \Phi(C, X_i) + \frac{\varepsilon}{4} \cdot \Delta(X_i')$$

$$\overset{(Lemma\ 20)}{\leq} \quad \Delta(X_i) + 2 \cdot \Phi(C_{init}, X_i) + \frac{\varepsilon}{2} \cdot (\Phi(C_{init}, X_i) + \Delta(X_i))$$

$$\overset{(Eqn.\ 6)}{\leq} \quad \left(1 + \frac{\varepsilon}{2}\right) \cdot \Delta(X_i) + \frac{\varepsilon}{2k} \cdot OPT.$$

This completes the proof of the lemma. $\square$

We know from Lemma 17 that there exists a (multi) subset of $X_i'$ of size $\tau$ such that the mean of these points satisfies the condition of the lemma above. Since $C_i'$ contains at least $\tau$ copies of every element of $C_{init}$, there is guaranteed to be a subset $T_i \subseteq C_i'$ that satisfies Eqn. (5). So, for any index $i \in \{1, ..., k\}$ such that $\frac{\Phi(C_{init}, X_i)}{\Phi(C_{init}, V)} \leq \frac{\varepsilon}{6\alpha k}$, $M_i$ has a good subset $T_i$ with probability 1.

**Case-II:** $\left(\Phi(C_{init}, X_i) > \frac{\varepsilon}{6\alpha k} \cdot \Phi(C_{init}, V)\right)$ If we can show that a $\mathcal{D}^2$-sampled set with respect to center set $C_{init}$ has a subset $S$ that may be considered a uniform sample from $X_i$, then we can use Lemma 17 to argue that $M_i$ has a subset $T_i$ such that $\mu(T_i)$ is a good center for $X_i$. Note that since $\frac{\Phi(C_{init}, X_j)}{\Phi(C_{init}, V)} > \frac{\varepsilon}{6\alpha k}$, and $D$ is $\delta$-close to $\mathcal{D}$, we can argue that if we $\mathcal{D}^2$-sample $poly(\frac{k}{\varepsilon})$ elements, then we will get a good representation from $X_i$. However, some of the points from $X_i$ may be very close to one of the centers in $C_{init}$ and hence will have a very small chance of being $\mathcal{D}^2$-sampled. In such a case, no subset $S$ of a $\mathcal{D}^2$-sampled set will behave like a uniform sample from $X_i$. So, we need to argue more carefully taking into consideration the fact that there may be points in $X_i$ for which the chance of being $\mathcal{D}^2$-sampled may be very small. Here is the high-level argument that we will build:

- Consider the set $X_i'$ which is same as $X_i$ except that points in $X_i$ that are very close to $C_{init}$ have been "collapsed" to their closest center in $C_{init}$.

- Argue that a good center for the set $X_i'$ is a good center for $X_i$.

- Show that a convex combination of copies of centers in $C_{init}$ (i.e., $C_i'$) and $\mathcal{D}^2$-sampled points from $X_i$ gives a good center for the set $X_i'$.

The closeness of point in $X_i$ to points in $C_{init}$ is quantified using radius $R$ that is defined by the equation:

$$R^2 \stackrel{defn.}{=} \frac{\varepsilon^2}{41} \cdot \frac{\Phi(C_{init}, X_i)}{|X_i|}. \tag{8}$$

Let $X_i^{near}$ be the points in $X_i$ that are within a distance of $R$ from a point in set $C_{init}$ and $X_i^{far} = X_i \setminus X_i^{near}$. That is, $X_i^{near} = \{x \in X_i : \min_{c \in C_{init}} \|x - c\| \leq R\}$ and $X_i^{far} = X_i \setminus X_i^{near}$. Using these, we define the multi-set $X_i'$ as:

$$X_i' = X_i^{far} \cup \{c(x) : x \in X_i^{near}\}$$

Note that $|X_i| = |X_i'|$. Let $m = \mu(X_i)$, $m' = \mu(X_i')$. Let $n = |X_i|$ and $\bar{n} = |X_i^{near}|$. We first show a lower bound on $\Delta(X_i)$ in terms of $R$.

**Lemma 22.** $\Delta(X_i) \geq \frac{16\bar{n}}{\varepsilon^2} R^2$.

*Proof.* Let $c = \arg\min_{c' \in C_{init}} \|m - c'\|$. We do a case analysis:

1. Case 1: $\|m - c\| \geq \frac{5}{\varepsilon} \cdot R$
   Consider any point $p \in X_i^{near}$. From triangle inequality, we have:

$$\|p - m\| \geq \|c(p) - m\| - \|c(p) - p\| \geq \frac{5}{\varepsilon} \cdot R - R \geq \frac{4}{\varepsilon} \cdot R.$$

   This gives: $\Delta(X_i) \geq \sum_{p \in X_i^{near}} \|p - m\|^2 \geq \frac{16\bar{n}}{\varepsilon^2} \cdot R^2$.

2. Case 2: $\|m - c\| < \frac{5}{\varepsilon} \cdot R$
   In this case, we have:

$$\Delta(X_i) = \Phi(c, X_i) - n \cdot \|m - c\|^2 \geq \Phi(C_{init}, X_i) - n \cdot \|m - c\|^2 \geq \frac{41n}{\varepsilon^2} \cdot R^2 - \frac{25n}{\varepsilon^2} \cdot R^2 \geq \frac{16\bar{n}}{\varepsilon^2} \cdot R^2.$$

This completes the proof of the lemma. $\qquad\square$

We now bound the distance between $m$ and $m'$ in terms of $R$.

**Lemma 23.** $\|m - m'\|^2 \leq \frac{\bar{n}}{n} \cdot R^2$.

*Proof.* Since $|X_i| = |X_i'|$ and the only difference between $X_i$ and $X_i'$ are the points corresponding to $X_i^{near}$, we have:

$$\|m - m'\|^2 = \frac{1}{(n)^2} \left\| \sum_{p \in X_i^{near}} (p - c(p)) \right\|^2 \leq \frac{\bar{n}}{(n)^2} \sum_{p \in X_i^{near}} \|p - c(p)\|^2 \leq \frac{\bar{n}^2}{(n)^2} R^2 \leq \frac{\bar{n}}{n} R^2.$$

The second inequality above follows from the Cauchy-Schwarz inequality. $\qquad\square$

We now show that $\Delta(X_i)$ and $\Delta(X_i')$ are close.

**Lemma 24.** $\Delta(X_i') \leq 4\bar{n}R^2 + 2 \cdot \Delta(X_i)$.

*Proof.* The lemma follows from the following sequence of inequalities:

$$\Delta(X_i') \quad = \quad \sum_{p \in X_i^{near}} \|c(p) - m'\|^2 + \sum_{p \in X_i^{far}} \|p - m'\|^2$$

$$\overset{(Fact\ 2)}{\leq} \quad \sum_{p\in X_i^{near}} 2\cdot\left(\|c(p)-p\|^2 + \|p-m'\|^2\right) + \sum_{p\in X_i^{far}} \|p-m'\|^2$$

$$\leq \quad 2\bar{n}R^2 + 2\cdot\Phi(m',X_i)$$

$$\overset{(Fact\ 1)}{\leq} \quad 2\bar{n}R^2 + 2\cdot\left(\Phi(m,X_i) + n\cdot\|m-m'\|^2\right)$$

$$\overset{(Lemma\ 23)}{\leq} \quad 4\bar{n}R^2 + 2\cdot\Delta(X_i)$$

This completes the proof of the lemma. $\qquad\square$

We now argue that any center that is good for $X_i'$ is also good for $X_i$.

**Lemma 25.** *Let $m''$ be such that $\Phi(m'',X_i') \leq \left(1+\frac{\varepsilon}{16}\right)\cdot\Delta(X_i')$. Then $\Phi(m'',X_i) \leq \left(1+\frac{\varepsilon}{2}\right)\cdot\Delta(X_i)$.*

*Proof.* The lemma follows from the following inequalities:

$$\Phi(m'',X_i) \quad = \quad \sum_{p\in X_i}\|m''-p\|^2$$

$$\overset{(Fact\ 1)}{=} \quad \sum_{p\in X_i}\|m-p\|^2 + n\cdot\|m-m''\|^2$$

$$\overset{(Fact\ 2)}{\leq} \quad \Delta(X_i) + 2n\left(\|m-m'\|^2 + \|m'-m''\|^2\right)$$

$$\overset{(Lemma\ 23)}{\leq} \quad \Delta(X_i) + 2\bar{n}R^2 + 2n\cdot\|m'-m''\|^2$$

$$\overset{(Fact\ 1)}{\leq} \quad \Delta(X_i) + 2\bar{n}R^2 + 2\cdot\left(\Phi(m'',X_i') - \Delta(X_i')\right)$$

$$\overset{(Lemma\ hypothesis)}{\leq} \quad \Delta(X_i) + 2\bar{n}R^2 + \frac{\varepsilon}{8}\cdot\Delta(X_i')$$

$$\overset{(Lemma\ 24)}{\leq} \quad \Delta(X_i) + 2\bar{n}R^2 + \frac{\varepsilon}{2}\cdot\bar{n}R^2 + \frac{\varepsilon}{4}\cdot\Delta(X_i)$$

$$\overset{(Lemma\ 22)}{\leq} \quad \left(1+\frac{\varepsilon}{2}\right)\cdot\Delta(X_i).$$

This completes the proof of the lemma. $\qquad\square$

Given the above lemma, all we need to argue is that our algorithm indeed considers a center $m''$ such that $\Phi(m'',X_i') \leq (1+\varepsilon/16)\cdot\Delta(X_i')$. For this we would need about $\Omega(\frac{1}{\varepsilon})$ uniform samples from $X_i'$. However, our algorithm can only sample using $\mathcal{D}^2$-sampling w.r.t. $C_{init}$. For ease of notation, let $c(X_i^{near})$ denote the multi-set $\{c(p) : p \in X_i^{near}\}$. Recall that $X_i'$ consists of $X_i^{far}$ and $c(X_i^{near})$. The first observation we make is that the probability of sampling an element from $X_i^{far}$ is reasonably large (proportional to $\frac{\varepsilon}{k}$). Using this fact, we show how to sample from $X_i'$ (almost uniformly). Finally, we show how to convert this almost uniform sampling to uniform sampling (at the cost of increasing the size of sample).

**Lemma 26.** *Let $x$ be a sample from $\mathcal{D}^2$-sampling w.r.t. $C_{init}$. Then, $\mathbf{Pr}[x \in X_i^{far}] \geq \frac{\varepsilon}{8\alpha k}$. Further, for any point $p \in X_i^{far}$, $\mathbf{Pr}[x=p] \geq \frac{\gamma}{|X_i|}$, where $\gamma$ denotes $\frac{(1-\delta)^4\varepsilon^3}{246\alpha k}$.*

*Proof.* Since points are $\mathcal{D}^2$-sampled with $\mathcal{D}$ being $\delta$-close to $D$, we note that $\sum_{p\in X_i^{near}}\mathbf{Pr}[x=p] \leq \frac{(1+\delta)^2}{(1-\delta)^2}\cdot\frac{R^2}{\Phi(C_{init},V)}\cdot|X_i| \leq \frac{(1+\delta)^2}{(1-\delta)^2}\cdot\frac{\varepsilon^2}{41}\cdot\frac{\Phi(C_{init},X_i)}{\Phi(C_{init},V)}$. Therefore, the fact that we are in case II implies that:

$$\mathbf{Pr}[x\in X_i^{far}] \geq \mathbf{Pr}[x\in X_i] - \mathbf{Pr}[x\in X_i^{near}] \geq \frac{\Phi(C_{init},X_i)}{\Phi(C_{init},V)} - \frac{(1+\delta)^2}{(1-\delta)^2}\cdot\frac{\varepsilon^2}{41}\cdot\frac{\Phi(C_{init},X_i)}{\Phi(C_{init},V)} \geq \frac{\varepsilon}{8\alpha k}.$$

Also, if $x \in X_i^{far}$, then $\Phi(C_{init}, \{x\}) \geq R^2 = \frac{\varepsilon^2}{41} \cdot \frac{\Phi(C_{init}, X_i)}{|X_i|}$. Therefore the probability that a point $p \in X_i^{far}$ gets $\mathcal{D}^2$-sampled is,

$$
\begin{aligned}
\mathbf{Pr}[x = p] &\geq \frac{(1-\delta)^2}{(1+\delta)^2} \cdot \frac{\Phi(C_{init}, \{x\})}{\Phi(C_{init}, V)} &&\geq \frac{(1-\delta)^2}{(1+\delta)^2} \cdot \frac{\varepsilon}{6\alpha k} \cdot \frac{R^2}{\Phi(C_{init}, X_i)} \\
&&&\geq \frac{(1-\delta)^2}{(1+\delta)^2} \cdot \frac{\varepsilon}{6\alpha k} \cdot \frac{\varepsilon^2}{41} \cdot \frac{1}{|X_i|} \\
&&&\geq \frac{(1-\delta)^2}{(1+\delta)^2} \cdot \frac{\varepsilon^3}{246\alpha k} \cdot \frac{1}{|X_i|} \geq \frac{(1-\delta)^4 \varepsilon^3}{246\alpha k} \cdot \frac{1}{|X_i|}.
\end{aligned}
$$

This completes the proof of the lemma. $\qquad\square$

Let $O_1, \ldots O_\rho$ be $\rho$ points sampled independently using $\mathcal{D}^2$-sampling w.r.t. $C_{init}$. We construct a new set of random variables $Y_1, \ldots, Y_\rho$. Each variable $Y_u$ will depend on $O_u$ only and will take values either in $X_i'$ or will be $\perp$. These variables are defined as follows: if $O_u \notin X_i^{far}$, we set $Y_u$ to $\perp$. Otherwise, we assign $Y_u$ to one of the following random variables with equal probability: (i) $O_u$ or (ii) a random element of the multi-set $c(X_i^{near})$. The following observation follows from Lemma 26.

**Corollary 1.** *For a fixed index $u$, and an element $x \in X_i'$, $\mathbf{Pr}[Y_u = x] \geq \frac{\gamma'}{|X_i'|}$, where $\gamma' = \gamma/2$.*

*Proof.* If $x \in X_i^{far}$, then we know from Lemma 26 that $O_u$ is $x$ with probability at least $\frac{\gamma}{|X_i'|}$ (note that $X_i'$ and $X_i$ have the same cardinality). Conditioned on this event, $Y_u$ will be equal to $O_u$ with probability $1/2$. Now suppose $x \in c(X_i^{near})$. Lemma 26 implies that $O_u$ is an element of $X_i^{far}$ with probability at least $\frac{\varepsilon}{8\alpha k}$. Conditioned on this event, $Y_u$ will be equal to $x$ with probability at least $\frac{1}{2} \cdot \frac{1}{|c(X_i^{near})|}$. Therefore, the probability that $O_u$ is equal to $x$ is at least

$$\frac{\varepsilon}{8\alpha k} \cdot \frac{1}{2|c(X_i^{near})|} \geq \frac{\varepsilon}{16\alpha k |X_i'|} \geq \frac{\gamma'}{|X_i'|}. \qquad\square$$

Corollary 1 shows that we can obtain samples from $X_i'$ which are nearly uniform (up to a constant factor). To convert this to a set of uniform samples, we use the idea of Jaiswal et al. (2014). For an element $x \in X_i'$, let $\gamma_x$ be such that $\frac{\gamma_x}{|X_i'|}$ denotes the probability that the random variable $Y_u$ is equal to $x$ (note that this is independent of $u$). Corollary 1 implies that $\gamma_x \geq \gamma'$. We define a new set of independent random variables $Z_1, \ldots, Z_\rho$. The random variable $Z_u$ will depend on $Y_u$ only. If $Y_u$ is $\perp$, $Z_u$ is also $\perp$. If $Y_u$ is equal to $x \in X_i'$, then $Z_u$ takes the value $x$ with probability $\frac{\gamma'}{\gamma_x}$, and $\perp$ with the remaining probability. We can now prove the key lemma.

**Lemma 27.** *Let $\rho$ be $\frac{256}{\gamma' \cdot \varepsilon}$, and $m''$ denote the mean of the non-null samples from $Z_1, \ldots, Z_\rho$. Then, with a probability at least $(3/4)$, $\Phi(m'', X_i') \leq (1 + \frac{\varepsilon}{16}) \cdot \Delta(X_i')$.*

*Proof.* Note that a random variable $Z_u$ is equal to a specific element of $X_i'$ with probability equal to $\frac{\gamma'}{|X_i'|}$. Therefore, it takes $\perp$ value with probability $1 - \gamma'$. Now consider a different set of iid random variables $Z_u'$, $1 \leq u \leq \rho$ as follows: each $Z_u$ tosses a coin with a probability of Heads being $\gamma'$. If we get Tails, it gets value $\perp$ otherwise, it is equal to a random element of $X_i'$. It is easy to check that the joint distribution of the random variables $Z_u'$ is identical to that of the random variables $Z_u$. Thus, it suffices to prove the statement of the lemma for the random variables $Z_u'$.

Now we condition on the coin tosses of the random variables $Z_u'$. Let $n'$ be the number of random variables which are not $\perp$. ($n'$ is a deterministic quantity because we have conditioned on the coin tosses). Let $m''$ be the mean of such non-$\perp$ variables among $Z_1', \ldots, Z_\rho'$. If $n'$ happens to be larger than $\frac{128}{\varepsilon}$, Lemma 17 implies that with probability at least $(7/8)$, $\Phi(m'', X_i') \leq (1 + \frac{\varepsilon}{16}) \cdot \Delta(X_i')$.

Finally, observe that the expected number of non-$\perp$ random variables is $\gamma' \cdot \rho \geq \frac{256}{\varepsilon}$. Therefore, with probability at least $\frac{7}{8}$ (using Chernoff-Hoeffding), the number of non-$\perp$ elements will be at least $\frac{128}{\varepsilon}$. $\qquad\square$

Let $C^{(\rho)}$ denote the multi-set obtained by taking $\rho$ copies of each of the centers in $C_{init}$. Now observe that all the non-$\perp$ elements among $Y_1, \ldots, Y_\rho$ are elements of $\{O_1, \ldots, O_\rho\} \cup C^{(\rho)}$, and so the same must hold for $Z_1, \ldots, Z_\rho$. Moreover, since we only need a uniform subset of size $\tau = \frac{128}{\varepsilon}$, $C_i'$ suffices instead of $C^{(\rho)}$. This implies that in steps 8-9 of the algorithm, we would have tried adding the point $m''$ as described in Lemma 27. This means that $M_i$ contains a subset $T_i$ such that $\Phi(\mu(T_i), X_i) \le (1 + \frac{\varepsilon}{2}) \cdot \Delta(X_i)$ with probability at least $3/4$. This concludes the proof of Theorem 10.

## F  ADDITIONAL EXPERIMENTS

In this section, we give additional experiments and some of the details of the experiments in Section 4.

### F.1  LARGER DATASETS AND AFKMC$^2$

We have added experiments on datasets significantly larger than the ones experimented on in Section 4 to further elaborate on the dependence of $\zeta, k, d$ and $N$ on QI-k-means++. The conditions under which the experiments were performed are the same as before. We also compare with the AFKMC$^2$ algorithm from Bachem et al. (2016a). In our implementation, we used the Markov chain length to be $m = 200$ as suggested by Bachem et al. (2016a) for all datasets.

| Dataset | $N$ | $d$ | QI-k-means++ Setup Time (s) | AFKMC$^2$ Setup Time (s) |
|---|---|---|---|---|
| KDD Caruana et al. (2004) | $145,751$ | 74 | 1.1 | 0.2 |
| SUSY Whiteson (2014) | $5,000,000$ | 18 | 11.2 | 2.5 |

Table 4: Details of the datasets experimented on and corresponding pre-processing times

We first compare the runtime for sampling (without taking into account the pre-processing step) for a range of values of $k$. The speedups are given in Table 5. For a better comparison, we also provide plots for the *cumulative runtime* (done for 9 values of $k \in \{2, .., 10\}$), which also includes the pre-processing cost.

| Dataset | Algorithm | $k = 5$ | $k = 10$ | $k = 50$ | $k = 100$ | $k = 200$ |
|---|---|---|---|---|---|---|
| KDD | QI-k-means++ | $1.9 \times 10^{-4}$ | $7.6 \times 10^{-4}$ | $1.9 \times 10^{-2}$ | $7.8 \times 10^{-2}$ | $3.2 \times 10^{-1}$ |
| KDD | k-means++ | $4.6 \times 10^{-1}$ | $9.3 \times 10^{-1}$ | 4.7 | 9.6 | 19.4 |
| KDD | AFKMC$^2$ | $3.2 \times 10^{-3}$ | $1.2 \times 10^{-2}$ | $2.7 \times 10^{-1}$ | 1.1 | 4.3 |
| SUSY | QI-kmeans++ | $1.8 \times 10^{-4}$ | $5.6 \times 10^{-4}$ | $1.6 \times 10^{-2}$ | $8.7 \times 10^{-2}$ | $4.3 \times 10^{-1}$ |
| SUSY | k-means++ | 5.0 | 9.7 | 48 | 95 | 195 |
| SUSY | AFKMC$^2$ | $2.4 \times 10^{-3}$ | $6.5 \times 10^{-3}$ | $8.8 \times 10^{-2}$ | $3.2 \times 10^{-1}$ | 1.2 |

Table 5: Runtime dependence (in seconds) for performing a single clustering on $k$

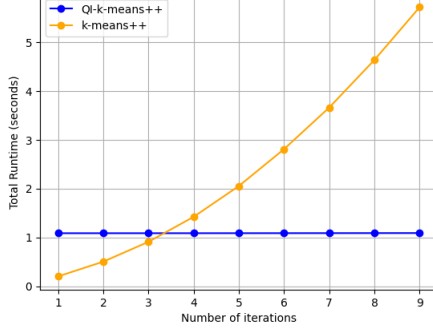

Figure 4: Cumulative runtime plot for KDD

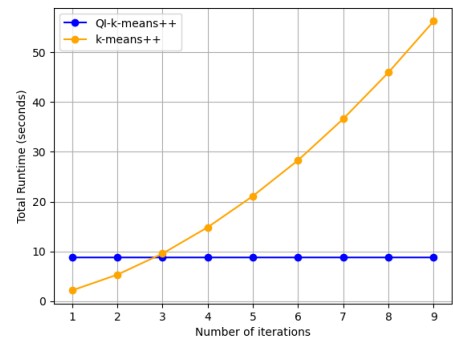

Figure 5: Cumulative runtime plot for SUSY

**Large Datasets :** For large datasets, the total runtime is dominated by the pre-processing cost. After the pre-processing, sampling the cluster centers becomes extremely fast and is several orders of magnitude faster than `k-means++`. This can also be seen by the cumulative runtime plots in Figures 4 and 5 - the plot for `QI-k-means++` is almost constant. This shows that `QI-k-means++` scales well for large datasets since the gain from logarithmic dependence on $N$ outweighs the poor dependence on $\zeta, k$, and $d$. As for the dependence on $k$, we can see from Table 5 that the quadratic dependence is immediately visible, but the runtime is still significantly less than that of `k-means++` even for intermediate values of $k$. Hence, such datasets lie in the advantageous regime of `QI-k-means++`. This contrasts small datasets with larger aspect ratios, as elucidated in Section 4.

**Comparison with `AFKMC`$^2$ :** The `AFKMC`$^2$ algorithm also consists of a pre-processing step which takes $O(Nd)$ time, and a main sampling step, which takes $O(mk^2d)$ number of samples from the initial probability distribution, where $m$ is the *markov chain length*. We note two major tradeoffs between `AFKMC`$^2$ and `QI-k-means++`. The first is that the sampling step itself is empirically faster for `QI-k-means++`, while the pre-processing step is more costly for `QI-k-means++` since it requires the setup of *sample and query access* data structures. But the *SQ* data structure allows the modification/addition of data points in $O(\log Nd)$ time without repeating the pre-processing step. In contrast, `AFKMC`$^2$'s pre-processing step calculates the initial probability distribution for the Markov chain, which requires calculating the distance of each point from an initial randomly chosen point. Hence, updating or adding a data point will require repeating the pre-processing step.

## F.2 Additional Details

We report the variance of the clustering costs mentioned in Section 4 (recall that the experiments were conducted over five runs). The seeds were chosen randomly for each run.

| k | 2 | 3 | 4 | 5 | 6 | 7 | 8 | 9 | 10 |
|---|---|---|---|---|---|---|---|---|---|
| QI-k-means++ | 1.89 | 3.19 | 0.97 | 1.36 | 1.44 | 1.24 | 0.17 | 0.20 | 0.47 |
| k-means++ | 1.93 | 3.29 | 1.90 | 1.44 | 1.57 | 1.09 | 0.73 | 0.32 | 0.52 |

Table 6: Variance of Clustering cost for binarized MNIST (values are scaled down by a factor of $10^{11}$)

| k | 2 | 3 | 4 | 5 | 6 | 7 | 8 | 9 | 10 |
|---|---|---|---|---|---|---|---|---|---|
| QI-k-means++ | 246523 | 48805 | 335 | 830 | 236 | 99 | 72 | 55 | 51 |
| k-means++ | 423718 | 27544 | 420 | 561 | 765 | 54 | 93 | 35 | 47 |

Table 7: Variance of Clustering cost for IRIS (values are rounded to 2 decimal places)

| k | 2 | 3 | 4 | 5 | 6 | 7 | 8 | 9 | 10 |
|---|---|---|---|---|---|---|---|---|---|
| QI-k-means++ | 3.77 | 4.05 | 1.44 | 5.96 | 1.10 | 2.37 | 1.10 | 0.57 | 1.07 |
| k-means++ | 1.91 | 2.00 | 2.24 | 3.92 | 2.28 | 0.38 | 1.34 | 1.51 | 1.06 |

Table 8: Variance of Clustering cost for DIGITS (values are scaled down by a factor of $10^{10}$)

