# OpenReview forum: "Quantum (Inspired)  $D^2$-sampling with Applications"
_ICLR.cc/2025/Conference — ICLR 2025 Poster_

### Official Review · Reviewer_ZoT4 · 2024-10-18

**Soundness:** 3
**Presentation:** 4
**Contribution:** 4
**Rating:** 8
**Confidence:** 3

**Summary:**

The authors give novel quantum and classical implementations of $k$-means++.
The main ingredient of their approach is a novel $D^2$-sampling method.

**Strengths:**

$k$-means++ is an important tool in ML, therefore this paper is significant as it gives novel quantum or classical implementations of it.

**Weaknesses:**

See the questions below.

**Questions:**

Page 1:
I find the abstract to have too much low-level information :)

Page 4:
Can you please expand on related work?
How about the connections of your work to computational complexity?

Page 5:
Please give some extra background knowledge in quantum computing and Dirac bra-ket notation :)

Page 6:
Can you please elaborate on the concept of dequantization and provide some examples?

Page 7:
I do not see why the states $(1),\ \dots,\ (4)$ are the right ones involved in the quantum $D^2$-sampling algorithm :)

Page 8:
I do not understand the Lines 412 -- 417.
What does pseudo approximate mean here?

Page 9:
I am not an expert with experiments :)

Page 10:
Where is the Conclusion? :) Can you please add a Conclusion and Future Work section?

---

> ### Author Response · Authors · 2024-11-15
> **We thank you for your positive comments on our work and appreciate your valuable and  constructive feedback**
>
> We address the issues pointed out by the reviewer.
>
> >Page 1: I find the abstract to have too much low-level information :)
>
> Thank you for the suggestion. We will try to make the abstract more concise in the revised version.
>
> >Page 4: Can you please expand on related work? How about the connections of your work to computational complexity?
>
> We discuss previous work done in relation to this problem  in the introduction section of the submission. This discussion includes previous work done on quantum algorithms for k means as well as recent implementations of kmeans++ with improved runtime  .We would also be including a table of comparisons (please see below) among various algorithms to serve as a summary of related work done on this problem. It would also make it simpler to see where our contributions lie in the algorithmic landscape. We would be happy to discuss about any specific work the reviewer would like to discuss about.
>
> >Page 5: Please give some extra background knowledge in quantum computing and Dirac bra-ket notation :)
>
> Thanks for pointing this out. We will make sure to include a brief section in the appendix which would contains pointers to literature covering the required background knowledge. We believe that this is a good suggestion as additional background will make our work more accessible to a broader audience at ICLR.
>
>
> >Page 6: Can you please elaborate on the concept of dequantization and provide some examples?
>
> Dequantization refers to a technique recently developed to compare quantum and classical machine learning algorithms on a more level ground starting from the breakthrough work of Tang on recommendation systems [1]. The main idea is to efficiently simulate a quantum algorithm assuming that one has both sampling and query access to the data being provided as input (which can be setup in linear time) , analogous to how data is assumed to be provided in a QRAM data structure to QML algorithms [2] . The PhD thesis [3] nicely classifies QML algorithms into those which have a speedup only due their input assumptions and those for which such results are known (or not expected to exist). We show that $D^2$ sampling belongs to the former class and hence illustrate efficient classical versions of our quantum algorithms.
>
>
>
> >Page 7: I do not see why the states
>  are the right ones involved in the quantum
> -sampling algorithm :)
>
> The states $(1),\dots,(4)$ represent a high level overview of our quantum algorithm for $D^2$ sampling.  In particular , state (4) is as follows :
>
> $$ | \Phi_{sample}\rangle = \frac{1}{\sqrt{N}} \sum_{i \in [N]} |i\rangle (\beta_i | 0 \rangle  + \sqrt{1 - |\beta_i|^2} |0\rangle ) $$
>
> Here $\beta_i = \frac{1}{Z}\min_{j \in [m]} D(v_i, c_j)$. Note that if we were to measure this state, the probability of getting the outcome as $(i,0)$ would be proportional to $\min_{j \in [m]} D^2(v_i, c_j)$, which is what we want -  the $D^2$ distribution itself. This means that if we can prepare this state efficiently, then we can also perform $D^2$ sampling efficiently.
>
>
> >Page 8: I do not understand the Lines 412 -- 417. What does pseudo approximate mean here?
>
> In the context of the k means problem , an $(\alpha, \beta)$ pseudo approximate solution means that instead of outputing a set of $k$ centers, we are allowed to output a set of $\beta k$ centers for some $\beta > 1$ such that the clustering cost using these $\beta k$ centers is atmost $\alpha$ times the optimal $k$ means cost.
>
>
> >Page 9: I am not an expert with experiments :)
>
> We would be more than happy to clarify any specific comments the reviewer might have on any of the experiments .
>
> >Page 10: Where is the Conclusion? :) Can you please add a Conclusion and Future Work section?
>
> Thanks again for the suggestion. We have discussed the pros and cons of QIkmeans++ in the experiments section. We will also  try to  add a separate conclusion and future work section in the revised edition which would also include the discussions done during this period. The major open problem from our work is to see whether we can improve the dependence on the aspect ratio while still maintaining logarithmic dependence on the size of the dataset.
>
>
> We would be happy to address any other concerns , clarifications or comments the reviewer may have.

---

> ### Author Response · Authors · 2024-11-15
>
> | Algorithm | Runtime | Approximation Guarantee | Key Parameters| Reference |
> |----------|----------|----------|------------------------------|--|
> |     $\text{QI-K\small{MEANS}++}$      | $O(Nd) + O(\zeta^2k^2 d \log k \log Nd)$     | $\mathbb{E}[\Phi] \leq O(\log k) \Phi^k_{OPT}$ | $\zeta$ is the aspect ratio i.e, the ratio of the maximum to minimum interpoint distance| This Work
> |  $\text{K-\small{MC}}^2$     |  $ O(\gamma k^2 d \log (k \beta) )$  , $ O(k^3 d \log^2 N \log k)$  | $\mathbb{E}[\Phi] \leq O(\log k) \Phi^k_{OPT}$ |  $\gamma = 4 N\frac{\max_{x \in V} D^2(x,\mu)}{\sum_{x\in V}D^2(x,\mu)} \frac{\Phi^1_{OPT}}{\Phi^k_{OPT}}$  $\beta = \frac{\Phi^1_{OPT}}{\Phi^k_{OPT}}$  Under certain assumptions, $\gamma = O(k\log^2 N)$ and $\beta = O(k)$| [1]
> |  $\text{AFK-\small{MC}}^2$   |  $O(Nd) + O\left(\frac{1}{\varepsilon}k^2 d \log \frac{k}{\varepsilon}\right)$ ,$O(Nd) + O(k^3 d \log k)$ | $\mathbb{E}[\Phi] \leq O(\log k) \Phi^k_{OPT} + 2\varepsilon \Phi^1_{OPT}$| $\varepsilon \in (0,1)$ is an error parameter Under certain assumptions $\varepsilon = \Omega(1/k)$ gives $O(\log k)$ approximation | [2]
> |  $\text{R\small{EJECTION}S\small{AMPLING}}$   | $O(Nd \log (\zeta d) +  N \log N\log (\zeta d)) + O\left(kc^2 d^3 \log \zeta (N \log \zeta)^{O(1/c^2)} \right)$  | $\mathbb{E}[\Phi] \leq O(c^6\log k) \Phi^k_{OPT}$ |$\zeta$ is the aspect ratio i.e, the ratio of the maximum to minimum interpoint distance   $c > 1$ is a parameter which controls the runtime - approximation quality tradeoff. | [3]
>
> In the above table , $\Phi$ represents the cost of the centers returned by the algoritm,  $\Phi^k_{OPT}$ represents the optimal $k$ means cost. $\mu$ is the mean of the dataset.  Since we are discussing $k$ means++ like seeding, all the cluster centers are from the dataset itself.
>
>
> > References
> >
> > [1] Olivier Bachem, Mario Lucic, S. Hamed Hassani, and Andreas Krause. Approximate k-means++
> in sublinear time. Proceedings of the AAAI Conference on Artificial Intelligence, 30(1), Feb.
> 2016b. doi: 10.1609/aaai.v30i1.10259. URL https://ojs.aaai.org/index.php/
> AAAI/article/view/10259.
> >
> > [2]Olivier Bachem, Mario Lucic, Hamed Hassani, and Andreas Krause. Fast and provably good
> seedings for k-means. In D. Lee, M. Sugiyama, U. Luxburg, I. Guyon, and R. Garnett
> (eds.), Advances in Neural Information Processing Systems, volume 29. Curran Associates, Inc.,
> 2016a. URL https://proceedings.neurips.cc/paper_files/paper/2016/
> file/d67d8ab4f4c10bf22aa353e27879133c-Paper.pdf.
> >
> > [3] Vincent Cohen-Addad, Silvio Lattanzi, Ashkan Norouzi-Fard, Christian Sohler, and Ola
> Svensson. Fast and accurate k-means++ via rejection sampling. In H. Larochelle,
> M. Ranzato, R. Hadsell, M.F. Balcan, and H. Lin (eds.), Advances in Neural Information Processing Systems, volume 33, pp. 16235–16245. Curran Associates, Inc 2020. URL https://proceedings.neurips.cc/paper_files/paper/2020/file/babcff88f8be8c4795bd6f0f8cccca61-Paper.pdf.

---

> ### Author Response · Authors · 2024-11-25
> **Request for Rebuttal Feedback - Discussion Period closing soon**
>
> Dear Reviewer ZoT4,
>
> Thank you for your valuable feedback on our submission. As the discussion phase closes soon, we would greatly appreciate your thoughts on our revision. Early feedback would allow us time to address any remaining concerns you may have.
>
> Thank you for your time and consideration.
>
> Best regards,
>
> Authors

---

> > ### Comment · Reviewer_ZoT4 · 2024-11-27
> >
> > Thank you very much! I will keep my positive score :)

---

> > > ### Author Response · Authors · 2024-11-27
> > >
> > > Dear Reviewer ZoT4,
> > >
> > > Thanks a lot for going through our response and for your positive evaluation of our submission.
> > >
> > > Best Regards,
> > >
> > > Authors.

---

> ### Comment · Area_Chair_xjbT · 2024-11-27
>
> Dear Reviewer,
>
> The authors have provided their rebuttal to your comments/questions. Given that we are not far from the end of author-reviewer discussions, it will be very helpful if you can take a look at their rebuttal and provide any further comments. Even if you do not have further comments, please also confirm that you have read the rebuttal. Thanks!
>
> Best wishes,
> AC

---

### Official Review · Reviewer_66Ux · 2024-10-29

**Soundness:** 3
**Presentation:** 3
**Contribution:** 3
**Rating:** 6
**Confidence:** 2

**Summary:**

The paper discusses a novel quantum-inspired algorithm for the k-means problem. Theoretical guarantees of the new algorithm are provided, showing it outperforms classical algorithms in certain areas. Experiments compare runtime of the quantum inspired k-means++ versus the classical k-means++.

**Strengths:**

Starting from originality, the paper extends ideas in quantum machine learning (primarily the sample-access-query model of Tang) to the classical k-means problem, providing novel algorithms and results. Connections to existing literature are well-discussed, and discussions of the strengths and weaknesses of the new algorithm compared to classical algorithms are provided.

In terms of quality and clarity, it is quite clear that effort has been made to polish the text. The problem formulations, motivations, related work, and the design of the algorithm from a quantum-inspired perspective are all clearly stated.

It is interesting to see that the new algorithm has better dependency in N but worse dependency in the aspect ratio and k. While this does not show a uniform improvement in all aspects, it is still a good contribution.

**Weaknesses:**

One of the downsides of the algorithm's runtime dependencies is the dependence of the preprocessing cost O(Nd), which is ignored when comparing against existing results. If we add in O(Nd) to the equation, then both the new algorithm and classical algorithms has a linear dependence on $N$. Perhaps it would be meaningful to have experiments that showcase runtime *after preprocessing* so the plots in Figures 2-3 can be more reflective of the runtime dependencies after preprocessing.

**Questions:**

There are some (minor) questions about the results:

1. In page 4, there is discussion about dependencies in the *assumption-free* case but seems like there are no results for the cases with assumptions. What comparisons and improvements can be made in scenarios with assumptions?

2. Considering different algorithms appear to have quite different dependencies with regards to N, d, k, aspect ratio, matrix conditions, etc., what are some impossibility results available for the k-means problem and how does the algorithm's runtime dependencies compare with it?

3. Can you elaborate on whether there are methods to improve the dependency on the aspect ratio $\zeta$? It seems to be a bit counterintuitive. For example, if there are two clusters of points, and the two clusters are very well-separated, intuition might suggest that k-means with $k=2$ should be relatively straightforward and should be easier as the two clusters are separated out even more. But, in this case $\zeta$ grows and the bound on the runtime worsens. Additionally, $\zeta$ becomes very large if there is even a single pair of points that are close by. Do these observations suggest that the dependencies on $\zeta$ can be tightened?

**Details Of Ethics Concerns:**

N/A.

---

> ### Author Response · Authors · 2024-11-15
> **We thank you for your positive comments on our work and appreciate your valuable and  constructive feedback**
>
> We discuss and address the issues pointed out in the review :
>
>
> > One of the downsides of the algorithm's runtime dependencies is the dependence of the preprocessing cost O(Nd), which is ignored when comparing against existing results. If we add in O(Nd) to the equation, then both the new algorithm and classical algorithms has a linear dependence on
> . Perhaps it would be meaningful to have experiments that showcase runtime after preprocessing so the plots in Figures 2-3 can be more reflective of the runtime dependencies after preprocessing.
>
>
> Our algorithm indeed has an $O(Nd)$ pre processing cost which is used to setup the SQ access data structures required by the algorithm. We point out that other fast implementations of k means++ like [3],[2]} also have an $O(Nd)$ pre-processing cost for setting up the Multi-Tree Embeddings and the initial distribution for Monte Carlo Markov Chain sampling respectively.
>
>  The SQ data structures setup in the pre-processing step also allows us to update / add data points in just $O(\log N)$ time so that we do not need to repeat the complete algorithm all over again.This is because we just need to traverse the height of the complete binary tree which represents the SQ data structure.
>
> We also mention that there exist implementations of the SQ data structure which can be set up in $O(\mathtt{nnz}(V))$ time, where $\mathtt{nnz}(V)$ is the number of non zero entries in the dataset $V$, which can be much lower than $O(Nd)$ for sparse datasets [4].
>
> Regarding the experiments, we do consider the pre-processing time for comparing the runtimes both with kmeans++ and AFK-MC2. The . We also compare the setup cost needed for both the algorithms in appendix F. Indeed, we observe that for datasets lying in the advatageous regime of QIkmeans++, the runtime after pre-processing is quite fast due to its logarithmic dependence on $N$ for performing multiple clusterings, and is almost constant (and this constant is precisely the pre-processing cost). To be more clear, the plots show the cumulative runtime for performing multiple clusterings i.e, $O(Nd) + \sum_{i \in [q]} \tilde{O}(\zeta^2 k_i^2 d)$
>
>
> > In page 4, there is discussion about dependencies in the assumption-free case but seems like there are no results for the cases with assumptions. What comparisons and improvements can be made in scenarios with assumptions?
>
> The results of [2] with and without assumptions are as follows ($\phi_{OPT}$ is the optimal k means cost and $\phi$ is the solution returned by their algorithm. For a dataset $V$, $\mathbb{V}[V]$ is the variance of the dataset):}
>
>   1.   Without any assumptions, the runtime is $O(Nd) + O(\frac{1}{\varepsilon}k^2d \log k)$ and the approximation guarantee is $\mathbb{E}[\phi] \leq 8(\log k + 2)\phi_{OPT} + \varepsilon \mathbb{V}[V]$.
>
>   2. Under certain assumptions, the runtime is $O(Nd) + O(k^3d \log k)$ and the approximation guarantee is $\mathbb{E}[\phi] \leq 8(\log k + 3)\phi_{OPT}$
>
>
> For the assumption free case, the approximation guarantee as an additive term which can be arbitrarily larger than the k means cost. For the same approximation guaarantee as k means++, $\varepsilon$ would have to scale as $\mathbb{V}[V] / \phi_{OPT}$. We see that the $\varepsilon^{-1}$ factor can be seen to be analogous to the factor of $\zeta^2$ in QI-kmeans++. For the case with certain assumptions, their algorithm's runtime is independent of any additional factors, but has a worse dependence on $k$ ($k^3$ vs $k^2$)
>
> Let us also clarify on the assumptions made by [2]. Their runtime ($O(Nd) + O(\beta(V)k^2d\log k)$) depends on a parameter which they denote as $\beta(V)$ defined as follows : $$\beta(V) = \frac{\mathbb{V}[V]}{\phi_{OPT}}$$ Note that estimating this factor involves solving the k means problem itself and hence it is not computationally efficient to compute. They then show that if the data comes from a distribution which "has exponentially decreasing tail bounds and is approximately uniform on a hypersphere" then $\beta(V)  = O(k)$

---

> > ### Author Response · Authors · 2024-11-15
> >
> > > Considering different algorithms appear to have quite different dependencies with regards to N, d, k, aspect ratio, matrix conditions, etc., what are some impossibility results available for the k-means problem and how does the algorithm's runtime dependencies compare with it?
> >
> > To the best of our knowledge, we summarize the relevant  hardness results known for the k means problem below :
> >
> >
> > The k means problem is NP hard for certain small approximation factors.
> >
> > 1. The k means problem is an NP hard optimization problem, even for k = 2 as shown in  [5],[6] . The problem remains NP hard even if the points are restricted to lie in a plane as shown in [11].
> >
> > 2. We also have hardness of approximation results. There exists a constant $\varepsilon$ such that approximating the k means problem to a factor of $1 + \varepsilon$ is NP hard . The work [7] shows that $\varepsilon \leq 0.36$. We would also like to point out that this remains true even when restricted to cases when the aspect ratio $\zeta$ is upper bounded by a small constant. Similar for different $L_p$ metrics can be found in [12].
> >
> > Due to the NP hardness results, we do not expect $\text{poly}(N,k,d)$ algorithms to exist for sufficiently low constant approximation factors.
> >
> > 3. [8] shows that if the Exponential Time Hypothesis (ETH) holds then there exists a constant $c > 1$
> > such that any $c$-approximation algorithm for the k-means problem cannot have running time better than $\text{poly}(N,d)2^{\Omega(\frac{k}{\text{polylog}(k)})}$.
> >
> >
> > A lot of research has also been put towards finding efficient FPTAS i.e, algorithms which have $(1 + \varepsilon)$ approximation guarantees considering some parameters like $k$ or $d$ to be a constant.  Examples include $O(2^{(k / \varepsilon)^{O(1)}}Nd)$[13] , $O(nkd  + d \text{poly}(k/\varepsilon) +  2^{\tilde{O}(k / \varepsilon)})$ [14] and many more .  We would like to point out that our result is the first one where the runtime has logarithmic dependence on $N$ .
> >
> >
> >
> > Focusing on the aspect ratio $\zeta$, we do not know of any impossibility results , except the one described in point 1 of this discussion. It seems that this quantity arises particularly in $D^2$ sampling. For example, the required length of the markov chain method used in [1],[2] also have dependence on similar "aspect ratio like quantities".
> >
> > Let us now consider matrix conditions like the condition number $\kappa$, which appears in the algorithms presented in [9] (note that this algorithm, q-means, is a quantum implementation of the k-means heuristic (also called lloyd's iterations) and hence does not have any theoretical guarantees). This parameter appears due to the use of certain quantum linear algebra techniques and recently, the work [10] removed the dependence of the condition number from this algorithm and hence we think that $\kappa$ is not an inherently necessary parameter. Unfortunately, analogous results for the aspect ration $\zeta$ are not known.

---

> ### Author Response · Authors · 2024-11-15
>
> > Can you elaborate on whether there are methods to improve the dependency on the aspect ratio
> ? It seems to be a bit counterintuitive. For example, if there are two clusters of points, and the two clusters are very well-separated, intuition might suggest that k-means with
>  should be relatively straightforward and should be easier as the two clusters are separated out even more. But, in this case
>  grows and the bound on the runtime worsens. Additionally,
>  becomes very large if there is even a single pair of points that are close by. Do these observations suggest that the dependencies on
>  can be tightened?
>
>
> That is a great observation. Interestingly, this example is also applicable to the previous work done on speeding up k means ++, including our present work. We first discuss this in context of the previous work done and then describe why our current techniques depend on the aspect ratio $\zeta$.
>
>
>
> 1. The algorithms presented in [1] and [2] also have dependence on a "aspect ratio like quantity" which they denote as $\beta$, which is related to the number of iterations that their algorithm requires to converge to the $D^2$ distribution. $\beta$ is essentially the ratio of the variance of the dataset to the optimal $k$ means cost. The example provided by the reviewer is precisely a case where this parameter can grow quite large. The authors of [1] provide evidence that under some assumptions on the data (for example it being generated from gaussian - like distributions) the probability that such a case arises is low and then go onto show that with high probability , $\beta = O(k)$.
>
> 2. The algorithm presented in [3] also depends on $\zeta$, but the reason for this is significantly different than ours. An essential part of [3] is to use multi-tree embeddings by hierarchically dividing the given data set into cubes, untill each point is assigned to a single cube. The case when two points are quite close to each other needs a larger number of iterations (although it scales as $\log \zeta$) for performing this embedding.
>
>
> We now explain why we require such dependence on $\zeta$. An essential part of our quantum algorithm for $D^2$ sampling requires us to estimate the clustering cost of the current centers. We prepare the following state :
>
> $$ |\Phi_C\rangle  = \frac{1}{\sqrt{N}} \sum_{i \in [N]} |i\rangle | \min_{j \in [t]} D(v_i,c_j) \rangle $$
>
> Essentially what this state allows us to do is to uniformly sample a point from the dataset (by measuring the first register) along with its distance to its nearest sample (which we would get from the second register). Given this sampling ability, we can use concentration inequalities to obtain a good enough estimate of the clustering cost. This is where the aspect ratio comes into our algorithm, since we need to bound how large the quantity $D(v_i, c_j)$ can be (in the worst case, it could be equal to maximum interpoint distance) . Hence we need sufficient number of samples to get the required error in estimating the clustering cost.
>
> Let us also clarify this in the context of the example provided by the reviewer. Suppose the two clusters are very imbalanced with one cluster containing very few points. In this case, unless we sample sufficiently many points, our estimate can be very bad, which will adversely impact the next $D^2$-sample.
>
>
> We would also like to point out that since the parameter $\zeta$ is efficiently computable, one can check beforehand how the performance of the algorithm should expected to be.
>
>
>
> We think that it might be possible to improve the  dependence on the aspect ratio . In particular we suspect that the dependence on $\zeta^2$ can be improved to a factor which is the ratio of the optimal 1 means cost to the optimal k means cost (this quantity can be better than the aspect ratio in some cases). We are currently trying to work out the details  . But we would like to emphasize that this is still an "aspect ratio like quantity" , and unfortunately we do not know how to get away with or significantly reduce the dependence on such quantities while still maintaing a runtime with logarithmic dependence on $N$. For example, the works [1],[2] also have similar reasons as to why the number of iterations depend on such parameters.  We think that understanding the tradeoff between $\zeta$ and $N$ is an interesting open problem.
>
>
> We would be happy to address any other concerns , clarifications or comments the reviewer may have.

---

> ### Author Response · Authors · 2024-11-15
>
> > References
> >
> > [1] Olivier Bachem, Mario Lucic, S. Hamed Hassani, and Andreas Krause. Approximate k-means++
> in sublinear time. Proceedings of the AAAI Conference on Artificial Intelligence, 30(1), Feb.
> 2016b. doi: 10.1609/aaai.v30i1.10259. URL https://ojs.aaai.org/index.php/
> AAAI/article/view/10259.
> >
> > [2]Olivier Bachem, Mario Lucic, Hamed Hassani, and Andreas Krause. Fast and provably good
> seedings for k-means. In D. Lee, M. Sugiyama, U. Luxburg, I. Guyon, and R. Garnett
> (eds.), Advances in Neural Information Processing Systems, volume 29. Curran Associates, Inc.,
> 2016a. URL https://proceedings.neurips.cc/paper_files/paper/2016/
> file/d67d8ab4f4c10bf22aa353e27879133c-Paper.pdf.
> >
> > [3] Vincent Cohen-Addad, Silvio Lattanzi, Ashkan Norouzi-Fard, Christian Sohler, and Ola
> Svensson. Fast and accurate k-means++ via rejection sampling. In H. Larochelle,
> M. Ranzato, R. Hadsell, M.F. Balcan, and H. Lin (eds.), Advances in Neural Information Processing Systems, volume 33, pp. 16235–16245. Curran Associates, Inc 2020. URL https://proceedings.neurips.cc/paper_files/paper/2020/file/babcff88f8be8c4795bd6f0f8cccca61-Paper.pdf.
> >
> > [4] Nai-Hui Chia, András Gilyén, Tongyang Li, Han-Hsuan Lin, Ewin Tang, and Chunhao Wang. Sampling-based sublinear low-rank matrix arithmetic framework for dequantizing quantum machine learning. In Proceedings of the 52nd Annual ACM SIGACT Symposium on Theory of
> Computing, STOC ’20. ACM, June 2020
> >
> > [5] S. Dasgupta. The hardness of k-means clustering. Technical report, University of California, San Diego, 2008.
> >
> > [6] Daniel Aloise, Amit Deshpande, Pierre Hansen, and Preyas Popat. NP-hardness of Euclidean sum-ofsquares clustering. Machine Learning, 75(2):245–248, 2009.
> >
> > [7] Pranjal Awasthi, Moses Charikar, Ravishankar Krishnaswamy, and Ali Kemal Sinop. The Hardness of Approximation of Euclidean k-Means. In 31st International Symposium on Computational Geometry (SoCG 2015). Leibniz International Proceedings in Informatics (LIPIcs), Volume 34, pp. 754-767, Schloss Dagstuhl – Leibniz-Zentrum für Informatik (2015)
> https://doi.org/10.4230/LIPIcs.SOCG.2015.754
> >
> > [8] Nir Ailon, Anup Bhattacharya, Ragesh Jaiswal, and Amit Kumar. Approximate Clustering with Same-Cluster Queries. In 9th Innovations in Theoretical Computer Science Conference (ITCS 2018). Leibniz International Proceedings in Informatics (LIPIcs), Volume 94, pp. 40:1-40:21, Schloss Dagstuhl – Leibniz-Zentrum für Informatik (2018)
> https://doi.org/10.4230/LIPIcs.ITCS.2018.40
> >
> > [9] Iordanis Kerenidis, Jonas Landman, Alessandro Luongo, and Anupam Prakash. 2019. Q-means: a quantum algorithm for unsupervised machine learning. Proceedings of the 33rd International Conference on Neural Information Processing Systems. Curran Associates Inc., Red Hook, NY, USA, Article 372, 4134–4144.
> >
> > [10] Doriguello, J. F., Luongo, A., & Tang, E. (2023). Do you know what q-means? arXiv. https://arxiv.org/abs/2308.09701
> >
> > [11] Meena Mahajan, Prajakta Nimbhorkar, and Kasturi Varadarajan. 2012. The planar k-means problem is NP-hard. Theor. Comput. Sci. 442 (July, 2012), 13–21. https://doi.org/10.1016/j.tcs.2010.05.034
> >
> > [12] Vincent Cohen-Addad, Karthik Srikanta. Inapproximability of Clustering in Lp-metrics. FOCS'19 - 60th Annual IEEE Symposium on Foundations of Computer Science, Nov 2019, Baltimore, United States. ⟨hal-02360762v2⟩
> >
> >[13] A. Kumar, Y. Sabharwal, and S. Sen, A simple linear time (1 + )-approximation algorithmfor K-means clustering in any dimensions, in Proceedings of the 45th Annual IEEE Sym-posium on Foundations of Computer Science (FOCS ’04), IEEE Computer Society, LosAlamitos, CA, 2004, pp. 454–462.
> >
> >[14]  D. Feldman, M. Monemizadeh, and C. Sohler, A PTAS for K-means clustering based onweak coresets, in Proceedings of the Twenty-third Annual Symposium on ComputationalGeometry (SoCG ’07), SoCG ’07, ACM, New York, 2007, pp. 11–18

---

> ### Author Response · Authors · 2024-11-25
> **Request for Rebuttal Feedback - Discussion Period closing soon**
>
> Dear Reviewer 66Ux,
>
> Thank you for your valuable feedback on our submission. As the discussion phase closes soon, we would greatly appreciate your thoughts on our revision. Early feedback would allow us time to address any remaining concerns you may have.
>
> Thank you for your time and consideration.
>
> Best regards,
>
> Authors

---

> ### Comment · Area_Chair_xjbT · 2024-11-27
>
> Dear Reviewer,
>
> The authors have provided their rebuttal to your comments/questions. Given that we are not far from the end of author-reviewer discussions, it will be very helpful if you can take a look at their rebuttal and provide any further comments. Even if you do not have further comments, please also confirm that you have read the rebuttal. Thanks!
>
> Best wishes,
> AC

---

> > ### Comment · Reviewer_66Ux · 2024-11-27
> > **Thank you for responses**
> >
> > I would like to thank the authors for responding to my concerns. I have no further comments and am keeping my score considering some of the concerns regarding the aspect ratio appears to be difficult to resolve, and other reviewers have also raised similar questions.

---

> > > ### Author Response · Authors · 2024-11-28
> > >
> > > Dear Reviewer 66Ux,
> > >
> > > Thanks a lot for going through our response as well as for the constructive feedback and positive evaluation of our submission. We also think that studying the tradeoff between the data size and dependence on aspect ratio like quantities is an interesting future research direction, since such such tradeoffs appear in many algorithms pertaining to speeding up k-means++.
> > >
> > > Best Regards,
> > >
> > > Authors.

---

### Official Review · Reviewer_D44y · 2024-11-02

**Soundness:** 4
**Presentation:** 3
**Contribution:** 3
**Rating:** 6
**Confidence:** 3

**Summary:**

The present paper gives new algorithms; both quantum and classical, for  the $k$-means problem. $k$-means problem is a significant clustering problem with broad spectrum of applications. Given a set $V$ of points in ${\mathbf R}^{d}$, find a set $C$ of $k$ points, called centers, in $\mathbf{R}^d$, so that  the sum of squared distance from points in $V$ to the closest point in $C$ is minimized. Because of its foundational algorithmic nature, this computational problem has been of attention from researchers from several perspectives.

The present paper investigates new quantum approaches to solving this problem. Given it is NP-hard, the goal is to develop approximation algorithms and an algorithm known as $k$-means++ (extending the well known $k$-means algorithm by Lloyd) by Arthur and Vassilvitskii has a $O(\log k)$ approximation guarantee in expectation. A basic ingredient of $k$-means++ algorithm is called $D^2$-$\mathit{sampling}$: a procedure that given a data set $V$ and a center set $C$, samples a point in $V$ with probability proportional to its squared distance from the nearest center in $C$.

The main contribution of this paper is a new quantum algorithm for $D^2$-sampling and a subsequent de-quantization of it. This algorithm then leads to new implementation of  (both quantum and classical) of $k$-means++. While there are other known quantum implementation of  $D^2$-sampling, the present algorithm appears to have different efficiency parameters and hence appears to perform better in certain regimes than known ones. The main parameter that the  running time of the algorithm depends on is the ${\mathit aspect ratio}$ $\zeta$: the ratio between the largest and smallest inter-point distance in the dataset. In particular, the quantum algorithm that the author/s develop runs in time $\tilde{O}(\zeta^2k^2)$ with $\tilde{O}$ hiding the poly-log dependencies on relevant parameters. The author/s show how this can be de-quantized to get a classical algorithm with running time $O(Nd)+\tilde{O}(\zeta^2k^2d)$.   The paper also presents some initial experimental results.

**Strengths:**

The paper designs new algorithms for a classical and significant computational problem: the  $k$-means problem. Any new algorithm for classical problems such as $k$-means is a contribution to the field of ML and data science.   Thus the paper has broad applicability (in particular will be of interest to ICLR audience).  This is the main strength of the paper.

**Weaknesses:**

The main weakness is  the algorithm's dependency on the aspect ratio. The algorithm has quadratic dependency on the aspect ratio. That makes one wonder whether it will be really  useful for broad applications. While the author/s briefly discuss this dependency in the paper, it is not at all clear why there needs to be a quadratic dependency. Is it possible to prove a lower bound (in a meaningful restricted model) showing that this dependency is needed? The authors cite a paper by Cohen-Addad et al that presents an algorithm that has closer to $\log$ dependency on $\zeta$ (at the expense of having a $N^{\theta(1)}$ factor in the running time) which appears to be much more attractive. The paper will be substantially stronger if it can bring more clarity on this issue. Following are concrete points that the authors may address to make the paper stronger.

1. Provide a more detailed justification for why the algorithm presented has quadratic dependence on the aspect ratio.
2. Discuss whether the author/s  believe this dependence is fundamental or if there may be ways to improve it.
3. Consider proving a lower bound, to show whether this dependence is necessary (if the authors believe it is).
4. Compare their aspect ratio dependence more thoroughly to other algorithms like the Cohen-Addad et al. result, analyzing the tradeoffs in more depth.

**Questions:**

The paper is nicely written, overall. However, the discussion through the prior work is not very clearly written. May be a table of relevant known results will be useful to understand the context of the algorithm in terms of efficiency. Also a more detailed discussion on aspect ratio and its relevance (and if there are fundamental obstacles in improving the running time in terms of this parameter) will be very useful. A table of relevant known results might include:

1. Key prior quantum and classical algorithms for k-means/k-means++
2. Their runtime complexities
3. Their approximation guarantees
4. Key parameters they depend on (e.g. aspect ratio)

---

> ### Author Response · Authors · 2024-11-15
> **We thank you for your positive comments on our work and appreciate your valuable and  constructive feedback**
>
> We discuss and address the issues pointed out by the reviewer :
>
> > Provide a more detailed justification for why the algorithm presented has quadratic dependence on the aspect ratio.
>
> Our algorithms (both quantum and classical) have a quadratic dependence on the aspect ratio $\zeta$ of the dataset. Below, we give a simplified description of our techniques for performing $D^2$ sampling and point out exactly where the dependence on $\zeta$ arises.
>
> #### The Quantum Case
>
> Essentially, the quantum algorithm performs $D^2$-sampling efficiently by preparing a quantum state such that measuring that state would lead to generating samples from (close to) the $D^2$ distribution. Let the dataset be $V$ and the points in $V$ be $v_1,\dots,v_n$ and let $C$ be the set of cluster centers with points $c_1,\dots,c_t$
>
> For brevity, let us assume that we can efficiently perform the operation $|i\rangle|j\rangle|0\rangle \rightarrow |i\rangle |j \rangle |D(v_i, c_j)\rangle $.  A more thorough discussion regarding this can be found in the introduction of section B.
>
> By using these operations, we can efficiently prepare the following state by iteratively applying controlled SWAP gates on the second register:
>
> $$ |\Phi_C\rangle  = \frac{1}{\sqrt{N}} \sum_{i \in [N]} |i\rangle | \min_{j \in [t]} D(v_i,c_j) \rangle $$
>
> Now we transform this state into the following state $|\Phi_{sample} \rangle$ where the term in the second register of $|\Phi_C\rangle $ acts as the amplitude (denoted by $ \beta_i = \frac{\min_{j \in [t]} D(v_i,c_j)}{\sqrt{2\Phi(V,C)}}$) by performing controlled rotations. We can see that we can now sample from the $D^2$ distribution by measuring $|\Phi_{sample}\rangle$.
>
> $$ | \Phi_{sample}\rangle = \frac{1}{\sqrt{N}} \sum_{i \in [N]} |i\rangle (\beta_i | 0 \rangle  + \sqrt{1 - |\beta_i|^2} |0\rangle ) $$
>
> But to perform the transformation $|\Phi_C\rangle \rightarrow |\Phi_{sample}\rangle$ we need to know the clustering cost $\Phi(V,C)$ to be able to apply the necessary controlled rotations. To be more specific, a rotation gate $R_y(\theta)$ can perform the transformation $|0\rangle \rightarrow \cos \frac{\theta}{2} |0 \rangle - \sin \frac{\theta}{2} |1\rangle $. We can use a controlled rotation gate (with the control on the second register of $|\Phi_C\rangle $) to calculate the required rotation angle to be applied, which needs $\Phi(V,C)$ to compute.
>
> It is in computing an approximation to $\Phi(V,C)$ that we get the factor of $\zeta^2$. To estimate $\Phi(V,C)$ , we measure $m$ copies of the state $|\Phi_C \rangle $ and compute the average of the squares of the values measured in the second register (note that these are bounded above by $\zeta^2$). We then use hoeffding's concentration inequality to get an upper bound on the number of samples required, which scales as $\zeta^2$.
>
>
> #### The Classical Case
>
> The classical dequantized algorithm works similarly. We want to get sample and query access to a vector which represents the $D^2$ distribution . We do this in the following two compositional steps :
>
> 1. Given SQ access to the dataset $SQ(V)$ and the cluster centers $SQ(c_1),\dots , SQ(c_t)$, we setup SQ access to vectors $u_1,\cdots , u_t$ defined as $u_j(i) = D(v_i, c_j)$.
>
> 2. Given SQ access to the vectors $SQ(u_1), \dots , SQ(u_t)$, we setup SQ access to a vector $w$ defined as $w(i) = \min_{j \in [t]} u_j(i)$
>
> Note that "setting up" SQ access for a desired vector means providing an algorithm that given SQ access to the inputs allows us to generate samples from the distribution represented by that vector . Note that it is not directly clear how to setup $SQ(w)$ given $SQ(u_1),\dots , SQ(u_t)$. The idea for overcoming this barrier is to  setup  SQ access to another vector $\tilde{w}$ whose SQ access is simple to construct and which acts as an appropriate upper bound on the original vector $w$. Then samples from $SQ(\tilde{w})$ can be converted to samples from $SQ(w)$ by using rejection sampling. The key step then is to design such a vector $\tilde{w}$. We prove in Lemma 10 that the following choice satisfies all the requirements :
>
> $$ \tilde{w}(i) = \sqrt{\frac{1}{t} \sum_{j \in [t]} |u_j(i)|^2} $$
>
>
> It is in converting $SQ$ access to $\tilde{w}$ to SQ access to $w$ that we get a factor of $\zeta^2$. The rejection sampling step incurrs an overhead of $\phi = \| \tilde{w}\|^2 / \| w\|^2$ in the runtime. We are able to show an upper bound of $\phi \leq 8\zeta^2$ in the proof of Theorem 2.

---

> > ### Author Response · Authors · 2024-11-15
> >
> > > Discuss whether the author/s believe this dependence is fundamental or if there may be ways to improve it.
> > Consider proving a lower bound, to show whether this dependence is necessary (if the authors believe it is).
> >
> >
> > We think that it might be possible to improve the  dependence on the aspect ratio . In particular we suspect that the dependence on $\zeta^2$ can be improved to a factor which is the ratio of the optimal 1 means cost to the optimal k means cost (this quantity can be better than the aspect ratio in some cases). We are currently trying to work out the details  . But we would like to emphasize that this is still an "aspect ratio like quantity" , and unfortunately we do not know how to get away with or significantly reduce the dependence on such quantities while still maintaing a runtime with logarithmic dependence on $N$. For example , the runtimes in [1] and [2] also end up depending on similar  "aspect ratio like quantities".  We think that understanding the tradeoff between $\zeta$ and $N$ is an interesting open problem. Our algorithm and the one presented in [3] seem to be lying on two extremes : QIkmeans++ has polynomial dependence on $\zeta$ but logarithmic dependence on $N$ as compared to the polylogarithmic dependence on $\zeta$ but polynomial dependence on $N$ in [3]. It would be interesting to see if there is a class of algorithms with a controllable tradeoff between $\zeta$ and $N$.
> >
> >
> >
> > > Compare their aspect ratio dependence more thoroughly to other algorithms like the Cohen-Addad et al. result, analyzing the tradeoffs in more depth.
> >
> > We analyze the tradeoffs with the algorithm preasented in [3] as follows :
> >
> > 1. Pre-processing : [3] uses $O(N(d+\log N) \log \zeta d)$ preprocessing time to setup multi-tree data structures while QIKMEANS++ uses $O(Nd)$ time to setup the SQ data structure. We would like to point out that setting up the SQ data structure has the added advantage of supporting addition / updation of data points in $O(\log Nd)$ time. The logarithmic dependence on $\zeta$ in [3] arises due to the multi-tree data structures being constructed by hierarchically dividing the space into cubes.  Hence, the dependence on $\zeta$ id due to significantly diffrent reasons for both.
> >
> > 2. Runtime : Let us compare the runtime for performing a single seeding. [3] requires $O\left(kc^2 d^3 \log \zeta (N \log \zeta)^{O(1/c^2)} \right)$ time as compared to $O(\zeta^2 k^2 d \log k \log Nd)$ of QIKMEANS++. Here, $c > 1$ is a parameter which appears due to using techniques such as locality sensitive hashing to find the  approximate closest center to a point. This improves the dependence on $k$ ($k$ vs $k^2$) at the cost of a weaker approximation guarantee. There is also an increased dependence on $d$ ($d^3$ vs $d$) in  REJECTIONSAMPLING due to their use of multi-tree embeddings (the distances of which are distorted by a factor of $d$ in expectation). In the case of dimension, we would like to point out that known dimensionality reduction techniques (for example the JL transformation) can applied to both the algorithms. The dependence of QIKMEANS++ on $N$ is logarithmic as compared to the small polynomial dependence in [3]. We can think of the performance of these algorithms as being lying in complementary ranges. For the case of large $N$ and small $k$ with bounded aspect ratio, one could expect QIKMEANS++ to perform better, while for the case of moderate $N$ and large $k$, [3] seems better.
> >
> > 3. Approximation Guarantee : Our algorithm has the same approximation guarantee as that of k means++ which is of $O(\log k)$ , while [3] has an approximation guarantee of $O(c^6 \log k)$. This can be seen as a runtime - quality tradeoff which is absent in our work.

---

> ### Author Response · Authors · 2024-11-15
>
> > However, the discussion through the prior work is not very clearly written. May be a table of relevant known results will be useful to understand the context of the algorithm in terms of efficiency. Also a more detailed discussion on aspect ratio and its relevance (and if there are fundamental obstacles in improving the running time in terms of this parameter) will be very useful.
>
> Thanks for  pointing this out. We will be including the following table which summarizes previous work on improving the run time of kmeans++. We understand that the algorithmic landscape for this problem is such that there is no single best algorithm in all scenarios and a concise summary would be quite helpful for the audience.
>
>
>
>
> | Algorithm | Runtime | Approximation Guarantee | Key Parameters| Reference |
> |----------|----------|----------|------------------------------|--|
> |     $\text{QI-K\small{MEANS}++}$      | $O(Nd) + O(\zeta^2k^2 d \log k \log Nd)$     | $\mathbb{E}[\Phi] \leq O(\log k) \Phi^k_{OPT}$ | $\zeta$ is the aspect ratio i.e, the ratio of the maximum to minimum interpoint distance| This Work
> |  $\text{K-\small{MC}}^2$     |  $O(\gamma k^2 d \log (k \beta) )$  ,$ O(k^3 d \log^2 N \log k)$  | $\mathbb{E}[\Phi] \leq O(\log k) \Phi^k_{OPT}$ |  $\gamma = 4 N\frac{\max_{x \in V} D^2(x,\mu)}{\sum_{x\in V}D^2(x,\mu)} \frac{\Phi^1_{OPT}}{\Phi^k_{OPT}}$  $\beta = \frac{\Phi^1_{OPT}}{\Phi^k_{OPT}}$  Under certain assumptions, $\gamma = O(k\log^2 N)$ and $\beta = O(k)$| [1]
> |  $\text{AFK-\small{MC}}^2$   |  $O(Nd) + O\left(\frac{1}{\varepsilon}k^2 d \log \frac{k}{\varepsilon}\right)$  $O(Nd) + O(k^3 d \log k)$ | $\mathbb{E}[\Phi] \leq O(\log k) \Phi^k_{OPT} + 2\varepsilon \Phi^1_{OPT}$| $\varepsilon \in (0,1)$ is an error parameter  Under certain assumptions $\varepsilon = \Omega(1/k)$ gives $O(\log k)$ approximation | [2]
> |  $\text{R\small{EJECTION}S\small{AMPLING}}$   | $O(Nd \log (\zeta d) +  N \log N\log (\zeta d)) + O\left(kc^2 d^3 \log \zeta (N \log \zeta)^{O(1/c^2)} \right)$  | $\mathbb{E}[\Phi] \leq O(c^6\log k) \Phi^k_{OPT}$ |$\zeta$ is the aspect ratio i.e, the ratio of the maximum to minimum interpoint distance  $c > 1$ is a parameter which controls the runtime - approximation quality tradeoff. | [3]
>
> In the above table , $\Phi$ represents the cost of the centers returned by the algoritm,  $\Phi^k_{OPT}$ represents the optimal $k$ means cost. $\mu$ is the mean of the dataset.  Since we are discussing $k$ means++ like seeding, all the cluster centers are from the dataset itself.
>
>
> We would be happy to address any other concerns , clarifications or comments the reviewer may have.
>
> > References
> >
> > [1] Olivier Bachem, Mario Lucic, S. Hamed Hassani, and Andreas Krause. Approximate k-means++
> in sublinear time. Proceedings of the AAAI Conference on Artificial Intelligence, 30(1), Feb.
> 2016b. doi: 10.1609/aaai.v30i1.10259. URL https://ojs.aaai.org/index.php/
> AAAI/article/view/10259.
> >
> > [2]Olivier Bachem, Mario Lucic, Hamed Hassani, and Andreas Krause. Fast and provably good
> seedings for k-means. In D. Lee, M. Sugiyama, U. Luxburg, I. Guyon, and R. Garnett
> (eds.), Advances in Neural Information Processing Systems, volume 29. Curran Associates, Inc.,
> 2016a. URL https://proceedings.neurips.cc/paper_files/paper/2016/
> file/d67d8ab4f4c10bf22aa353e27879133c-Paper.pdf.
> >
> > [3] Vincent Cohen-Addad, Silvio Lattanzi, Ashkan Norouzi-Fard, Christian Sohler, and Ola
> Svensson. Fast and accurate k-means++ via rejection sampling. In H. Larochelle,
> M. Ranzato, R. Hadsell, M.F. Balcan, and H. Lin (eds.), Advances in Neural Information Processing Systems, volume 33, pp. 16235–16245. Curran Associates, Inc 2020. URL https://proceedings.neurips.cc/paper_files/paper/2020/file/babcff88f8be8c4795bd6f0f8cccca61-Paper.pdf.

---

> ### Author Response · Authors · 2024-11-25
> **Request for Rebuttal Feedback - Discussion Period closing soon**
>
> Dear Reviewer D44y,
>
> Thank you for your valuable feedback on our submission. As the discussion phase closes soon, we would greatly appreciate your thoughts on our revision. Early feedback would allow us time to address any remaining concerns you may have.
>
> Thank you for your time and consideration.
>
> Best regards,
>
> Authors

---

> > ### Comment · Reviewer_D44y · 2024-11-26
> >
> > I really appreciate the effort from the author/s to make suggested changes, especially including the comparison table, and thorough response. Please also include some of these discussion about the quadratic dependence on aspect ratio in the next version as appropriate. As my original score of 6 indicates, I am positive about the paper. But still guarded to fully approve an acceptance score as the quadratic dependency on the aspect ratio makes me cautious  about the broad applicability.  I will maintain my score. Thank you.

---

> > > ### Author Response · Authors · 2024-11-27
> > >
> > > Dear Reviewer D44y,
> > >
> > > Thanks a lot for going through our response. We have included the discussions in the revised submission under the section of Comparison with previous work. We appreciate your constructive feedback and positive evaluation of our submission.
> > >
> > > Best Regards,
> > >
> > > Authors.

---

### Official Review · Reviewer_9Gxa · 2024-11-13

**Soundness:** 3
**Presentation:** 3
**Contribution:** 2
**Rating:** 6
**Confidence:** 3

**Summary:**

This paper provides a quantum algorithm as well as a dequantized (i.e quantum inspired classical) version of it for the problem of $D^2$ sampling. This type of sampling is important for the kmeans++ algorithm and is defined as sampling points in $\mathbb{R}^d$ with probability proportional to the nearest center from a set $C$ of cluster centers in $\mathbb{R}^d$. The classical implementation, which gives a $O(log k)$ approximation guarantee, involves a preprocessing step that builds a sample-query access data structure in time O(Nd). Depending on the implementation, the running time after this data structure is set up is $\tilde{O}(\zeta^2k^2d)$ or $\tilde{O}(\zeta^6k^2)$, where $\zeta$ is the ratio between the largest and smallest pairwise distance in the pointset. The results are incomparable with prior work, since there is no algorithm that is the best in all parameter regimes. This work does asymptotically improve on all prior works for sufficiently small values of $\zeta$ and $k$ though. The authors also give a quantum fixed parameter approximation scheme to get a $(1+\varepsilon)$ approximation for the k-means problem for any $\varepsilon>0$. The k-means problem has been shown to be NP-hard to approximate below a 1.07 constant factor due to prior work and therefore, this approximation scheme has exponential dependence on $1/\varepsilon$, while it is efficient in terms of the other parameters. Finally, the authors provide experiments that verify the performance of the algorithms and make the comparison with the classical k-means++ algorithm.

**Strengths:**

The paper gives novel algorithms for k-means, which is a fundamental problem for machine learning. The paper is also fairly well written.

**Weaknesses:**

The performance of the algorithm is only better in certain parameter regimes compared to prior work.

Minor comments:
-Line 94: I would call it “result” instead of “observation”
-In many places throughout the paper, I believe the tilde notation is a bit abused. It should only be used to hide factors logarithmic in the function displayed, not arbitrary ones as in Theorem 1 for example.

**Questions:**

How can someone choose which algorithm to use between the ones in Theorem 2 and Theorem 3? For example, is there an efficient way to compute $\zeta$ in advance?

---

> ### Author Response · Authors · 2024-11-15
> **We thank you for your positive comments on our work and appreciate your valuable and  constructive feedback**
>
> We discuss and address the issues pointed out in the review :
>
> >The performance of the algorithm is only better in certain parameter regimes compared to prior work.
>
> That is indeed true. The runtime of QI-$k$-means++ is $O(Nd)$ (pre-processing) + $O(\zeta^2 k^2 d \log Nd)$ and we expect it to perform better when the dataset has large $N$ , small $k$ and a bounded aspect ratio (for example, the binary MNIST dataset). We would also like to point out that due to the setup of the sample and query access (SQ) data structures in the pre-processing step, our algorithm can also handle fast data addition / updation in $O(\log Nd)$ time as compared with previous work.
>
> We also see from the previous work that there is no one single algorithm which performs best in all scenarios and would be adding a table of comparision for runtimes, approximation guarantees and the quantities on which the runtimes depend.
>
>
> | Algorithm | Runtime | Approximation Guarantee | Key Parameters| Reference |
> |----------|----------|----------|------------------------------|--|
> |     $\text{QI-K\small{MEANS}++}$      | $O(Nd) + O(\zeta^2k^2 d \log k \log Nd)$     | $\mathbb{E}[\Phi] \leq O(\log k) \Phi^k_{OPT}$ | $\zeta$ is the aspect ratio i.e, the ratio of the maximum to minimum interpoint distance| This Work
> |  $\text{K-\small{MC}}^2$     |  $ O(\gamma k^2 d \log (k \beta) )$ ,  $ O(k^3 d \log^2 N \log k)$  | $\mathbb{E}[\Phi] \leq O(\log k) \Phi^k_{OPT}$ |  $\gamma = 4 N\frac{\max_{x \in V} D^2(x,\mu)}{\sum_{x\in V}D^2(x,\mu)} \frac{\Phi^1_{OPT}}{\Phi^k_{OPT}}$ ,  $\beta = \frac{\Phi^1_{OPT}}{\Phi^k_{OPT}}$ ,  Under certain assumptions, $\gamma = O(k\log^2 N)$ and $\beta = O(k)$| [1]
> |  $\text{AFK-\small{MC}}^2$   |  $O(Nd) + O\left(\frac{1}{\varepsilon}k^2 d \log \frac{k}{\varepsilon}\right)$ ,  $O(Nd) + O(k^3 d \log k)$ | $\mathbb{E}[\Phi] \leq O(\log k) \Phi^k_{OPT} + 2\varepsilon \Phi^1_{OPT}$| $\varepsilon \in (0,1)$ is an error parameter, Under certain assumptions $\varepsilon = \Omega(1/k)$ gives $O(\log k)$ approximation | [2]
> |  $\text{R\small{EJECTION}S\small{AMPLING}}$   | $O(Nd \log (\zeta d) +  N \log N\log (\zeta d)) + O\left(kc^2 d^3 \log \zeta (N \log \zeta)^{O(1/c^2)} \right)$  | $\mathbb{E}[\Phi] \leq O(c^6\log k) \Phi^k_{OPT}$ |$\zeta$ is the aspect ratio i.e, the ratio of the maximum to minimum interpoint distance  $c \geq 1$ is a parameter which controls the runtime - approximation quality tradeoff. | [3]
>
> In the above table , $\Phi$ represents the cost of the centers returned by the algoritm,  $\Phi^k_{OPT}$ represents the optimal $k$ means cost. $\mu$ is the mean of the dataset.  Since we are discussing $k$ means++ like seeding, all the cluster centers are from the dataset itself.
>
>
>
>
>
> > Minor comments: -Line 94: I would call it “result” instead of “observation” In many places throughout the paper, I believe the tilde notation is a bit abused. It should only be used to hide factors logarithmic in the function displayed, not arbitrary ones as in Theorem 1 for example.
>
> Thanks for pointing that out. We will make the necessary adjustments in the revised version and also be more explicit about the factors hidden in the $\tilde{O}(\cdot)$ notation while stating our results.
>
>
> > How can someone choose which algorithm to use between the ones in Theorem 2 and Theorem 3? For example, is there an efficient way to compute $\zeta$ in advance?
>
> The aspect ratio parameter $\zeta$ can be efficiently computed in advance, since it is the ratio of the maximum to minimum interpoint distance. $\zeta$ can be computed exactly in $O(N\log N)$ deterministic time [4] and in $O(N)$ time using randomization [5]. We can also get an upper bound on the maximum interpoint distance by  choosing any point of the dataset and computing twice the maximum distance from all other points in $O(N)$ time. We can then compare $d$ and $\zeta$ to appropriately decide on the algorithm to be used.
>
> We would be happy to address any other concerns , clarifications or comments the reviewer may have.

---

> ### Author Response · Authors · 2024-11-15
>
> > References
> >
> > [1] Olivier Bachem, Mario Lucic, S. Hamed Hassani, and Andreas Krause. Approximate k-means++
> in sublinear time. Proceedings of the AAAI Conference on Artificial Intelligence, 30(1), Feb.
> 2016b. doi: 10.1609/aaai.v30i1.10259. URL https://ojs.aaai.org/index.php/
> AAAI/article/view/10259.
> >
> > [2]Olivier Bachem, Mario Lucic, Hamed Hassani, and Andreas Krause. Fast and provably good
> seedings for k-means. In D. Lee, M. Sugiyama, U. Luxburg, I. Guyon, and R. Garnett
> (eds.), Advances in Neural Information Processing Systems, volume 29. Curran Associates, Inc.,
> 2016a. URL https://proceedings.neurips.cc/paper_files/paper/2016/
> file/d67d8ab4f4c10bf22aa353e27879133c-Paper.pdf.
> >
> > [3] Vincent Cohen-Addad, Silvio Lattanzi, Ashkan Norouzi-Fard, Christian Sohler, and Ola Svensson. Fast and accurate k-means++ via rejection sampling. In H. Larochelle,
> M. Ranzato, R. Hadsell, M.F. Balcan, and H. Lin (eds.), Advances in Neural Information Processing Systems, volume 33, pp. 16235–16245. Curran Associates, Inc.,
> >2020. URL https://proceedings.neurips.cc/paper_files/paper/2020/
> >file/babcff88f8be8c4795bd6f0f8cccca61-Paper.pdf.
> >
> > [4] M. I. Shamos and D. Hoey, "Closest-point problems," 16th Annual Symposium on Foundations of Computer Science (sfcs 1975), USA, 1975, pp. 151-162, https://doi.org/10.1109/SFCS.1975.8
> >
> > [5] Dietzfelbinger, M., Hagerup, T., Katajainen, J., & Penttonen, M. (1997). A Reliable Randomized Algorithm for the Closest-Pair Problem. Journal of Algorithms, 25(1), 19–51. https://doi.org/10.1006/jagm.1997.0873

---

> ### Author Response · Authors · 2024-11-25
> **Request for Rebuttal Feedback - Discussion Period closing soon**
>
> Dear Reviewer 9Gxa,
>
> Thank you for your valuable feedback on our submission. As the discussion phase closes soon, we would greatly appreciate your thoughts on our revision. Early feedback would allow us time to address any remaining concerns you may have.
>
> Thank you for your time and consideration.
>
> Best regards,
>
> Authors

---

> ### Comment · Reviewer_9Gxa · 2024-11-26
>
> I want to thank the authors for the reply and the changes to the manuscript. I will keep my positive evaluation of the paper.

---

> > ### Author Response · Authors · 2024-11-27
> >
> > Dear reviewer 9Gxa,
> >
> > Thanks a lot for going through our response and for your positive evaluation of our submission.
> >
> > Best Regards,
> >
> > Authors.

---

### Author Response · Authors · 2024-11-19
**Revised Version of our Submission**

Dear Reviewers,

We thank everyone for the constructive and thoughtful feedback provided in their reviews for helping us improve our work. We have tried our best to incorporate the suggestions provided
by the reviewers and updated our submission. The places where changes were made are currently highlighted in red for the reviewers to see. We list down the key alterations made as follows :

1. We have included a new section on comparison with previous work. This includes a table of relevant past works on speeding up the kmeans++ seeding, as suggested by reviewer D44y.  We also discuss the tradeoffs arising in this algorithmic landscape.
2. We have included a conclusion and future work section, which also includes the very interesting fact pointed out by reviewer 66Ux, that all of the fast versions of kmeans++ (including both past work and our work) perform worse when the clusters are too far apart. We think that improving the dependence on the aspect ratio like quantities for the algorithms related to this problem is an interesting open problem. It is also an interesting research direction to see whether we can better understand the tradeoff between such aspect ratio like quantities and the data size.
4. We state our results more explicitly (instead of using \tilde notation) and clarified that the aspect ratio is efficiently computable so that we can have an idea about the performance of the algorithm in advance, as suggested by reviewer 9Gxa.
5. We have included a quantum preliminaries section in the appendix as suggested by reviewer ZoT4 so that our work is more accessible to a broader range of audience at ICLR.
6. The abstract has been made more concise without changing the content.

We will be more than happy to engage in further discussions and  make subsequent improvements to our submission.

---

### Meta-Review · Area_Chair_xjbT · 2024-12-07

**Metareview:**

This paper gave a quantum algorithm for D^2-sampling in clustering algorithms. The main contribution is on the theoretical side, in the sense that the proposed quantum algorithm has provable speedup over existing classical algorithms. Numerical experiments also demonstrated the benign performance of the proposed algorithm.

The reviewers in general found the results favorable for ICLR 2025. During the rebuttal period, the authors further updated the manuscript by 1) adding a new section on comparison with previous work, 2) adding a conclusion and future work section, 3) a table summarizing all results, and 4) adding a preliminary section for better accessible to a broader range of audience at ICLR. These further improved the paper, and the decision is to accept this paper at ICLR 2025.

**Additional Comments On Reviewer Discussion:**

There were adequate discussions during the rebuttal period - the authors further clarified comparison with previous work, the result proved by this new paper, and many others.

---

### Decision · Program_Chairs · 2025-01-22

Accept (Poster)